# Sparse Regression with $\ell_0$ Constraints for $\alpha$-Mixing Time Series: Algorithms and Guarantees

**Ruoxin Yuan**[1]  **Lijun Ding**[2]

## Abstract

Exact sparse methods based on $\ell_0$ constraints are increasingly used for interpretable and scalable time series modeling, where one aims to recover a small set of informative lags/factors while maintaining strong predictive performance and low computational cost. Despite their empirical success, finite-sample and computational guarantees for such methods under temporal dependence remain limited. In this paper, we study $\ell_0$-constrained least squares for time series generated by $\alpha$-mixing stationary Gaussian processes with sparse coefficients. We establish high-probability restricted strong convexity/smoothness (RSC/RSS) for the empirical quadratic loss. Leveraging these conditions, we derive nonasymptotic statistical guarantees and computational complexities for a series of exact sparse methods, including iterative hard thresholding (IHT). We apply our theoretical results to Gaussian vector autoregressive (VAR) models and obtain new guarantees. Experiments on synthetic sparse VAR models and real-world mobility time series demonstrate that exact sparse methods recover lag structure more accurately and interpretably than some classical methods, while achieving comparable prediction error with substantially lower computational cost.

## 1. Introduction

Time series modeling plays a central role in domains such as traffic flow (Vlahogianni et al., 2014), product sales (Fildes et al., 2022), and energy consumption (Hong & Fan, 2016), as they help decision-makers to anticipate system behavior, detect disruptions, and make informed operational decisions. Many real-world time series exhibit rich temporal structure, including periodicity, seasonality, and anomalies. Detecting and characterizing these patterns in a data-driven way is essential for effective time series modeling, with applications in human activities (Aledavood et al., 2022; Piccardi et al., 2024), urban human mobility (Chen et al., 2025a;b), and global climate variables (Chen et al., 2025b).

**Sparse regression (SR) for capturing temporal structure.** To capture the underlying temporal structure, in this work, we consider sparse regression problems in a time series setting. The goal is to learn a sparse linear relationship between a regressor and a regressand time series, where only a small subset of the regressor affects the regressand. Such models are widely used in practice to capture temporal structure (Davis et al., 2016; Nicholson et al., 2017). Sparsity reflects the common empirical observation that temporal dependence is often driven by a few dominant effects, and it also enhances models' interpretability. For example, planners and operators often seek to identify a few factors that primarily explain current demand. Likewise, in an autoregressive setting in human mobility (Chen et al., 2025a), we expect a small number of dominant lag effects, which correspond to interpretable cycles such as 24-hour (daily) and 168-hour (weekly) periodicities (see Section 4.2).

**SR with $\ell_1$ regularization.** To recover the underlying sparse relationship, substantial progress has been made using the convex $\ell_1$-regularized least squares estimation, i.e., LASSO (Hastie et al., 2015), for dependent data in time series; see Section 1.1 for a detailed discussion. In particular, Basu & Michailidis (2015) and Wong et al. (2020) investigate the finite-sample guarantees for LASSO under dependence and show that it can achieve guarantees similar to the independent data case.

**SR with $\ell_0$ constraints: a gap between theory and practice.** By contrast, theoretical understanding of *nonconvex $\ell_0$-regularized/constrained* least squares estimation for time series remains limited, despite its computational efficiency and recent empirical success in interpretability (Chen et al., 2025a;b). While many iterative algorithms for SR with $\ell_0$ constraints have been developed, such as iterative hard thresholding (IHT) (Blumensath & Davies, 2008) and CoSaMP (Needell & Tropp, 2009), and they are often computationally efficient because they maintain exactly sparse it-

---

[1]Fudan University [2]University of California San Diego. Correspondence to: Lijun Ding <l2ding@ucsd.edu>.

*Proceedings of the 43rd International Conference on Machine Learning*, Seoul, South Korea. PMLR 306, 2026. Copyright 2026 by the author(s).

erates throughout, their theoretical guarantees are restricted to the i.i.d. setting. Recent empirical studies (Chen et al., 2025a;b) examine autoregressive (AR) models from an interpretable machine learning perspective and show that exact sparse regression can recover simple, human-readable temporal structures that align with domain knowledge, yet no statistical guarantees are provided in this work.

In short, exact sparse methods in practice (i) have substantially reduced runtime and memory usage for comparable estimation accuracy to LASSO, and (ii) recover sparse lag structures more accurately, leading to stronger interpretability. However, existing theory largely focuses on the i.i.d. setting. This restriction limits practitioners' ability to apply these methods with confidence to time series data, resulting in a gap between theory and practice.

**Our contributions.** To address this challenge, we consider time series generated by stationary Gaussian processes with $\alpha$-mixing dependence and an unknown *sparse parameter* (see Section 2 for the formal setup). Within this framework, we close the gap in the following ways:

- **Establishing structural conditions RSC/RSS:** In Proposition 3.2, based on the mixing and Gaussian condition, we establish the key *RSC/RSS properties* (Definition 3.1) of the least-square objective in Problem (2), laying the foundation for analyzing many exact sparse methods.

- **Algorithmic and statistical guarantees:** In Theorem 3.3, we show that a series of exact sparse methods, such as IHT and Subspace Pursuit, indeed recover the underlying sparse parameter in a logarithmic number of steps up to the minimax statistical estimation error.

- **Application to Vector AR (VAR):** In Section 3.3, we specialize our theory to Gaussian VAR models, one of the most popular time series models, obtaining consistent estimation guarantees and characterizing how the estimation error depends on interpretable model parameters, such as the condition number of the innovation covariance.

- **Synthetic and real-world data validation:** In Section 4, experiments on synthetic sparse VAR processes and large-scale real-world time series show that exact sparse solvers recover lag structure more accurately and interpretably than LASSO, while achieving comparable predictive performance with substantially lower runtime.

To the best of our knowledge, this is the first work to provide finite-sample theoretical analysis of exact sparse methods in a genuinely time series setting under mild assumptions.

**Paper organization.** The rest of the paper is organized as follows. Section 1.1 reviews related work and introduces the notation. Section 2 presents the problem setup and the assumptions for stationary Gaussian $\alpha$-mixing time series. In Section 3, we develop the main theory: we establish

RSC/RSS for the empirical objective, analyze representative algorithms, and specialize the results to Gaussian VAR models. Section 4 reports experiments on synthetic sparse VAR processes and real-world time series, with additional results deferred to the appendix. We conclude in Section 5.

## 1.1. Related Work and Notations

In this section, we review related work and summarize the key notation used throughout the paper.

**Sparse time series regression methods and theories.** A substantial literature studied theoretical properties of high-dimensional time series regression with $\ell_1$-regularization. Basu & Michailidis (2015) develop non-asymptotic error bounds for LASSO estimators in high-dimensional VAR models by verifying the key concentration and restricted-eigenvalue conditions. In the univariate setting, Nardi & Rinaldo (2011) analyze LASSO for autoregressive processes with growing lag order, establishing model-selection, estimation, and prediction consistency. For multivariate systems, Kock & Callot (2015) derive oracle inequalities for adaptive LASSO in high-dimensional VARs, while Wong et al. (2020) and Masini et al. (2022) extend these results to weakly dependent and heavy-tailed innovations. Beyond estimation, Adamek et al. (2023) develop desparsified LASSO inference for high-dimensional linear time series models under near-epoch dependence, obtaining asymptotically normal estimators and valid confidence intervals. Li et al. (2021) study high-dimensional contextual linear bandits using an $\ell_1$-based confidence-set method, showing that sparse estimators can also be analyzed with adaptively collected sequential data. Collectively, these works provide a rich theory for *convex*, primarily $\ell_1$-based, estimators under temporal dependence. But they do not address statistical and computational guarantees for exact sparse methods, which are intrinsically *nonconvex* and are the focus of this paper.

**Interpretable modeling of temporal patterns.** A complementary line of empirical work uses sparse and structured modeling to explicitly quantify temporal patterns such as seasonality, periodicity, and structured lag dependence. Baek et al. (2017) propose sparse seasonal and periodic VAR models where regularization selects dominant components. In the context of irregularly sampled astronomical light curves, Kato & Uemura (2012) cast period estimation as a frequency-domain sparse regression problem and use LASSO to recover a few dominant frequencies, enhancing the detection of complex periodic signals. Structured penalties, such as lag-based and own/other-grouped regularization in the VARX-L family, leverage lagged design to enhance interpretability and forecasting (Nicholson et al., 2017; 2020). Recently, unlike past work that used convex methods, Chen et al. (2025a;b) propose nonconvex exact sparse autoregression (SAR) for better interpretability and apply it to peri-

odicity quantification in urban human mobility and global climate variables. These approaches demonstrate how sparsity and structure can be leveraged to extract interpretable temporal patterns, but they largely emphasize modeling and empirical performance, and provide limited theoretical analysis, particularly regarding exact sparse methods.

**$\ell_0$-Constrained exact sparse regression.** Exact sparse regression via $\ell_0$-constraints, also known as best subset selection, is NP-hard in general (Natarajan, 1995). Nevertheless, modern exact–approximate hybrids have made best subset selection practically viable at moderate scales: mixed-integer optimization (MIO) can deliver certifiable global solutions, while local search combined with coordinate descent yields fast, high-quality solutions with strong empirical and theoretical support (Bertsimas et al., 2016; Hazimeh & Mazumder, 2020; Hazimeh et al., 2023). For large-scale exact sparse regression with theoretical guarantees, methods typically require restricted isometry or related conditions, including greedy-type algorithms (Tropp & Gilbert, 2007; Needell & Tropp, 2009; Dai & Milenkovic, 2009), as well as projection-based methods (Blumensath & Davies, 2009; Foucart, 2011; Yuan et al., 2018). Noisy variants of IHT have also been used as sparse regression oracles in adaptive sequential problems, such as the jointly differentially private high-dimensional linear bandit algorithm of Roy et al. (2025). However, existing guarantees for large-scale settings largely assume that the underlying data are i.i.d., unlike typical time series data, where temporal dependence is crucial.

In short, our work aims to fill a gap in the theoretical analysis of scalable and interpretable exact sparse methods in the time series setting, where temporal dependence is critical.

**Notation.** For a symmetric matrix $M$, let $\lambda_{\max}(M)$ and $\lambda_{\min}(M)$ denote its maximum and minimum eigenvalues, respectively. We denote the condition number by $\kappa(M) := \lambda_{\max}(M)/\lambda_{\min}(M)$. For any square matrix $M$ with rank $d$, let $\lambda_i(M)$, $i = 1, \ldots, d$ denote its eigenvalues (from largest to smallest). Then $\rho(M)$ denotes its spectral radius $\max_i\{|\lambda_i(M)|\}$. For any vector $v \in \mathbb{R}^p$, $\|v\|_q$ denotes its $\ell_q$ norm $(\sum_{i=1}^p |v_i|^q)^{1/q}$, and we use $\|v\|_0$ and $\|v\|_\infty$ to denote $\sum_{i=1}^p \mathbf{1}\{v_i \neq 0\}$ and $\max_i |v_i|$, respectively. Similarly, for any matrix $M$, $\|M\|_0 = \|\text{vec}(M)\|_0$ where $\text{vec}(M)$ is the vector obtained by concatenating the rows of $M$. Let $\|M\|$, $\|M\|_\infty$ and $\|M\|_F$ denote its operator norm $\sqrt{\lambda_{\max}(M^\top M)}$, entry-wise $\ell_\infty$ norm $\max_{i,j} |M_{i,j}|$, and Frobenius norm $\sqrt{\text{tr}(M^\top M)}$, respectively. We say that matrix $M$ (vector $v$) is $s$-sparse if $\|M\|_0 = s$ ($\|v\|_0 = s$). We use $v^\top$ and $M^\top$ to denote the transposes of $v$ and $M$. For matrix $M \in \mathbb{R}^{p \times q}$, $1 \leq i \leq p, 1 \leq j \leq q$, we define $M_{i:} := e_i^\top M$ and $M_{:j} := M e_j$, where $e_i$ is the $i$-th standard vector (i.e., all zeros except a one in the $i$-th coordinate).

## 2. Preliminaries

In this section, we introduce the problem setup, the main optimization problem, the empirical least squares objective that governs the design of methods in later sections, and the assumptions needed for later theoretical analysis.

**Time series model and parameter of interest.** Consider a time series $X = [X_1, \ldots, X_T]^\top$, and a target series $Y = [Y_1, \ldots, Y_T]^\top$, where $X_t \in \mathbb{R}^p$ and $Y_t \in \mathbb{R}^q$ for $1 \leq t \leq T$. A common task is to predict $Y_t$ given $X_t$ under a linear model. To formulate the problem and identify the parameter matrix of interest, $\Theta^* \in \mathbb{R}^{p \times q}$, we define the following population least squares problem:

$$\Theta^* = \underset{\Theta \in \mathbb{R}^{p \times q}}{\arg\min} \, \mathbb{E}\left(\left\|Y_t - \Theta^\top X_t\right\|_2^2\right). \quad (1)$$

Under the strict stationarity assumption on $(X_t, Y_t)$ (Assumption 2.2 below), $\Theta^*$ is well-defined in the sense that it is unique and independent of $t$. Let $W := Y - X\Theta^*$ denote the matrix of residuals.

**Empirical least squares.** To recover $\Theta^*$, we consider the following empirical least squares problem with an $\ell_0$ constraint on the parameter matrix:

$$\min_{\Theta \in \mathbb{R}^{p \times q}} f\left(\text{vec}(\Theta)\right), \quad \text{s.t.} \quad \|\Theta\|_0 \leq s, \quad (2)$$

$$\text{with } f(\theta) = \frac{1}{2T} \left\|\tilde{y} - \tilde{X}\theta\right\|_2^2, \quad (3)$$

where $\theta := \text{vec}(\Theta) \in \mathbb{R}^{pq}$, $\tilde{y} := \text{vec}(Y) \in \mathbb{R}^{Tq}$, and $\tilde{X} := I_q \otimes X \in \mathbb{R}^{Tq \times pq}$. The sparsity $s \in \mathbb{Z}^+$ specifies the maximum number of nonzero entries in the matrix variable $\Theta$. We require $s \geq \|\Theta^*\|_0$ to ensure $\Theta^*$ is feasible for (2).

Since Problem (2) is nonconvex due to the sparsity constraint, establishing statistical guarantees for its minimizer is not sufficient, as the minimizer is not computationally tractable. Instead, as we develop in Section 3.2, we analyze the iterates of practical algorithms and show they converge to $\Theta^*$ exponentially fast under assumptions stated below.

**Assumptions.** To recover $\Theta^*$ in the high-dimensional regime where $T < pq$ and to obtain an interpretable model, we impose the following sparsity assumption on $\Theta^*$.

**Assumption 2.1** (Sparsity). Denote $\theta^* = \text{vec}(\Theta^*)$. The true parameter matrix $\Theta^*$ is $s^*$-sparse; that is, $\|\theta^*\|_0 = s^*$.

Following standard practice in time series analysis (Brockwell & Davis, 1991; Hamilton, 1994; Brillinger, 2001) and to ensure the problem is well posed, we state our second assumption on stationarity.

**Assumption 2.2** (Stationarity). The process $\{(\boldsymbol{X}_t, \boldsymbol{Y}_t)\}_{-\infty}^{\infty}$ is strictly stationary; that is, for any integers $m$ and nonnegative integers $n, t$,

$$((\boldsymbol{X}_m, \boldsymbol{Y}_m), \ldots, (\boldsymbol{X}_{m+n}, \boldsymbol{Y}_{m+n}))$$
$$\stackrel{d}{=} ((\boldsymbol{X}_{m+t}, \boldsymbol{Y}_{m+t}), \ldots, (\boldsymbol{X}_{m+n+t}, \boldsymbol{Y}_{m+n+t})),$$

where "$\stackrel{d}{=}$" denotes equality in distribution. Moreover, the covariance matrices of $\boldsymbol{X}_t$ and $\boldsymbol{Y}_t$, denoted as $\Sigma_{\boldsymbol{X}}$ and $\Sigma_{\boldsymbol{Y}}$, exist, and $\Sigma_{\boldsymbol{X}}$ is positive definite.

While the blanket stationary assumption ensures $\boldsymbol{\Theta}^*$ is well-defined, to quantitatively describe the temporal dependence of time series, we consider the mixing conditions, which are originally introduced by Rosenblatt (1956) to extend classical limit theorems from the i.i.d. case and have since become a standard tool in the probability and statistics literature (Kolmogorov & Rozanov, 1960; Doukhan, 1994; Bradley, 2005; Merlevède et al., 2011). In this work, we focus on $\alpha$-**mixing Gaussian processes** stated below. We believe extensions to heavy-tailed settings are feasible given existing work (Wong et al., 2020; Masini et al., 2022).

**Assumption 2.3** (Gaussianity). The process $\{(\boldsymbol{X}_t, \boldsymbol{Y}_t)\}_{-\infty}^{\infty}$ is a Gaussian process; that is, any finite collection of its elements follows a joint Gaussian distribution.

**Assumption 2.4** ($\alpha$-Mixing). The process $(\boldsymbol{X}_t, \boldsymbol{Y}_t)$ is $\alpha$-mixing. Specifically, for lag $\ell \geq 1$, the $\alpha$-mixing coefficient

$$\alpha(\ell) := \sup_{t \in \mathbb{Z}} \sup_{A \in \mathcal{F}_{-\infty}^t, B \in \mathcal{F}_{t+\ell}^{\infty}} |\mathbb{P}(A \cap B) - \mathbb{P}(A)\mathbb{P}(B)| \to 0,$$

as $\ell \to \infty$. Here, the two sigma algebras $\mathcal{F}_{-\infty}^t$ and $\mathcal{F}_{t+\ell}^{\infty}$ are $\mathcal{F}_{-\infty}^t := \sigma\{(\boldsymbol{X}_s, \boldsymbol{Y}_s) : s \leq t\}$ and $\mathcal{F}_{t+\ell}^{\infty} := \sigma\{(\boldsymbol{X}_s, \boldsymbol{Y}_s) : s \geq t + \ell\}$, respectively.

We further define $S_\alpha(T) := \sum_{t=0}^T \alpha(t)$. The process is said to have summable mixing if $\lim_{T \to \infty} S_\alpha(T) < \infty$.

Lastly, to streamline the presentation, we impose the following centering assumption.

**Assumption 2.5** (Centering). The process $(\boldsymbol{X}_t, \boldsymbol{Y}_t)$ is centered; that is, $\forall t$, $\mathbb{E}(\boldsymbol{X}_t) = \boldsymbol{0}_{p \times 1}$, and $\mathbb{E}(\boldsymbol{Y}_t) = \boldsymbol{0}_{q \times 1}$.

In the main text, we shall assume $(\boldsymbol{X}_t, \boldsymbol{Y}_t)_{t=1}^T$ satisfies Assumptions 2.2–2.5. In particular, we have $\boldsymbol{X}_t \sim \mathcal{N}(0, \Sigma_{\boldsymbol{X}})$ and $\boldsymbol{Y}_t \sim \mathcal{N}(0, \Sigma_{\boldsymbol{Y}})$ for all $1 \leq t \leq T$.

The centering assumption is imposed mainly for notational convenience. All our main results in Section 3 remain valid under a minor yet practical modification of (2); see Appendix B for details. Consequently, our theory applies to realistic time series, which are rarely centered at zero.

# 3. $\ell_0$-**Constrained Sparse Regression**

In this section, we first establish the structural properties RSC/RSS under the previous assumptions. Then, we study the widely used algorithms for solving the $\ell_0$-constrained sparse regression problem, focusing on projected gradient methods, exemplified by IHT, and greedy pursuit methods such as CoSaMP and Subspace Pursuit (SP). After obtaining the geometric convergence rates for those exact sparse methods, we apply our framework to VAR.

## 3.1. RSC/RSS under $\alpha$-Mixing and Gaussianity

We analyze several algorithms for solving the $\ell_0$-constrained regression problem (2). The core of the guarantee for these methods is the following restricted strong convexity / smoothness of the empirical objective $f(\boldsymbol{\theta})$ in (3).

**Definition 3.1** (Restricted Strong Convexity / Smoothness). Let $s > 0$ be an integer. A differentiable function $f : \mathbb{R}^p \to \mathbb{R}$ is said to satisfy restricted strong convexity (RSC) with curvature parameter $m_s > 0$, and restricted strong smoothness (RSS) with smoothness parameter $M_s > 0$, if for all $\boldsymbol{x} \neq \boldsymbol{y} \in \mathbb{R}^p$ such that $\|\boldsymbol{x} - \boldsymbol{y}\|_0 \leq s$,

$$m_s \leq \frac{f(\boldsymbol{x}) - f(\boldsymbol{y}) - \langle \nabla f(\boldsymbol{y}), \boldsymbol{x} - \boldsymbol{y} \rangle}{\|\boldsymbol{x} - \boldsymbol{y}\|_2^2} \leq M_s.$$

In compressed sensing and sparse regression scenarios (Candes & Tao, 2005; Candes et al., 2006; Negahban et al., 2012), such conditions are typically established under strong i.i.d. assumptions on the rows of the design matrix $\boldsymbol{X}$. A major technical endeavor of this work is to establish RSC/RSS in the time series setting, where $\boldsymbol{X}$ is constructed from overlapping lag vectors and therefore has highly dependent rows. The result stated below bridges the assumptions on the underlying process with the convergence analyses of exact sparse algorithms.

**Proposition 3.2** (RSC/RSS for $\alpha$-mixing Gaussian processes). *Suppose Assumptions 2.2 - 2.5 hold. There exists a universal constant $c > 0$, such that if sample size*

$$T \geq \frac{\log(p) \max\{120, 16s\}}{c \min\{1, \nu^2\}},$$

*where $\nu = \frac{\lambda_{\min}(\Sigma_{\boldsymbol{X}})}{108\pi S_\alpha(T)\lambda_{\max}(\Sigma_{\boldsymbol{X}})}$, then the function $f$ in Equation (3) satisfies $m_s$-RSC and $M_s$-RSS property with probability at least $1 - 2\exp\left(-\frac{c}{2}T\min\{1, \nu^2\}\right)$, where*

$$m_s = \frac{3}{16}\lambda_{\min}(\Sigma_{\boldsymbol{X}}), \quad M_s = \frac{13}{16}\lambda_{\max}(\Sigma_{\boldsymbol{X}}).$$

Our analysis builds on Wong et al. (2020), which studies a convex formulation for $\alpha$-mixing Gaussian processes. We present a detailed proof of Proposition 3.2 in Appendix A.

## 3.2. Algorithms and Guarantees

We analyze several widely used exact sparse algorithms and establish their corresponding estimation error guarantees.

**Iterative Hard Thresholding (IHT).** IHT is the projected gradient descent applied to Equation (2). Specifically, it generates iterates $\{\boldsymbol{\theta}^k\}_{k \geq 0}$ with step size $\eta > 0$ via

$$\boldsymbol{\theta}^{k+1} = P_s\left(\boldsymbol{\theta}^k - \eta \nabla_{\boldsymbol{\theta}} f(\boldsymbol{\theta}^k)\right), \qquad (4)$$

where the initial parameter is typically set to $\boldsymbol{\theta}^0 = \mathbf{0}$, and $\boldsymbol{\theta}^k$ denotes the $k$-th iterate. The operator $P_s(\boldsymbol{a})$ is the nonlinear orthogonal projection onto the set of $s$-sparse vectors: it keeps the $s$ entries of $\boldsymbol{a}$ with largest magnitudes and sets the rest to zero. If the top-$s$ set is not unique, ties can be resolved either randomly or according to a fixed index order. We summarize the full procedure in Algorithm 1.

---

**Algorithm 1** Iterative Hard Thresholding

---

**Input:** Function $f$, step size $\eta$, number of iterations $K$, sparsity level $s$
1: Initialize $\boldsymbol{\theta}^0 = \mathbf{0}$
2: **for** $k = 0, 1, \ldots, K - 1$ **do**
3: $\quad \boldsymbol{\theta}^{k+1} = P_s\left(\boldsymbol{\theta}^k - \eta \nabla_{\boldsymbol{\theta}} f(\boldsymbol{\theta}^k)\right)$
4: **end for**
**Output:** Final estimator $\boldsymbol{\theta}^K$

---

**Greedy Pursuit with Hard Thresholding.** These methods iteratively expand and refine the support set, thereby constructing an approximation incrementally through locally optimal updates at each step. The general procedure is outlined in Algorithm 2, where $\mathrm{top}_S(\cdot)$ denotes the index set of the $S$ largest-magnitude entries and $|\nabla_{\boldsymbol{\theta}} f(\boldsymbol{\theta}^k)|$ denotes the vector obtained by taking the absolute value of $\nabla_{\boldsymbol{\theta}} f(\theta^k)$ entrywise.

---

**Algorithm 2** Greedy Pursuit with Hard Thresholding

---

**Input:** Function $f$, number of iterations $K$, sparsity level $s$, sparsity expansion level $S$
1: Initialize $\boldsymbol{\theta}^0 = \mathbf{0}$
2: **for** $k = 0, 1, \ldots, K - 1$ **do**
3: $\quad \mathcal{I}^k = \mathrm{supp}(\boldsymbol{\theta}^k) \cup \mathrm{top}_S\left(|\nabla_{\boldsymbol{\theta}} f(\boldsymbol{\theta}^k)|\right)$
4: $\quad \tilde{\boldsymbol{\theta}}^k = P_s\left(\arg\min_{\boldsymbol{\theta},\, \mathrm{supp}(\boldsymbol{\theta}) \subseteq \mathcal{I}^k} f(\boldsymbol{\theta})\right)$
5: $\quad \boldsymbol{\theta}^{k+1} = \arg\min_{\boldsymbol{\theta},\, \mathrm{supp}(\boldsymbol{\theta}) \subseteq \mathrm{supp}(\tilde{\boldsymbol{\theta}}^k)} f(\boldsymbol{\theta})$
6: **end for**
**Output:** Final estimator $\boldsymbol{\theta}^K$

---

For $\alpha$-mixing Gaussian process, we can derive the following estimation guarantees of $\boldsymbol{\theta}^*$ based on the RSC/RSS property.

**Theorem 3.3** (Sample and iteration complexity of exact sparse methods)**.** *Suppose Assumptions 2.1–2.4 hold. There exist absolute constants $c, \tilde{c}, C_1, C_2 > 0$, and a free parameter $b > 0$ such that the following holds. Let $\nu$ be as in Proposition 3.2 and the sample size $T \geq \max\{\frac{\log(p)}{c \min\{1, \nu^2\}} \max\{120, 16s\}, \log(2pq)\sqrt{\frac{b+1}{\tilde{c}}}\}$. Consider either Algorithm 1 or Algorithm 2 run on $f(\boldsymbol{\theta})$ with*

*step size $\eta = \frac{16}{3(8\lambda_{\max}(\Sigma_{\boldsymbol{X}}) + 5\lambda_{\min}(\Sigma_{\boldsymbol{X}}))}$, and assume the following conditions hold:*

- **(IHT)** *Algorithm 1 is invoked with sparsity level*

$$s \geq C_1 \kappa^2(\Sigma_{\boldsymbol{X}}) s^*.$$

- **(Greedy pursuit)** *Algorithm 2 is invoked with sparsity expansion level $S$ and sparsity level*

$$s \geq C_2 \kappa^2(\Sigma_{\boldsymbol{X}}) S \quad and \quad S \geq s^*.$$

*Define $C_{\nabla} := 1 + \frac{2M_{2s^*+1}\sqrt{2s^*}}{m_{2s^*}}$, and*

$$\epsilon_{\mathrm{GP}} := \frac{M_{s+S}^2}{2m_{s+S+s^*}^3} \cdot \frac{S(s+S+s^*)}{s+S-s^*} \cdot C_{\nabla}^2 Q^2 S_\alpha(T)^2 \frac{\log(2pq)}{T}.$$

*Then, with probability at least $1 - 2\exp\left(-\frac{c}{2}T\min\{1, \nu^2\}\right) - 8\exp(-b\log(2pq))$, we know that for any $\epsilon > \epsilon_{\mathrm{GP}}$, and any $k \geq C_1\kappa(\Sigma_{\boldsymbol{X}})\log\left(\frac{f(\mathbf{0})}{\epsilon - \epsilon_{\mathrm{GP}}}\right)$, the estimation error at $k$-th iterate of both methods satisfies*

$$\|\boldsymbol{\theta}^k - \boldsymbol{\theta}^*\|_2 \leq \frac{Q S_\alpha(T)}{\lambda_{\min}(\Sigma_{\boldsymbol{X}})}\sqrt{\frac{(s+s^*)\log(2pq)}{T}} + \sqrt{\frac{2\epsilon}{\lambda_{\min}(\Sigma_{\boldsymbol{X}})}},$$

*where*

$$Q = 16\pi\sqrt{\frac{b+1}{\tilde{c}}}\left(\|\Sigma_{\boldsymbol{X}}\|_2\left(1 + \max_{1 \leq i \leq p}\|\boldsymbol{\Theta}_{:i}^*\|_2^2\right) + \|\Sigma_{\boldsymbol{Y}}\|_2\right).$$

We provide the detailed proof in Appendix C, building on the analysis of Jain et al. (2014).

*Remark* 3.4 (Error rate and sample size requirement)**.** Suppose $\alpha$-mixing coefficients are summable, i.e., $\forall T$, $S_\alpha(T) \leq \tilde{\alpha} < \infty$, the sparsity $s = \mathcal{O}(s^*)$, and the parameter $\tilde{\alpha}$ and $Q/\lambda_{\min}(\Sigma_{\boldsymbol{X}})$ are constants independent of $p, q, s$. Then, with high probability, the method converges exponentially fast to the error rate $\lim_{\epsilon \to \epsilon_{\mathrm{GP}}} \|\boldsymbol{\theta}^{k(\epsilon)} - \boldsymbol{\theta}^*\|_2 \leq \mathcal{O}\left(\sqrt{\frac{s\log(pq)}{T}}\right)$, with sample-size requirement $T = \Theta(s\log(pq))$, almost matching the classical LASSO guarantees under i.i.d. designs (Bickel et al., 2009; Raskutti et al., 2011). We discuss the conditions on $\tilde{\alpha}$ and $Q/\lambda_{\min}(\Sigma_{\boldsymbol{X}})$ in more detail in the next section under the VAR setting. Indeed, we can achieve this rate in $\mathcal{O}\left(\kappa(\Sigma_{\boldsymbol{X}})\log\frac{q\lambda_{\max}(\Sigma_{\boldsymbol{Y}})}{\epsilon}\right)$ iterations with $\epsilon = \mathcal{O}\left(\frac{s\log(pq)}{T\lambda_{\min}(\Sigma_{\boldsymbol{X}})}\right)$.

If $\alpha(l)$ is not summable but other conditions hold, the bound degrades to $\mathcal{O}\left(S_\alpha(T)\sqrt{\frac{s\log(pq)}{T}}\right)$, which is still consistent as long as $S_\alpha(T) \in o(\sqrt{T})$.

## 3.3. Gaussian VAR($d$)

Our previous results apply to several scenarios; here, we illustrate the result for the Gaussian Vector Autoregression with lag $d$ (VAR($d$)).

A sequence of random vectors $\{\boldsymbol{Z}_t\}_{-\infty}^{+\infty}$, where $\boldsymbol{Z}_t \in \mathbb{R}^p$, is a finite-order Gaussian VAR($d$) process if

$$\boldsymbol{Z}_t = \boldsymbol{A}_1 \boldsymbol{Z}_{t-1} + \cdots + \boldsymbol{A}_d \boldsymbol{Z}_{t-d} + \mathcal{E}_t, \qquad (5)$$

where each $\boldsymbol{A}_k \in \mathbb{R}^{p \times p}$ ($k = 1, \ldots, d$) is a sparse coefficient matrix, and the innovations $\mathcal{E}_t$ are $p$-dimensional random vectors from $\mathcal{N}(0, \Sigma_\varepsilon)$. The error covariance satisfies $0 < \lambda_{\min}(\Sigma_\varepsilon) \leq \lambda_{\max}(\Sigma_\varepsilon) < \infty$. Identify $\boldsymbol{X}_t := (\boldsymbol{Z}_{t-1}^\top, \boldsymbol{Z}_{t-2}^\top, \ldots, \boldsymbol{Z}_{t-d}^\top)^\top \in \mathbb{R}^{dp}$, $\boldsymbol{Y}_t := \boldsymbol{Z}_t \in \mathbb{R}^p$, and $\Theta^* = (\boldsymbol{A}_1, \ldots, \boldsymbol{A}_d)^\top \in \mathbb{R}^{dp \times p}$. For the infinite sequence, we observe a finite truncation $(\boldsymbol{Z}_t)_{t=1}^{T+d}$.

Assume the VAR($d$) process is stable; that is, $\det\left(\boldsymbol{I}_{p \times p} - \sum_{k=1}^d \boldsymbol{A}_k z^k\right) \neq 0, \; \forall |z| \leq 1$. We can verify that Assumptions 2.2 - 2.4 hold (see Appendix D for details). Consequently, Proposition 3.2 and Theorem 3.3 hold. Below, we relate the estimation error bound to the innovation covariance $\Sigma_\epsilon$ and the coefficient matrix $\Theta^*$.

Introduce the following companion matrix

$$\mathcal{A} = \begin{pmatrix} \boldsymbol{A}_1 & \boldsymbol{A}_2 & \ldots & \boldsymbol{A}_{d-1} & \boldsymbol{A}_d \\ \boldsymbol{I}_p & 0 & \ldots & 0 & 0 \\ & \ddots & \ddots & \vdots & \vdots \\ & & \boldsymbol{I}_p & 0 & 0 \end{pmatrix} \in \mathbb{R}^{pd \times pd},$$

therefore, it is readily verified that the stability condition is equivalent to the spectral radius $\rho(\mathcal{A}) < 1$. Consequently there exist constants $C_\mathcal{A} > 0$ and $0 < r < 1$ such that

$$\|\mathcal{A}^j\|_2 \leq C_\mathcal{A} \, r^j \quad (j \geq 0).$$

Moreover, as verified in Appendix D, for all $\ell \geq 0$, the $\alpha$-mixing coefficients satisfy $\alpha(\ell) \leq \kappa(\Sigma_{\boldsymbol{X}})^{1/2} C_\mathcal{A} r^\ell$ and the condition number $\kappa(\Sigma_{\boldsymbol{X}}) \leq \kappa(\Sigma_\epsilon) \cdot \frac{C_\mathcal{A}^2}{1-r^2}$.

Consequently,

$$S_\alpha(T) = \sum_{t=0}^T \alpha(t) \leq \sum_{t=0}^T \kappa(\Sigma_{\boldsymbol{X}})^{1/2} C_\mathcal{A} r^t$$

$$\leq \kappa(\Sigma_\epsilon)^{1/2} C_\mathcal{A}^3 \frac{1 - r^{T+1}}{(1-r)(1-r^2)} := \tilde{\alpha},$$

and

$$\frac{Q}{\lambda_{\min}(\Sigma_{\boldsymbol{X}})} = 16\pi \sqrt{\frac{b+1}{\tilde{c}}} \kappa(\Sigma_{\boldsymbol{X}}) \left(2 + \max_{1 \leq i \leq p} \|\Theta_{:i}^*\|_2^2\right)$$

$$\leq 16\pi \sqrt{\frac{b+1}{\tilde{c}}} \kappa(\Sigma_\epsilon) \frac{C_\mathcal{A}^2}{1-r^2} \left(2 + C_\mathcal{A}^2 r^2\right).$$

Therefore, $\tilde{\alpha}$ and $Q/\lambda_{\min}(\Sigma_{\boldsymbol{X}})$ are dimension-independent provided that the condition number of innovation covariance $\Sigma_\epsilon$ and the constants $r$ and $C_\mathcal{A}$ are dimension-independent. Consequently, under the same condition, the estimation error satisfies $\|\boldsymbol{\theta}^k - \boldsymbol{\theta}^*\|_2 = \mathcal{O}\left(\sqrt{\frac{s \log(pq)}{T}}\right)$.

# 4. Experiments

In this section, we conduct simulations and real data analysis, demonstrating the computational and interpretability benefits of exact sparse methods.

## 4.1. Simulation

In this section, we conduct simulation studies on synthetic sparse regression problems to assess $\ell_0$-constrained exact sparse algorithms against the $\ell_1$-penalized LASSO and the oracle least squares estimator on Gaussian VAR data.

**Data.** To evaluate the estimators in the sparse VAR setting, we simulate a $p$-dimensional Gaussian VAR($d$) process as follows. We first generate sparse coefficient matrices $\{\boldsymbol{A}_k\}_{k=1}^d \subset \mathbb{R}^{p \times p}$ by selecting a subset of $s_{\text{active}}$ lags from $\{1, \ldots, d\}$ uniformly at random and setting the remaining $d - s_{\text{active}}$ lag matrices to $\boldsymbol{0}$. For each active lag $k$, we sample $s_{\text{perlag}}$ indices uniformly at random from the $p^2$ entries of $\boldsymbol{A}_k$ and assign i.i.d. weights from $\mathcal{N}(0, \sigma_A^2)$ to the selected entries, with all other entries set to zero. The resulting true sparsity is $s^* = s_{\text{active}} s_{\text{perlag}}$. We then apply an additional procedure to ensure stability, and simulate the VAR trajectory $(\boldsymbol{Z}_t)_{t=1}^T$ as in Equation (5) using a standard burn-in scheme (See Appendix F for further details). Prior to estimation, we standardize each column of $\boldsymbol{X}$ to have unit empirical variance. For all experiments, we partition the time series chronologically into training/validation/test splits of 60%/20%/20%, respectively.

**Algorithms.** We implement a few exact sparse solvers, including (i) IHT, (ii) Subspace Pursuit (SP), and (iii) CoSaMP, each run with sparsity $s$ set to the true sparsity $s^*$. We compare these methods against the $\ell_1$-penalized LASSO, solved using the cvxpy package (Diamond & Boyd, 2016), and an oracle least squares (Oracle LS) baseline that fits least squares on the true support. Implementation details and parameter settings for all methods are provided in Appendix F.

**Evaluation.** We evaluate all methods along four axes.

(i) Runtime: we record the time required to fit each model under its selected hyperparameters as a measure of computational cost.

(ii) Estimation error: $\ell_2$-error $= \|\hat{\boldsymbol{\theta}} - \boldsymbol{\theta}^\star\|_2$.

(iii) Predictive accuracy: mean-squared error on held-out

data, computed as $\mathrm{MSE} = \frac{1}{n}\sum_{i=1}^{n}\|\boldsymbol{Y}_i - \langle\hat{\boldsymbol{\theta}}, \boldsymbol{X}_i\rangle\|_2^2$.

(iv) Support recovery: we report precision and recall. Let $\bar{S} = \{j : \boldsymbol{\theta}_j^\star \neq 0\}$ and $S = \{j : \hat{\boldsymbol{\theta}}_j \neq 0\}$ with $\mathrm{TP} = |S \cap \bar{S}|$. We define $\mathrm{Precision} = \mathrm{TP}/|S|$ and $\mathrm{Recall} = \mathrm{TP}/|\bar{S}|$.

**Experimental Results.** Table 1 reports a representative univariate, 10-sparse VAR setting with $d = 100$ lags. All methods attain comparable prediction MSE. The key distinctions lie not in predictive accuracy but in sparsity control, support recovery, and computational efficiency. While the LASSO achieves high recall, it selects a denser model (32 nonzeros), resulting in low precision and many false positives. In contrast, IHT/SP/CoSaMP enforce exact sparsity and deliver markedly higher precision at the target sparsity level, yielding a more faithful recovery of the underlying lag structure. Moreover, all exact sparse methods run substantially faster than the LASSO.

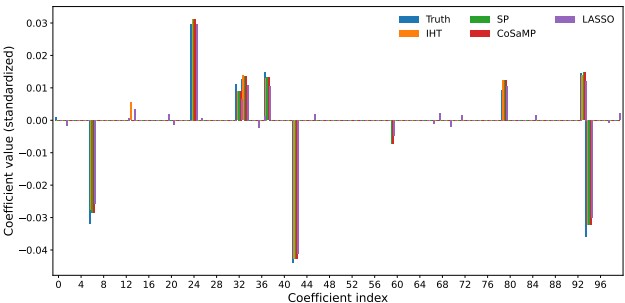

*Figure 1.* Coefficient estimates for the VAR instance in Table 1.

Figure 1 visualizes coefficient recovery for the $p = 1$ case. All methods reliably recover moderate-to-large coefficients, whereas recovery deteriorates for very small signals; in particular, the LASSO tends to distribute mass across many small-magnitude entries, consistent with its denser support. To isolate the effect of signal heterogeneity, we further hard-threshold the ground-truth coefficients at $0.01$ while keeping the overall sparsity unchanged. Under this regime, IHT achieves more accurate support recovery; see Appendix G.1.

We next consider a multivariate VAR with $p = 5$ (Table 2), where the parameter dimension increases and the true sparsity is $s^\star = 20$. The same pattern persists: the oracle and exact sparse methods achieve near-oracle prediction error with sparse solutions, whereas the LASSO again over-selects (85 nonzeros) and exhibits low precision despite high recall. The computational gap is also more pronounced in this higher-dimensional setting, while IHT, SP, and CoSaMP remain fast and maintain strong support recovery. Overall, the simulations support our main message: exact sparse methods can match the predictive performance of $\ell_1$ regularization while providing more accurate support recovery and substantially lower runtime.

The main simulation results in Tables 1 and 2 use the true sparsity level $s = s^*$ for exact sparse methods, which provides a stringent setting for support recovery. Appendix G.2 shows that the methods are robust to moderate sparsity over-specification: prediction remains competitive, recall often improves, and runtime remains below that of LASSO. Appendix G.3 further reports sensitivity experiments with varying VAR stability margins. As the process moves closer to the unit-root boundary, the lagged design covariance becomes more ill-conditioned, and the empirical trends are consistent with the theory: larger condition numbers lead to larger estimation error and weaker support recovery.

### 4.2. Real Data

In this section, we evaluate the performance of exact sparse regression methods on real-world time series data.

**Data.** Dynamic systems such as urban transportation often exhibit pronounced periodic patterns driven by commuting rhythms, business cycles, and recurrent travel demand. As representative examples, we consider ridesharing mobility datasets from New York City and Chicago,[1] both of which display rich daily and weekly periodicities. In the main text, we focus on the NYC dataset, and defer additional results on the Chicago ridesharing dataset to Appendix G.4.

For the NYC dataset, we focus on hourly mobility associated with John F. Kennedy International Airport from February 2019 to December 2024. Airport-related ridesharing trips are known to follow strong daily and weekly rhythms, largely shaped by fixed flight schedules and recurrent passenger flows. Figure 2 plots the hourly number of ridesharing trips, where the series exhibits pronounced daily and weekly

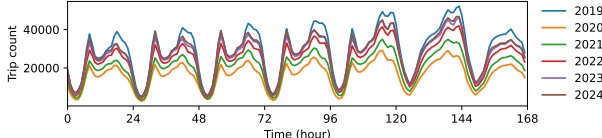

*(a)* Hourly NYC ridesharing activity over a representative week.

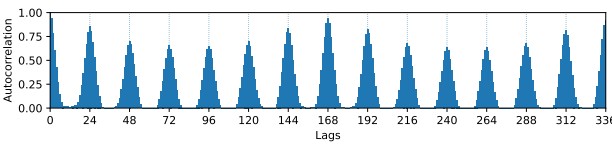

*(b)* Autocorrelation of the hourly NYC ridesharing series, with prominent peaks at multiples of 24 hours.

*Figure 2.* NYC ridesharing exhibits strong periodic structure.

---

[1]We use the dataset provided by Chen et al. (2025a), available at `https://github.com/xinychen/integers`. The underlying raw data are from the TLC trip record data and Transportation Network Providers.

*Table 1.* Simulation results on a univariate sparse Gaussian VAR($d$) model ($p = 1$, $d = 100$, $T = 10{,}000$, $\sigma_A = 0.25$, $\sigma_\varepsilon = 0.2$, $s_{\text{perlag}} = 1$, $s_{\text{active}} = 10$). We compare Oracle LS, LASSO, and exact sparse estimators (IHT, SP, CoSaMP).

| Setting | Method | Runtime | Train MSE | Test MSE | $\ell_2$-error | Precision | Recall | Sparsity level |
|---|---|---|---|---|---|---|---|---|
| | Oracle LS | 0.00 | 0.0414 | 0.0397 | 0.007 | 1.00 | 1.00 | 10 |
| | LASSO | 0.10 | 0.0412 | 0.0399 | 0.015 | 0.28 | 0.90 | 32 |
| VAR | IHT$_{(s=10)}$ | 0.02 | 0.0414 | 0.0397 | 0.029 | 0.90 | 0.90 | 10 |
| | SP$_{(s=10)}$ | 0.02 | 0.0413 | 0.0397 | 0.010 | 0.90 | 0.90 | 10 |
| | CoSaMP$_{(s=10)}$ | 0.01 | 0.0413 | 0.0397 | 0.010 | 0.90 | 0.90 | 10 |

*Table 2.* Simulation results on a multivariate sparse Gaussian VAR($d$) model ($p = 5$, $d = 100$, $T = 10{,}000$, $\sigma_A = 0.25$, $\sigma_\varepsilon = 0.2$, $s_{\text{perlag}} = 2$, $s_{\text{active}} = 10$). We compare Oracle LS, LASSO, and exact sparse estimators (IHT, SP, CoSaMP).

| Setting | Method | Runtime | Train MSE | Test MSE | $\ell_2$-error | Precision | Recall | Sparsity level |
|---|---|---|---|---|---|---|---|---|
| | Oracle LS | 0.00 | 0.0397 | 0.0401 | 0.006 | 1.00 | 1.00 | 20 |
| | LASSO | 0.83 | 0.0397 | 0.0403 | 0.013 | 0.21 | 0.90 | 85 |
| VAR | IHT$_{(s=20)}$ | 0.20 | 0.0397 | 0.0402 | 0.010 | 0.80 | 0.80 | 20 |
| | SP$_{(s=20)}$ | 0.17 | 0.0397 | 0.0401 | 0.010 | 0.85 | 0.85 | 20 |
| | CoSaMP$_{(s=20)}$ | 0.13 | 0.0397 | 0.0401 | 0.010 | 0.85 | 0.85 | 20 |

fluctuations. The sample autocorrelation function shows prominent peaks at multiples of 24 hours, providing clear evidence of recurrent, cycle-driven demand.

**Problem Setup.** To predict ridesharing demand from its history, we specialize our formulation to a univariate AR($d$) model and set $d = 24 \times 8 = 192$ lags to capture daily and weekly dependencies. In this case, problem (2) reduces to

$$\min_{\alpha \in \mathbb{R},\, \boldsymbol{\theta} \in \mathbb{R}^d,\, \|\boldsymbol{\theta}\|_0 \leq s^\star} \frac{1}{2(T-d)} \sum_{t=d+1}^{T} \left( x_t - \sum_{k=1}^{d} \boldsymbol{\theta}_k x_{t-k} - \alpha \right)^2,$$

(6)

where $x_t$ denotes the transformed trip count at time $t$ ($x_t = \log(1 + r_t)$ for raw counts $r_t$), $\boldsymbol{\theta} = (\boldsymbol{\theta}_1, \ldots, \boldsymbol{\theta}_d)^\top \in \mathbb{R}^d$ is the coefficient vector, and $\alpha$ denotes the intercept.

We split the data chronologically into training ($\mathcal{T}_{\text{tr}}$), validation ($\mathcal{T}_{\text{val}}$), and test ($\mathcal{T}_{\text{te}}$) sets with proportions 60%/20%/20%, respectively. Let $\mu_{\text{tr}} := \frac{1}{n_{\text{tr}}} \sum_{t \in \mathcal{T}_{\text{tr}}} x_t$ be the training-set mean and define the centered response and lagged regressors by $x_t^c := x_t - \mu_{\text{tr}}$. We then assemble the training design matrix $\boldsymbol{X}_{\text{tr}} \in \mathbb{R}^{n_{\text{tr}} \times d}$ with columns $\boldsymbol{X}_{\text{tr}, \cdot k} = (x_{d+1-k}^c, \ldots, x_{T-k}^c)^\top$, and perform column standardization so that each column has unit empirical norm. Denote the resulting estimate by $\hat{\boldsymbol{\theta}}$ (on the transformed $x_t$ scale). The intercept is then recovered via $\hat{\alpha} = \mu_{\text{tr}}\left(1 - \sum_{k=1}^{d} \hat{\boldsymbol{\theta}}_k\right)$, which is equivalent to solving Equation (6) with an explicit, unpenalized intercept.

**Algorithms.** We compare exact sparse algorithms with LASSO for estimating the ridesharing AR($d$) model. The LASSO baseline reported in the main text is implemented using `cvxpy`, with the penalty parameter $\lambda$ selected by a ten-point grid search on the validation set. For a fair compar-

ison, the target sparsity levels of IHT, SP, and CoSaMP are set equal to the sparsity level obtained by LASSO, so that the exact sparse methods are evaluated against models of comparable effective complexity. For IHT, we additionally vary the target sparsity level $s$ to examine how performance changes as different numbers of dominant lags are retained.

Since runtime comparisons can be sensitive to implementation details, Appendix G.5 further reports optimized coordinate-descent baselines for LASSO and elastic net under the same data split and validation protocol.

**Experimental Results.** Table 3 summarizes runtime and prediction MSE. Among methods producing a 10-sparse model, the exact sparse solvers are substantially faster than the LASSO and achieve lower test error. IHT exhibits a clear sparsity–fit trade-off: small $s$ yields underfitting, while increasing $s$ improves prediction and approaches the performance of the other sparse estimators. Additional experimental results in Appendix G.5 show that this runtime advantage persists even when compared with LASSO and elastic net implementations based on fast pathwise coordinate descent.

*Table 3.* NYC ridesharing AR($d$) estimation results.

| Method | Runtime | Train MSE | Test MSE | Sparsity |
|---|---|---|---|---|
| LASSO | 1.92 | 0.031 | 0.015 | 10 |
| IHT$_{(s=2)}$ | 0.26 | 0.183 | 0.126 | 2 |
| IHT$_{(s=3)}$ | 0.29 | 0.121 | 0.082 | 3 |
| IHT$_{(s=5)}$ | 0.30 | 0.077 | 0.042 | 5 |
| IHT$_{(s=10)}$ | 0.32 | 0.062 | 0.030 | 10 |
| SP$_{(s=10)}$ | 0.04 | 0.028 | 0.012 | 10 |
| CoSaMP$_{(s=10)}$ | 0.06 | 0.028 | 0.013 | 10 |

Beyond prediction, Figure 3 illustrates how the IHT-selected support set and the objective value evolve as $s$ increases.

For example, at $s = 3$, IHT yields the coefficient vector

$$\hat{\boldsymbol{\theta}}_{\text{IHT}} = \big(\underbrace{0.21}_{\text{lag}=1}, 0, \ldots, \underbrace{0.20}_{\text{lag}=24}, 0, \ldots, \underbrace{0.20}_{\text{lag}=168}, 0, \ldots\big)^{\top},$$

and $\hat{\alpha} = 3.80$. The results align with domain knowledge: lag 1 captures short-term autocorrelation, while lags 24 and 168 capture daily and weekly periodicity. As $s$ increases further, additional nearby lags (e.g., 169) enter the support and the objective value continues to decrease, but the marginal gain becomes smaller, consistent with diminishing returns when adding weaker effects.

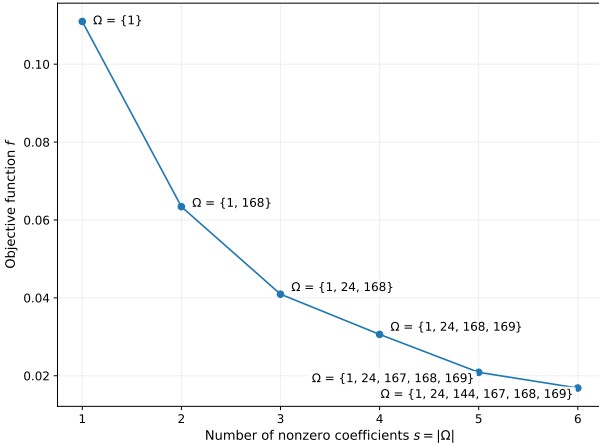

*Figure 3.* Illustration of the IHT estimations on the NYC ridesharing trips at different sparsity levels. The support set and the number of nonzero coefficients are denoted as $\Omega$ and $|\Omega|$, respectively.

In contrast, LASSO yields a 10-sparse solution and selects lags $\{1, 167, 23, 143, 3, 24, 171, 172, 27, 168\}$ (ordered by coefficient magnitude). While this set contains the key periodic lags, it also includes several scattered lags (e.g., 171, 172) whose interpretation is less direct and may reflect shrinkage-induced leakage across correlated features. This comparison illustrates that prediction error alone does not fully capture the quality of a sparse time-series model. For downstream interpretation, the selected support itself is part of the output: a small set of lags should identify the dominant temporal mechanisms rather than merely approximate the predictive span of many correlated lagged features.

Overall, these results highlight that, for interpretability-driven time series modeling, high-quality sparse solutions are critical: exact sparse solvers not only remain computationally efficient but also provide lag selections that more cleanly expose the underlying periodic structure.

## 5. Discussion

We studied exact sparse regression for time series via the $\ell_0$-constrained least squares formulation. Under stationary $\alpha$-mixing Gaussian assumptions, we established high-

probability RSC/RSS conditions for the empirical loss and further derived finite-sample estimation and geometric convergence guarantees for IHT, CoSaMP, and Subspace Pursuit. We further specialized the theory to sparse VAR models, expressing the dependence and sample-size requirements through interpretable quantities such as the companion-matrix spectrum and innovation covariance. The analysis also extends beyond centered processes, making it applicable to realistic non-mean-zero time series. Empirically, on synthetic sparse Gaussian VAR processes and large-scale urban mobility time series, exact sparse solvers achieve predictive performance comparable to LASSO while providing substantially better sparse lag recovery, lower runtime, and more interpretable supports that align with domain-relevant periodic patterns such as daily and weekly effects. Overall, these results suggest that exact sparse methods offer a statistically principled and computationally efficient alternative to convex sparse estimators for high-dimensional dependent time-series modeling.

A limitation of the current theory is the Gaussianity assumption. Extending the results to broader sub-Gaussian, or more generally sub-Weibull, classes is a promising direction supported by existing work. Such extensions, however, would involve a clear tail–dependence trade-off: relaxing Gaussianity typically requires stronger dependence conditions, for example moving from $\alpha$-mixing to $\beta$-mixing, in order to obtain comparable high-probability quadratic-form bounds.

Several additional directions remain open. First, developing exact sparse guarantees under heavier-tailed innovations, nonlinear dependence structures, or alternative mixing assumptions would further broaden the applicability of the framework. Second, the current experiments emphasize recovery and prediction; combining exact sparse estimation with post-selection inference, confidence intervals, or uncertainty quantification for selected lag effects would be an important step toward fully trustworthy and interpretable time-series modeling.

## Acknowledgements

L. Ding would like to thank Xinyu Chen for introducing the sparse time series regression problem and for pointing us to the ridesharing dataset. We also thank the reviewers for their constructive feedback and helpful suggestions.

## Impact Statement

This paper presents work whose goal is to advance the field of machine learning theory, with an emphasis on providing estimation guarantees for exact sparse methods in time series settings. There are many potential societal consequences of our work, none of which we feel must be specifically highlighted here.

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

## A. Proofs for Section 3.1

To prove Proposition 3.2, we first state a lemma on the quadratic form induced by $\frac{\boldsymbol{X}^\top \boldsymbol{X}}{T}$.

**Lemma A.1.** *Suppose Assumptions 2.2 - 2.4 hold. Define* $\hat{\Gamma} := \frac{\boldsymbol{X}^\top \boldsymbol{X}}{T} \in \mathbb{R}^{p \times p}$. *If* $T \geq \frac{\log(p)}{c \min\{1, \nu^2\}} \max\{120, 16s\}$, *then with probability* $\geq 1 - 2 \exp\left\{-T\frac{c}{2} \min\{1, \nu^2\}\right\}$, *we have the following holds for* all $\boldsymbol{\Delta} \in \mathbb{R}^p$ *with* $\|\boldsymbol{\Delta}\|_2 = 1$,

$$\frac{1}{2}\lambda_{\min}(\Sigma_{\boldsymbol{X}})\left(\|\boldsymbol{\Delta}\|_2^2 - \frac{\|\boldsymbol{\Delta}\|_1^2}{k}\right) \leq \boldsymbol{\Delta}^\top \hat{\Gamma} \boldsymbol{\Delta} \leq \lambda_{\max}(\Sigma_{\boldsymbol{X}})\left(\|\boldsymbol{\Delta}\|_2^2 + \frac{\|\boldsymbol{\Delta}\|_2^2}{2} + \frac{\|\boldsymbol{\Delta}\|_1^2}{2k}\right),$$

*where* $\nu = \frac{\lambda_{\min}(\Sigma_{\boldsymbol{X}})}{108\pi S_\alpha(T)\lambda_{\max}(\Sigma_{\boldsymbol{X}})}$, $k = \left\lceil c\frac{T}{4\log(p)} \min\{1, \nu^2\}\right\rceil$, *and* $\Sigma_{\boldsymbol{X}} = \mathrm{Cov}(\boldsymbol{X}_t)$.

With the above lemma, we are ready to prove Proposition 3.2.

*Proof of Proposition 3.2.* For $f(\boldsymbol{\theta}) = \frac{1}{2T}\left\|\tilde{\boldsymbol{y}} - \tilde{\boldsymbol{X}}\boldsymbol{\theta}\right\|_2^2$, where $\tilde{\boldsymbol{X}} := \boldsymbol{I}_q \otimes \boldsymbol{X} \in \mathbb{R}^{Tq \times pq}$, let $\tilde{\Gamma} = \frac{1}{T}\tilde{\boldsymbol{X}}^\top \tilde{\boldsymbol{X}}$.

By Kronecker identities,

$$\tilde{\Gamma} := \frac{1}{T}\tilde{\boldsymbol{X}}^\top \tilde{\boldsymbol{X}} = \frac{1}{T}(\boldsymbol{I}_q \otimes \boldsymbol{X})^\top(\boldsymbol{I}_q \otimes \boldsymbol{X}) = \boldsymbol{I}_q \otimes \left(\frac{1}{T}\boldsymbol{X}^\top \boldsymbol{X}\right) = \boldsymbol{I}_q \otimes \hat{\Gamma},$$

where $\hat{\Gamma} = \frac{\boldsymbol{X}^\top \boldsymbol{X}}{T}$.

Thanks to the fact that $f$ is quadratic, we have that for any $\boldsymbol{\theta}, \boldsymbol{\delta} \in \mathbb{R}^{pq}$,

$$f(\boldsymbol{\theta} + \boldsymbol{\delta}) - f(\boldsymbol{\theta}) - \langle\nabla f(\boldsymbol{\theta}), \boldsymbol{\delta}\rangle = \frac{1}{2}\boldsymbol{\delta}^\top \tilde{\Gamma} \boldsymbol{\delta}.$$

Write $\boldsymbol{\delta} = \mathrm{vec}(\boldsymbol{\Delta})$ with $\boldsymbol{\Delta} \in \mathbb{R}^{p \times q}$ and columns $\boldsymbol{\Delta}_{:,j} \in \mathbb{R}^p$. Then

$$\boldsymbol{\delta}^\top \tilde{\Gamma} \boldsymbol{\delta} = \sum_{j=1}^q \boldsymbol{\Delta}_{:,j}^\top \hat{\Gamma} \boldsymbol{\Delta}_{:,j}.$$

Assume that $T \geq \frac{\log(p)}{c \min\{1, \nu^2\}} \max\{120, 16s\}$. Applying Lemma A.1 to each column and summing over $j$ gives

$$\boldsymbol{\delta}^\top \tilde{\Gamma} \boldsymbol{\delta} \geq \frac{1}{2}\lambda_{\min}(\Sigma_{\boldsymbol{X}}) \sum_{j=1}^q \|\boldsymbol{\Delta}_{:,j}\|_2^2 - \frac{\lambda_{\min}(\Sigma_{\boldsymbol{X}})}{2k} \sum_{j=1}^q \|\boldsymbol{\Delta}_{:,j}\|_1^2$$

$$\geq \frac{1}{2}\lambda_{\min}(\Sigma_{\boldsymbol{X}})\|\boldsymbol{\delta}\|_2^2 - \frac{\lambda_{\min}(\Sigma_{\boldsymbol{X}})}{2k}\|\boldsymbol{\delta}\|_1^2,$$

$$\boldsymbol{\delta}^\top \tilde{\Gamma} \boldsymbol{\delta} \leq \left(\lambda_{\max}(\Sigma_{\boldsymbol{X}}) + \frac{1}{2}\lambda_{\max}(\Sigma_{\boldsymbol{X}})\right) \sum_{j=1}^q \|\boldsymbol{\Delta}_{:,j}\|_2^2 + \frac{\lambda_{\max}(\Sigma_{\boldsymbol{X}})}{2k} \sum_{j=1}^q \|\boldsymbol{\Delta}_{:,j}\|_1^2$$

$$\leq \left(\lambda_{\max}(\Sigma_{\boldsymbol{X}}) + \frac{1}{2}\lambda_{\max}(\Sigma_{\boldsymbol{X}})\right)\|\boldsymbol{\delta}\|_2^2 + \frac{\lambda_{\max}(\Sigma_{\boldsymbol{X}})}{2k}\|\boldsymbol{\delta}\|_1^2,$$

with probability at least $1 - 2\exp\left\{-\frac{c}{2} T \min\{1, \nu^2\}\right\}$.

Finally, restrict to $s$-sparse directions, i.e., $\|\boldsymbol{\delta}\|_0 \leq s$. Then $\|\boldsymbol{\delta}\|_1^2 \leq s\|\boldsymbol{\delta}\|_2^2$, and we obtain

$$\frac{1}{2}\lambda_{\min}(\Sigma_{\boldsymbol{X}})\left(1 - \frac{s}{k}\right)\|\boldsymbol{\delta}\|_2^2 \leq \boldsymbol{\delta}^\top \tilde{\Gamma} \boldsymbol{\delta} \leq \lambda_{\max}(\Sigma_{\boldsymbol{X}})\left(1 + \frac{1}{2} + \frac{s}{2k}\right)\|\boldsymbol{\delta}\|_2^2.$$

In particular, to ensure a strictly positive curvature constant $m_s > 0$ in the RSC condition, we assume that the sparsity level $s$ satisfies $4s \leq k$. Consequently, it suffices to require the sample size to satisfy $T \geq \frac{\log(p)}{c \min\{1, \nu^2\}} \max\{120, 16s\}$. Then,

$$\frac{1}{2\|\boldsymbol{\delta}\|_2^2}\boldsymbol{\delta}^\top \tilde{\Gamma} \boldsymbol{\delta} \geq \frac{3}{16}\lambda_{\min}(\Sigma_{\boldsymbol{X}}) := m_s, \quad \text{and} \quad \frac{1}{2\|\boldsymbol{\delta}\|_2^2}\boldsymbol{\delta}^\top \tilde{\Gamma} \boldsymbol{\delta} \leq \frac{13}{16}\lambda_{\max}(\Sigma_{\boldsymbol{X}}) := M_s.$$

Substituting these back into the Taylor expansion of $f$ yields, for all $\boldsymbol{\delta} \in \mathbb{R}^{pq}$ with $\|\boldsymbol{\delta}\|_0 \le s$,

$$m_s \|\boldsymbol{\delta}\|_2^2 \;\le\; f(\boldsymbol{\theta} + \boldsymbol{\delta}) - f(\boldsymbol{\theta}) - \langle \nabla f(\boldsymbol{\theta}), \boldsymbol{\delta} \rangle \;\le\; M_s \|\boldsymbol{\delta}\|_2^2,$$

which is the RSC/RSS property with parameters $m_s$ and $M_s$. $\qquad\square$

*Proof of Lemma A.1.* Recall the design matrix $\boldsymbol{X} \in \mathbb{R}^{T \times p}$. For a fixed unit test vector $\boldsymbol{\Delta} \in \mathbb{R}^p$, $\|\boldsymbol{\Delta}\|_2 = 1$, consider the Gaussian vector $\boldsymbol{X}\boldsymbol{\Delta} \in \mathbb{R}^T$. The covariance matrix $\boldsymbol{Q}$ of $\boldsymbol{X}\boldsymbol{\Delta}$ is

$$\boldsymbol{Q} = \begin{bmatrix} \boldsymbol{\Delta}^\top \mathbb{E}[\boldsymbol{X}_1 \boldsymbol{X}_1^\top] \boldsymbol{\Delta} & \cdots & \boldsymbol{\Delta}^\top \mathbb{E}[\boldsymbol{X}_1 \boldsymbol{X}_j^\top] \boldsymbol{\Delta} & \cdots & \boldsymbol{\Delta}^\top \mathbb{E}[\boldsymbol{X}_1 \boldsymbol{X}_T^\top] \boldsymbol{\Delta} \\ \vdots & \ddots & & & \vdots \\ \boldsymbol{\Delta}^\top \mathbb{E}[\boldsymbol{X}_t \boldsymbol{X}_1^\top] \boldsymbol{\Delta} & & \boldsymbol{\Delta}^\top \mathbb{E}[\boldsymbol{X}_t \boldsymbol{X}_t^\top] \boldsymbol{\Delta} & & \boldsymbol{\Delta}^\top \mathbb{E}[\boldsymbol{X}_t \boldsymbol{X}_T^\top] \boldsymbol{\Delta} \\ \vdots & & & \ddots & \vdots \\ \boldsymbol{\Delta}^\top \mathbb{E}[\boldsymbol{X}_T \boldsymbol{X}_1^\top] \boldsymbol{\Delta} & \cdots & \boldsymbol{\Delta}^\top \mathbb{E}[\boldsymbol{X}_T \boldsymbol{X}_t^\top] \boldsymbol{\Delta} & \cdots & \boldsymbol{\Delta}^\top \mathbb{E}[\boldsymbol{X}_T \boldsymbol{X}_T^\top] \boldsymbol{\Delta} \end{bmatrix}.$$

Using Fact A.2 and Lemma A.3, we can bound the operator norm:

$$\|\boldsymbol{Q}\| \le \sum_{t=0}^{T} \rho(t) \|\Sigma_{\boldsymbol{X}}(0)\|, \tag{7}$$

where $\rho(t)$ denotes the $\rho$-mixing coefficient defined in Definition E.2, and $\Sigma_{\boldsymbol{X}}(\ell)$ denotes the lag-$\ell$ autocovariance matrix of $\{\boldsymbol{X}_t\}$, i.e., $\Sigma_{\boldsymbol{X}}(\ell) := \mathrm{Cov}(\boldsymbol{X}_t, \boldsymbol{X}_{t+\ell})$.

Define $\hat{\Gamma} := \frac{\boldsymbol{X}^\top \boldsymbol{X}}{T} \in \mathbb{R}^{p \times p}$. Applying Lemma A.4 to any fixed unit vector $\boldsymbol{\Delta} \in \mathbb{R}^p$, we have, for any $\nu' > 0$,

$$\mathbb{P}\left[\left|\boldsymbol{\Delta}^\top(\hat{\Gamma} - \Sigma_{\boldsymbol{X}}(0))\boldsymbol{\Delta}\right| > \nu' \|\boldsymbol{Q}\|\right] = \mathbb{P}\left[\frac{1}{T}\left|\|\boldsymbol{X}\boldsymbol{\Delta}\|_2^2 - \mathbb{E}\|\boldsymbol{X}\boldsymbol{\Delta}\|_2^2\right| > \nu' \|\boldsymbol{Q}\|\right] \le 2\exp\{-cT\min(\nu', \nu'^2)\},$$

$$\implies \mathbb{P}\left[\left|\boldsymbol{\Delta}^\top(\hat{\Gamma} - \Sigma_{\boldsymbol{X}}(0))\boldsymbol{\Delta}\right| > \nu' \sum_{t=0}^{T} \rho(t) \|\Sigma_{\boldsymbol{X}}(0)\|\right] \le 2\exp\{-cT\min(\nu', \nu'^2)\}. \tag{8}$$

Using Lemma A.5, for any integer $k > 0$, we restrict to sparse vectors in $\mathcal{J}(2k) := \{\boldsymbol{\Delta} \in \mathbb{R}^p : \|\boldsymbol{\Delta}\| \le 1, \|\boldsymbol{\Delta}\|_0 \le 2k\}$:

$$\mathbb{P}\left[\sup_{\boldsymbol{\Delta} \in \mathcal{J}(2k)} \left|\boldsymbol{\Delta}^\top(\hat{\Gamma} - \Sigma_{\boldsymbol{X}}(0))\boldsymbol{\Delta}\right| > \nu' \sum_{t=0}^{T} \rho(t) \|\Sigma_{\boldsymbol{X}}(0)\|\right]$$
$$\le 2\exp\left\{-cT\min(\nu', \nu'^2) + 2k\min\left\{\log(p), \log\left(\frac{21ep}{2k}\right)\right\}\right\} \tag{9}$$

By Lemma A.6, we further extend the bound to all vectors $\boldsymbol{\Delta} \in \mathbb{R}^p$:

$$\mathbb{P}\left\{\left|\boldsymbol{\Delta}^\top(\hat{\Gamma} - \Sigma_{\boldsymbol{X}}(0))\boldsymbol{\Delta}\right| > 27\nu' \sum_{t=0}^{T} \rho(t) \|\Sigma_{\boldsymbol{X}}(0)\| \left(\|\boldsymbol{\Delta}\|_2^2 + \frac{1}{k}\|\boldsymbol{\Delta}\|_1^2\right)\right\}$$
$$\le 2\exp\left\{-cT\min(\nu', \nu'^2) + 2k\min\left(\log(p), \log\left(\frac{21ep}{2k}\right)\right)\right\} \tag{10}$$

Equivalently,

$$\mathbb{P}\left\{\left|\boldsymbol{\Delta}^\top(\hat{\Gamma} - \Sigma_{\boldsymbol{X}}(0))\boldsymbol{\Delta}\right| \le 27\nu' \sum_{t=0}^{T} \rho(t) \|\Sigma_{\boldsymbol{X}}(0)\| \left(\|\boldsymbol{\Delta}\|_2^2 + \frac{1}{k}\|\boldsymbol{\Delta}\|_1^2\right)\right\}$$
$$> 1 - 2\exp\left\{-cT\min(\nu', \nu'^2) + 2k\min\left(\log(p), \log\left(\frac{21ep}{2k}\right)\right)\right\} \tag{11}$$

Select $\nu' = \frac{\lambda_{\min}(\Sigma_{\boldsymbol{X}}(0))}{54 \sum_{t=0}^{T} \rho(t) \lambda_{\max}(\Sigma_{\boldsymbol{X}}(0))}$. We have

$$\boldsymbol{\Delta}^{\top} \hat{\Gamma} \boldsymbol{\Delta} \geq \frac{1}{2} \lambda_{\min}(\Sigma_{\boldsymbol{X}}(0)) \|\boldsymbol{\Delta}\|_2^2 - \frac{\lambda_{\min}(\Sigma_{\boldsymbol{X}}(0))}{2k} \|\boldsymbol{\Delta}\|_1^2, \tag{12}$$

$$\text{and} \quad \boldsymbol{\Delta}^{\top} \hat{\Gamma} \boldsymbol{\Delta} \leq \lambda_{\max}(\Sigma_{\boldsymbol{X}}(0)) \|\boldsymbol{\Delta}\|_2^2 + \frac{1}{2} \lambda_{\min}(\Sigma_{\boldsymbol{X}}(0)) \|\boldsymbol{\Delta}\|_2^2 + \frac{\lambda_{\min}(\Sigma_{\boldsymbol{X}}(0))}{2k} \|\boldsymbol{\Delta}\|_1^2, \tag{13}$$

with probability $\geq 1 - 2 \exp \left\{ -cT \min(1, \nu'^2) + 2k \min \left( \log(p), \log \left( \frac{21ep}{2k} \right) \right) \right\}$, since $\min(1, \nu'^2) \leq \min(\nu', \nu'^2)$. To make sure the first component in the exponential dominates, we assume $p \leq \frac{21ep}{2k}$, and choose $k = \left\lceil c \frac{T}{4 \log(p)} \min\{1, \nu'^2\} \right\rceil$. Then for $T \geq \frac{42e \log(p)}{c \min\{1, \nu'^2\}}$, with probability at least $1 - 2 \exp \left\{ -T \frac{c}{2} \min\{1, \nu'^2\} \right\}$, equation (12) and (13) hold.

Define $\nu := \frac{\lambda_{\min}(\Sigma_{\boldsymbol{X}})}{108\pi S_\alpha(T) \lambda_{\max}(\Sigma_{\boldsymbol{X}})}$. By Fact E.3, we know $\nu \leq \nu'$ and we have that for $T \geq \frac{42e \log(p)}{c \min\{1, \nu^2\}}$, with probability at least $1 - 2 \exp \left\{ -T \frac{c}{2} \min\{1, \nu^2\} \right\}$, equation (12) and (13) hold.

Since $\lambda_{\min}(\Sigma_{\boldsymbol{X}}(0)) \leq \lambda_{\max}(\Sigma_{\boldsymbol{X}}(0))$, the above equation (13) further implies

$$\boldsymbol{\Delta}^{\top} \hat{\Gamma} \boldsymbol{\Delta} \leq \lambda_{\max}(\Sigma_{\boldsymbol{X}}(0)) \left( \|\boldsymbol{\Delta}\|_2^2 + \frac{\|\boldsymbol{\Delta}\|_2^2}{2} + \frac{\|\boldsymbol{\Delta}\|_1^2}{2k} \right).$$

$\square$

*Fact* A.2 (Schur Test). For any matrix $\boldsymbol{M} \in \mathbb{R}^{m \times n}$, we have

$$\|\boldsymbol{M}\|^2 \leq \left( \max_{1 \leq i \leq m} \|\boldsymbol{M}_{i,:}\|_1 \right) \cdot \left( \max_{1 \leq j \leq n} \|\boldsymbol{M}_{:,j}\|_1 \right).$$

Therefore, for any symmetric matrix $\boldsymbol{M} \in \mathbb{R}^{n \times n}$,

$$\|\boldsymbol{M}\| \leq \max_{1 \leq i \leq n} \|\boldsymbol{M}_{i,:}\|_1.$$

**Lemma A.3** (Lemma 10 from (Wong et al., 2020)). *For a second-order stationary $\rho$-mixing sequence of random vectors $\{\boldsymbol{X}_t\}$, the $l$-th autocovariance matrix can be bounded as follows:*

$$\|\Sigma_{\boldsymbol{X}}(l)\| \leq \rho(l) \|\Sigma_{\boldsymbol{X}}(0)\|, \quad \forall l \in \mathbb{Z}.$$

**Lemma A.4** (Lemma 11 from (Wong et al., 2020)). *If $\boldsymbol{Y} \sim \mathcal{N}(\boldsymbol{0}_{n \times 1}, \boldsymbol{Q}_{n \times n})$, then there exists a universal constant $c > 0$ such that for any $\nu > 0$,*

$$\mathbb{P} \left[ \frac{1}{n} \left| \|\boldsymbol{Y}\|_2^2 - \mathbb{E} \|\boldsymbol{Y}\|_2^2 \right| > \nu \|\boldsymbol{Q}\| \right] \leq 2 \exp \left[ -cn \min\{\nu, \nu^2\} \right] \tag{14}$$

**Lemma A.5** (Lemma F.2 from (Basu & Michailidis, 2015)). *Consider a symmetric matrix $\boldsymbol{M} \in \mathbb{R}^{p \times p}$. If, for any vector $\boldsymbol{v} \in \mathbb{R}^p$ with $\|\boldsymbol{v}\| \leq 1$, and any $\nu > 0$,*

$$\mathbb{P} \left[ \left| \boldsymbol{v}^{\top} \boldsymbol{M} \boldsymbol{v} \right| > C\nu \right] \leq 2 \exp \left[ -cT \min(\nu, \nu^2) \right],$$

*then, for any integer $k \geq 1$, we have*

$$\mathbb{P} \left[ \sup_{\boldsymbol{v} \in \mathcal{J}(k)} \left| \boldsymbol{v}^{\top} \boldsymbol{M} \boldsymbol{v} \right| > C\nu \right] \leq 2 \exp \left[ -cT \min(\nu, \nu^2) + k \min \left\{ \log(p), \log \left( \frac{21ep}{k} \right) \right\} \right],$$

*where $\mathcal{J}(k) := \{\boldsymbol{v} \in \mathbb{R}^p : \|\boldsymbol{v}\| \leq 1, \|\boldsymbol{v}\|_0 \leq k\}$.*

**Lemma A.6** (Lemma 12 from (Loh & Wainwright, 2012)). *For a fixed matrix $\boldsymbol{M} \in \mathbb{R}^{p \times p}$, integer $k \geq 1$, and tolerance $C\nu > 0$, suppose we have $\left| \boldsymbol{v}^{\top} \boldsymbol{M} \boldsymbol{v} \right| \leq C\nu, \forall \boldsymbol{v} \in \mathcal{J}(2k)$. Then,*

$$\left| \boldsymbol{v}^{\top} \boldsymbol{M} \boldsymbol{v} \right| \leq 27C\nu \left( \|\boldsymbol{v}\|_2^2 + \frac{1}{k} \|\boldsymbol{v}\|_1^2 \right), \quad \forall \boldsymbol{v} \in \mathbb{R}^p.$$

## B. Analysis for the uncentered case

**Proposition B.1** (RSC/RSS for Gaussian processes with nonzero means)**.** *Suppose Assumptions 2.2, 2.3, and 2.4 hold. Let $X_t \in \mathbb{R}^p, Y_t \in \mathbb{R}^q$ be as in Proposition 3.2, but without the centering Assumption 2.5. Let*

$$\bar{X} := \frac{1}{T} \sum_{t=1}^{T} X_t, \qquad \bar{Y} := \frac{1}{T} \sum_{t=1}^{T} Y_t,$$

*and define the empirically centered variables*

$$X_t^{emp} := X_t - \bar{X}, \qquad Y_t^{emp} := Y_t - \bar{Y}.$$

*Let $X^{emp} \in \mathbb{R}^{T \times p}$ and $Y^{emp} \in \mathbb{R}^{T \times q}$ be the matrices whose rows are ${X_t^{emp}}^\top$ and ${Y_t^{emp}}^\top$, respectively, and set $\tilde{y}^{emp} := \operatorname{vec}(Y^{emp}) \in \mathbb{R}^{Tq}$, $\tilde{X}^{emp} := I_q \otimes X^{emp} \in \mathbb{R}^{Tq \times pq}$. Consider the least squares objective with an intercept $a \in \mathbb{R}^q$:*

$$\tilde{f}(\theta) := \min_{a \in \mathbb{R}^q} \frac{1}{2T} \sum_{t=1}^{T} \left\| Y_t - a - X_t^\top \Theta \right\|_2^2 = \frac{1}{2T} \left\| \tilde{y}^{emp} - \tilde{X}^{emp} \theta \right\|_2^2,$$

*where $\Theta \in \mathbb{R}^{p \times q}$ and $\theta := \operatorname{vec}(\Theta) \in \mathbb{R}^{pq}$.*

*Let constant $c$, sample size $T$, $\nu, m_s, M_s$ be as in Proposition 3.2. Then there exist universal constants $C > 0$ such that, on an event of probability at least $1 - 2 \exp\left( -\frac{c}{2} T \min\{1, \nu^2\} \right) - (2p)^{1-C}$, the empirical loss $\tilde{f}$ satisfies RSC/RSS with parameters*

$$\tilde{m}_s = m_s - C \lambda_{\max}(\Sigma_X)\left(1 + 4\pi S_\alpha(T)\right) \frac{s \log(2p)}{T}, \qquad \tilde{M}_s = M_s,$$

*in the sense that for all $\theta \in \mathbb{R}^{pq}$ and all $\delta \in \mathbb{R}^{pq}$ with $\|\delta\|_0 \le s$,*

$$\tilde{m}_s \|\delta\|_2^2 \le \tilde{f}(\theta + \delta) - \tilde{f}(\theta) - \langle \nabla \tilde{f}(\theta), \delta \rangle \le \tilde{M}_s \|\delta\|_2^2.$$

*In particular, if $T \ge \frac{2C}{m_s} \lambda_{\max}(\Sigma_X)\left(1 + 4\pi S_\alpha(T)\right) s \log(2p)$, then $\tilde{m}_s > 0$ and hence $\tilde{f}$ satisfies RSC/RSS with strictly positive parameters.*

*Proof of Proposition B.1.* Since $\tilde{f}(\theta) = \frac{1}{2T} \left\| \tilde{y}^{emp} - \tilde{X}^{emp} \theta \right\|_2^2$, Therefore, for any $\theta, \delta \in \mathbb{R}^{pq}$ we have

$$\tilde{f}(\theta + \delta) - \tilde{f}(\theta) - \langle \nabla \tilde{f}(\theta), \delta \rangle = \frac{1}{2} \delta^\top (I_q \otimes \Gamma^{\mathrm{emp}}) \delta,$$

where $\Gamma^{\mathrm{emp}} = \frac{1}{T} {X^{\mathrm{emp}}}^\top X^{\mathrm{emp}}$. Consequently, to establish the RSC/RSS property for $\tilde{f}$ it is sufficient to show that, on a high-probability event,

$$\tilde{m}_s \|\delta\|_2^2 \le \frac{1}{2} \delta^\top (I_q \otimes \Gamma^{\mathrm{emp}}) \delta \le \tilde{M}_s \|\delta\|_2^2, \qquad \forall \delta \in \mathbb{R}^{pq} \text{ with } \|\delta\|_0 \le s, \tag{15}$$

with $\tilde{m}_s, \tilde{M}_s$ as stated in the proposition.

*Step 1: RSC/RSS for the centered case.* Let $X^\circ \in \mathbb{R}^{T \times p}$ be the matrix with rows $Z_t^\top$, where $Z_t := X_t - \mu_X$, $\mu_X = \mathbb{E}[X_t]$, and set $\Gamma^\circ := \frac{1}{T} \sum_{t=1}^{T} Z_t Z_t^\top = \frac{1}{T} (X^\circ)^\top X^\circ$. In the centered setting, the design matrix is $\tilde{X}^\circ = I_q \otimes X^\circ$ and the objective function in Equation (3) has Hessian $I_q \otimes \Gamma^\circ$.

By Proposition 3.2, there exists an event

$$\mathcal{E}_{\mathrm{cen}} := \left\{ \forall \delta \in \mathbb{R}^{pq}, \|\delta\|_0 \le s : m_s \|\delta\|_2^2 \le \frac{1}{2} \delta^\top (I_q \otimes \Gamma^\circ) \delta \le M_s \|\delta\|_2^2 \right\}$$

such that

$$\mathbb{P}(\mathcal{E}_{\mathrm{cen}}) \ge 1 - 2 \exp\left( -\frac{c}{2} T \min\{1, \nu^2\} \right).$$

In particular, for any $\boldsymbol{v} \in \mathbb{R}^p$ with $\|\boldsymbol{v}\|_0 \leq s$, take $\boldsymbol{\delta} = \boldsymbol{e}_k \otimes \boldsymbol{v}$ for any $k \in \{1, \ldots, q\}$, where $\boldsymbol{e}_k$ is the $k$-th standard basis vector in $\mathbb{R}^q$. Then

$$\boldsymbol{\delta}^\top (\boldsymbol{I}_q \otimes \Gamma^\circ)\boldsymbol{\delta} = \boldsymbol{v}^\top \Gamma^\circ \boldsymbol{v}, \qquad \|\boldsymbol{\delta}\|_2^2 = \|\boldsymbol{v}\|_2^2,$$

so that on $\mathcal{E}_{\mathrm{cen}}$,

$$m_s \|\boldsymbol{v}\|_2^2 \ \leq \ \frac{1}{2} \boldsymbol{v}^\top \Gamma^\circ \boldsymbol{v} \ \leq \ M_s \|\boldsymbol{v}\|_2^2, \qquad \forall\, \boldsymbol{v} \in \mathbb{R}^p \text{ with } \|\boldsymbol{v}\|_0 \leq s. \tag{16}$$

*Step 2: Transferring RSC/RSS from $\Gamma^\circ$ to $\Gamma^{emp}$.* Define the sample mean error $\boldsymbol{\Delta} := \bar{\boldsymbol{X}} - \mu_{\boldsymbol{X}} = \frac{1}{T} \sum_{t=1}^T \boldsymbol{Z}_t$. Since $\boldsymbol{X}_t^{\mathrm{emp}} = \boldsymbol{X}_t - \bar{\boldsymbol{X}} = \boldsymbol{Z}_t - \boldsymbol{\Delta}$, we obtain

$$\Gamma^{\mathrm{emp}} = \frac{1}{T} \sum_{t=1}^T (\boldsymbol{Z}_t - \boldsymbol{\Delta})(\boldsymbol{Z}_t - \boldsymbol{\Delta})^\top = \Gamma^\circ - \boldsymbol{\Delta}\boldsymbol{\Delta}^\top.$$

Consequently, for any $\boldsymbol{v} \in \mathbb{R}^p$,

$$\boldsymbol{v}^\top \Gamma^{\mathrm{emp}} \boldsymbol{v} = \boldsymbol{v}^\top \Gamma^\circ \boldsymbol{v} - (\boldsymbol{v}^\top \boldsymbol{\Delta})^2.$$

Let $\boldsymbol{v} \in \mathbb{R}^p$ with $\|\boldsymbol{v}\|_0 \leq s$. By Hölder's inequality and sparsity,

$$|\boldsymbol{v}^\top \boldsymbol{\Delta}| \leq \|\boldsymbol{v}\|_1 \|\boldsymbol{\Delta}\|_\infty \leq \sqrt{s}\, \|\boldsymbol{v}\|_2 \|\boldsymbol{\Delta}\|_\infty,$$

and hence

$$(\boldsymbol{v}^\top \boldsymbol{\Delta})^2 \leq s \|\boldsymbol{\Delta}\|_\infty^2 \|\boldsymbol{v}\|_2^2.$$

Combining this with Equation (16), on the event $\mathcal{E}_{\mathrm{cen}}$ we have

$$\Big(m_s - \frac{1}{2} s \|\boldsymbol{\Delta}\|_\infty^2\Big) \|\boldsymbol{v}\|_2^2 \ \leq \ \frac{1}{2} \boldsymbol{v}^\top \Gamma^{\mathrm{emp}} \boldsymbol{v} \ \leq \ M_s \|\boldsymbol{v}\|_2^2, \qquad \forall\, \boldsymbol{v} \in \mathbb{R}^p \text{ with } \|\boldsymbol{v}\|_0 \leq s. \tag{17}$$

*Step 3: From $\Gamma^{emp}$ to $\boldsymbol{I}_q \otimes \Gamma^{emp}$.* Partition any $\boldsymbol{\delta} \in \mathbb{R}^{pq}$ as $\boldsymbol{\delta} = (\boldsymbol{\delta}_1^\top, \ldots, \boldsymbol{\delta}_q^\top)^\top$, $\boldsymbol{\delta}_k \in \mathbb{R}^p$. If $\|\boldsymbol{\delta}\|_0 \leq s$, then each block $\boldsymbol{\delta}_k$ also satisfies $\|\boldsymbol{\delta}_k\|_0 \leq s$. Using the Kronecker structure,

$$\boldsymbol{\delta}^\top (\boldsymbol{I}_q \otimes \Gamma^{\mathrm{emp}})\boldsymbol{\delta} = \sum_{k=1}^q \boldsymbol{\delta}_k^\top \Gamma^{\mathrm{emp}} \boldsymbol{\delta}_k.$$

Applying Equation (17) to each $\boldsymbol{\delta}_k$ and summing, we obtain that on the event $\mathcal{E}_{\mathrm{cen}} \cap \{\|\boldsymbol{\Delta}\|_\infty \leq r_T\}$,

$$\Big(m_s - \frac{1}{2} s r_T^2\Big) \|\boldsymbol{\delta}\|_2^2 \ \leq \ \frac{1}{2} \boldsymbol{\delta}^\top (\boldsymbol{I}_q \otimes \Gamma^{\mathrm{emp}})\boldsymbol{\delta} \ \leq \ M_s \|\boldsymbol{\delta}\|_2^2,$$

for all $\boldsymbol{\delta} \in \mathbb{R}^{pq}$ with $\|\boldsymbol{\delta}\|_0 \leq s$. Thus, on this event, Equation (15) holds with

$$\tilde{m}_s := m_s - \frac{1}{2} s r_T^2, \qquad \tilde{M}_s := M_s.$$

*Step 4: High-probability bound for $r_T$, i.e. a bound on $\|\boldsymbol{\Delta}\|_\infty$.* Fix a coordinate $j \in \{1, \ldots, p\}$ and consider the scalar process $Z_{t,j} := (\boldsymbol{Z}_t)_j$. Under our assumptions, $(Z_{t,j})_{t \geq 1}$ is a mean-zero, stationary Gaussian process that is $\alpha$-mixing. Its sample mean is

$$\bar{Z}_j := \frac{1}{T} \sum_{t=1}^T Z_{t,j} = \bar{\boldsymbol{X}}_j - \mu_{\boldsymbol{X},j} = \boldsymbol{\Delta}_j.$$

The variance of $\bar{Z}_j$ is

$$\mathrm{Var}(\bar{Z}_j) = \frac{1}{T^2} \sum_{t,s=1}^T \mathrm{Cov}(Z_{t,j}, Z_{s,j}) = \frac{1}{T}\Big(\gamma_j(0) + 2 \sum_{h=1}^{T-1} \Big(1 - \frac{h}{T}\Big)\gamma_j(h)\Big),$$

where $\gamma_j(h) := \text{Cov}(Z_{t,j}, Z_{t+h,j})$. Note that as $\mathbf{Z}_t$ is the shifted $\mathbf{X}_t$, we know $\gamma_j(0) \le \lambda_{\max}(\mathbf{X})$ for any $j$.

Under the $\alpha$-mixing Gaussian assumptions, Definition E.2 and Fact E.3 imply that

$$|\gamma_j(h)| \le 2\pi\, \alpha(h)\, \gamma_j(0), \qquad \text{for all } h \ge 0.$$

Thus

$$\sum_{h=1}^{T-1} |\gamma_j(h)| \;\le\; 2\pi\, S_\alpha(T)\, \gamma_j(0),$$

where $S_\alpha(T)$ is the mixing-sum quantity used in Proposition 3.2. Consequently,

$$\text{Var}(\bar{Z}_j) \le \frac{1}{T}\Big(\gamma_j(0) + 2\sum_{h=1}^{T-1} |\gamma_j(h)|\Big) \le \frac{\gamma_j(0)}{T}\Big(1 + 4\pi\, S_\alpha(T)\Big) \;\le\; \frac{\lambda_{\max}(\Sigma_{\mathbf{X}})}{T}\Big(1 + 4\pi\, S_\alpha(T)\Big).$$

Since $\bar{Z}_j$ is Gaussian, for any $u > 0$,

$$\mathbb{P}\big(|\bar{Z}_j| \ge u\big) \le 2\exp\left(-\frac{u^2}{2\text{Var}(\bar{Z}_j)}\right) \le 2\exp\left(-\frac{u^2 T}{2\,\lambda_{\max}(\Sigma_{\mathbf{X}})\big(1 + 4\pi S_\alpha(T)\big)}\right). \tag{18}$$

Choose

$$u = \sqrt{2C\,\lambda_{\max}(\Sigma_{\mathbf{X}})\big(1 + 4\pi S_\alpha(T)\big)\,\frac{\log(2p)}{T}}$$

with constant $C > 0$ large enough. Then Equation (18) yields

$$\mathbb{P}\left(|\bar{Z}_j| \ge \sqrt{2C\,\lambda_{\max}(\Sigma_{\mathbf{X}})\big(1 + 4\pi S_\alpha(T)\big)\,\frac{\log(2p)}{T}}\right) \le 2(2p)^{-C}.$$

A union bound over $j = 1, \ldots, p$ gives the event

$$\mathcal{E}_{\text{mean}} := \left\{\|\bar{\mathbf{Z}}\|_\infty \le \sqrt{2C\,\lambda_{\max}(\Sigma_{\mathbf{X}})\big(1 + 4\pi S_\alpha(T)\big)\,\frac{\log(2p)}{T}}\right\}$$

satisfying

$$\mathbb{P}(\mathcal{E}_{\text{mean}}) \ge 1 - (2p)^{1-C}.$$

*Step 5: Conclusion.* On the event $\mathcal{E}_{\text{cen}} \cap \mathcal{E}_{\text{mean}}$, we have

$$\mathbb{P}(\mathcal{E}_{\text{cen}} \cap \mathcal{E}_{\text{mean}}) \ge 1 - 2\exp\left(-\tfrac{c}{2}T\min\{1, \nu^2\}\right) - (2p)^{1-C}.$$

We may take

$$r_T := \sqrt{2C\,\lambda_{\max}(\Sigma_{\mathbf{X}})\big(1 + 4\pi S_\alpha(T)\big)\,\frac{\log(2p)}{T}},$$

so that

$$\tilde{m}_s = m_s - C\,\lambda_{\max}(\Sigma_{\mathbf{X}})\big(1 + 4\pi S_\alpha(T)\big)\,\frac{s\log(2p)}{T}, \qquad \tilde{M}_s = M_s,$$

and Equation (15) holds. This is the desired RSC/RSS property for $\tilde{f}$ with parameters $(\tilde{m}_s, \tilde{M}_s)$.

Finally, if

$$T \;\ge\; \frac{2C}{m_s}\,\lambda_{\max}(\Sigma_{\mathbf{X}})\big(1 + 4\pi S_\alpha(T)\big)\,s\log(2p),$$

then $\tilde{m}_s \ge \frac{m_s}{2} > 0$, and hence $\tilde{f}$ satisfies RSC/RSS with strictly positive parameters. This completes the proof. $\square$

## C. Proofs for Section 3.2

*Proof of Theorem 3.3.* Define $\hat{\boldsymbol{\theta}} = \arg\min_{\|\boldsymbol{\theta}\|_0 \leq s^*} f(\boldsymbol{\theta})$. Let $\Delta\boldsymbol{\theta}^k := \boldsymbol{\theta}^{k+1} - \boldsymbol{\theta}^k$, and $\boldsymbol{g}^k = \nabla_{\boldsymbol{\theta}} f(\boldsymbol{\theta}^k)$. For all $k$-th iteration, let $\mathcal{S}^k = \operatorname{supp}(\boldsymbol{\theta}^k)$, and $\hat{\mathcal{S}} = \operatorname{supp}(\hat{\boldsymbol{\theta}})$. For any index set $\mathcal{I}$, $\boldsymbol{g}_{\mathcal{I}}^k$ denotes the subvector of $\boldsymbol{g}^k$ restricted to coordinates in $\mathcal{I}$.

- **(IHT).** The IHT algorithm is $\boldsymbol{\theta}^{k+1} = P_s(\boldsymbol{\theta}^k - \eta\boldsymbol{g}^k)$. Let $M = 2M_{2s+s^*}$ and $m = 2m_{2s+s^*}$.

  Let $\mathcal{I}^k = \hat{\mathcal{S}} \cup \mathcal{S}^k \cup \mathcal{S}^{k+1}$. Because of the fact that $\operatorname{supp}(\boldsymbol{\theta}^k) \subseteq \mathcal{I}^k$, $\operatorname{supp}(\boldsymbol{\theta}^{k+1}) \subseteq \mathcal{I}^k$ and the choice $\eta' := \eta M = \frac{2}{3}$, by adding and subtracting the terms $\frac{(\eta')^2}{2M}\|\boldsymbol{g}_{\mathcal{I}^k}^k\|_2^2 + \eta'\langle\Delta\boldsymbol{\theta}^k, \boldsymbol{g}^k\rangle$ and completing the square, the RSS property can be rewritten as

$$f(\boldsymbol{\theta}^{k+1}) - f(\boldsymbol{\theta}^k) \leq \frac{M}{2}\left\|\Delta\boldsymbol{\theta}_{\mathcal{I}^k}^k + \frac{\eta'}{M}\boldsymbol{g}_{\mathcal{I}^k}^k\right\|_2^2 - \frac{(\eta')^2}{2M}\|\boldsymbol{g}_{\mathcal{I}^k}^k\|_2^2 + (1-\eta')\langle\Delta\boldsymbol{\theta}^k, \boldsymbol{g}^k\rangle. \tag{19}$$

As $\mathcal{S}^k \setminus \mathcal{S}^{k+1}$ and $\mathcal{S}^{k+1}$ are disjoint, we have

$$\langle\boldsymbol{\theta}^{k+1} - \boldsymbol{\theta}^k, \boldsymbol{g}^k\rangle = -\langle\boldsymbol{\theta}_{\mathcal{S}^k\setminus\mathcal{S}^{k+1}}^k, \boldsymbol{g}_{\mathcal{S}^k\setminus\mathcal{S}^{k+1}}^k\rangle + \langle\boldsymbol{\theta}_{\mathcal{S}^{k+1}}^{k+1} - \boldsymbol{\theta}_{\mathcal{S}^{k+1}}^k, \boldsymbol{g}_{\mathcal{S}^{k+1}}^k\rangle.$$

Using the gradient step $\boldsymbol{\theta}_{\mathcal{S}^{k+1}}^{k+1} = \boldsymbol{\theta}_{\mathcal{S}^{k+1}}^k - \frac{\eta'}{M}\boldsymbol{g}_{\mathcal{S}^{k+1}}^k$, we get

$$\langle\Delta\boldsymbol{\theta}^k, \boldsymbol{g}^k\rangle = -\langle\boldsymbol{\theta}_{\mathcal{S}^k\setminus\mathcal{S}^{k+1}}^k, \boldsymbol{g}_{\mathcal{S}^k\setminus\mathcal{S}^{k+1}}^k\rangle - \frac{\eta'}{M}\|\boldsymbol{g}_{\mathcal{S}^{k+1}}^k\|_2^2.$$

Since $\boldsymbol{\theta}^{k+1}$ is obtained by hard-thresholding $\left(\boldsymbol{\theta}^k - \frac{\eta'}{M}\boldsymbol{g}^k\right)$, we have

$$\left\|\boldsymbol{\theta}_{\mathcal{S}^k\setminus\mathcal{S}^{k+1}}^k - \frac{\eta'}{M}\boldsymbol{g}_{\mathcal{S}^k\setminus\mathcal{S}^{k+1}}^k\right\|_2^2 \leq \left\|\boldsymbol{\theta}_{\mathcal{S}^{k+1}\setminus\mathcal{S}^k}^{k+1}\right\|_2^2 = \frac{(\eta')^2}{M^2}\left\|\boldsymbol{g}_{\mathcal{S}^{k+1}\setminus\mathcal{S}^k}^k\right\|_2^2.$$

Therefore,

$$\langle\Delta\boldsymbol{\theta}^k, \boldsymbol{g}^k\rangle \leq \frac{\eta'}{2M}\left\|\boldsymbol{g}_{\mathcal{S}^{k+1}\setminus\mathcal{S}^k}^k\right\|_2^2 - \frac{\eta'}{2M}\left\|\boldsymbol{g}_{\mathcal{S}^k\setminus\mathcal{S}^{k+1}}^k\right\|_2^2 - \frac{\eta'}{M}\left\|\boldsymbol{g}_{\mathcal{S}^{k+1}}^k\right\|_2^2.$$

Using this and the fact that $\left\|\boldsymbol{g}_{\mathcal{S}^{k+1}}^k\right\|_2^2 = \left\|\boldsymbol{g}_{\mathcal{S}^{k+1}\cap\mathcal{S}^k}^k\right\|_2^2 + \left\|\boldsymbol{g}_{\mathcal{S}^{k+1}\setminus\mathcal{S}^k}^k\right\|_2^2$, we obtain

$$\langle\Delta\boldsymbol{\theta}^k, \boldsymbol{g}^k\rangle \leq -\frac{\eta'}{2M}\left\|\boldsymbol{g}_{\mathcal{S}^{k+1}\setminus\mathcal{S}^k}^k\right\|_2^2 - \frac{\eta'}{2M}\left\|\boldsymbol{g}_{\mathcal{S}^k\setminus\mathcal{S}^{k+1}}^k\right\|_2^2 - \frac{\eta'}{M}\left\|\boldsymbol{g}_{\mathcal{S}^{k+1}\cap\mathcal{S}^k}^k\right\|_2^2 \leq -\frac{\eta'}{2M}\left\|\boldsymbol{g}_{\mathcal{S}^{k+1}\cup\mathcal{S}^k}^k\right\|_2^2.$$

Combining this with (19) yields

$$\begin{aligned} f(\boldsymbol{\theta}^{k+1}) - f(\boldsymbol{\theta}^k) \leq &\frac{M}{2}\left\|\Delta\boldsymbol{\theta}_{\mathcal{I}^k}^k + \frac{\eta'}{M}\boldsymbol{g}_{\mathcal{I}^k}^k\right\|_2^2 - \frac{(\eta')^2}{2M}\|\boldsymbol{g}_{\mathcal{I}^k}^k\|_2^2 - \frac{\eta'(1-\eta')}{2M}\|\boldsymbol{g}_{\mathcal{S}^{k+1}\cup\mathcal{S}^k}^k\|_2^2, \\ = &\frac{M}{2}\left\|\Delta\boldsymbol{\theta}_{\mathcal{I}^k}^k + \frac{\eta'}{M}\boldsymbol{g}_{\mathcal{I}^k}^k\right\|_2^2 - \frac{(\eta')^2}{2M}\left\|\boldsymbol{g}_{\mathcal{I}^k\setminus(\hat{\mathcal{S}}\cup\mathcal{S}^k)}^k\right\|_2^2 - \frac{(\eta')^2}{2M}\|\boldsymbol{g}_{\hat{\mathcal{S}}\cup\mathcal{S}^k}^k\|_2^2 - \frac{\eta'(1-\eta')}{2M}\|\boldsymbol{g}_{\mathcal{S}^{k+1}\cup\mathcal{S}^k}^k\|_2^2. \end{aligned} \tag{20}$$

Let $(*) := \frac{M}{2}\left\|\Delta\boldsymbol{\theta}_{\mathcal{I}^k}^k + \frac{\eta'}{M}\boldsymbol{g}_{\mathcal{I}^k}^k\right\|_2^2 - \frac{(\eta')^2}{2M}\left\|\boldsymbol{g}_{\mathcal{I}^k\setminus(\hat{\mathcal{S}}\cup\mathcal{S}^k)}^k\right\|_2^2$ denotes the first two terms on the right hand side above. Note that $\mathcal{I}^k \setminus (\mathcal{S}^k \cup \hat{\mathcal{S}}) = \mathcal{S}^{k+1} \setminus (\mathcal{S}^k \cup \hat{\mathcal{S}}) \subseteq \mathcal{S}^{k+1}$, and because $\boldsymbol{\theta}_{\mathcal{I}^k\setminus\mathcal{S}^k}^k = 0$, we have $\boldsymbol{\theta}_{\mathcal{I}^k\setminus(\mathcal{S}^k\cup\hat{\mathcal{S}})}^{k+1} = \boldsymbol{\theta}_{\mathcal{I}^k\setminus(\mathcal{S}^k\cup\hat{\mathcal{S}})}^k - \frac{\eta'}{M}\boldsymbol{g}_{\mathcal{I}^k\setminus(\mathcal{S}^k\cup\hat{\mathcal{S}})}^k = -\frac{\eta'}{M}\boldsymbol{g}_{\mathcal{I}^k\setminus(\mathcal{S}^k\cup\hat{\mathcal{S}})}^k$. Choose a set $\mathcal{D} \subseteq \mathcal{S}^k \setminus \mathcal{S}^{k+1}$ such that $|\mathcal{D}| = |\mathcal{S}^{k+1} \setminus (\mathcal{S}^k \cup \hat{\mathcal{S}})|$, which is possible because $\left|\mathcal{S}^{k+1} \setminus (\mathcal{S}^k \cup \hat{\mathcal{S}})\right| = |\mathcal{S}^k \setminus \mathcal{S}^{k+1}| - |(\mathcal{S}^{k+1} \cap \hat{\mathcal{S}}) \setminus \mathcal{S}^k|$. By the hard-thresholding property,

$$\frac{(\eta')^2}{M^2}\left\|\boldsymbol{g}_{\mathcal{I}^k\setminus(\mathcal{S}^k\cup\hat{\mathcal{S}})}^k\right\|_2^2 = \left\|\boldsymbol{\theta}_{\mathcal{I}^k\setminus(\mathcal{S}^k\cup\hat{\mathcal{S}})}^{k+1}\right\|_2^2 \geq \left\|\boldsymbol{\theta}_{\mathcal{D}}^k - \frac{\eta'}{M}\boldsymbol{g}_{\mathcal{D}}^k\right\|_2^2.$$

Using this, with the fact that $\boldsymbol{\theta}_{\mathcal{D}}^{k+1} = 0$, we get

$$(*) \leq \frac{M}{2} \left\| \Delta \boldsymbol{\theta}_{\mathcal{I}^k}^k + \frac{\eta'}{M} \boldsymbol{g}_{\mathcal{I}^k}^k \right\|_2^2 - \frac{M}{2} \left\| -\boldsymbol{\theta}_{\mathcal{D}}^k + \frac{\eta'}{M} \boldsymbol{g}_{\mathcal{D}}^k \right\|_2^2,$$

$$= \frac{M}{2} \left\| \boldsymbol{\theta}_{\mathcal{I}^k \setminus \mathcal{D}}^{k+1} - \boldsymbol{\theta}_{\mathcal{I}^k \setminus \mathcal{D}}^k + \frac{\eta'}{M} \boldsymbol{g}_{\mathcal{I}^k \setminus \mathcal{D}}^k \right\|_2^2. \tag{21}$$

We have bound $|\mathcal{I}^k \setminus \mathcal{D}| \leq |\mathcal{S}^{k+1}| + |(\mathcal{S}^k \setminus \mathcal{S}^{k+1}) \setminus \mathcal{D}| + |\hat{\mathcal{S}}| \leq s + |(\mathcal{S}^{k+1} \cap \hat{\mathcal{S}}) \setminus \mathcal{S}^k| + s^* \leq s + 2s^*$. Also, since $\mathcal{S}^{k+1} \subseteq \mathcal{I}^k \setminus \mathcal{D}$, we have $\boldsymbol{\theta}_{\mathcal{I}^k \setminus \mathcal{D}}^{k+1} = P_s(\boldsymbol{\theta}_{\mathcal{I}^k \setminus \mathcal{D}}^k - \frac{\eta'}{M} \boldsymbol{g}_{\mathcal{I}^k \setminus \mathcal{D}}^k)$. Using Lemma C.1 with (21):

$$(*) \leq \frac{M}{2} \cdot \frac{|\mathcal{I}^k \setminus \mathcal{D}| - s}{|\mathcal{I}^k \setminus \mathcal{D}| - s^*} \left\| \hat{\boldsymbol{\theta}}_{\mathcal{I}^k \setminus \mathcal{D}} - \boldsymbol{\theta}_{\mathcal{I}^k \setminus \mathcal{D}}^k + \frac{\eta'}{M} \boldsymbol{g}_{\mathcal{I}^k \setminus \mathcal{D}}^k \right\|_2^2.$$

Using $|\mathcal{I}^k \setminus \mathcal{D}| \leq s + 2s^*$ and the fact that $\frac{x-s}{x-s^*}$ is increasing for $x \geq s$ when $s \geq s^* \geq 0$, we get

$$\frac{|\mathcal{I}^k \setminus \mathcal{D}| - s}{|\mathcal{I}^k \setminus \mathcal{D}| - s^*} \leq \frac{(s + 2s^*) - s}{(s + 2s^*) - s^*} = \frac{2s^*}{s + s^*}.$$

Thus,

$$(*) \leq \frac{2s^*}{s + s^*} \cdot \frac{M}{2} \left\| \hat{\boldsymbol{\theta}}_{\mathcal{I}^k} - \boldsymbol{\theta}_{\mathcal{I}^k}^k + \frac{\eta'}{M} \boldsymbol{g}_{\mathcal{I}^k}^k \right\|_2^2,$$

$$= \frac{2s^*}{s + s^*} \left( \eta' \langle \hat{\boldsymbol{\theta}} - \boldsymbol{\theta}^k, \boldsymbol{g}^k \rangle + \frac{M}{2} \left\| \hat{\boldsymbol{\theta}} - \boldsymbol{\theta}^k \right\|_2^2 + \frac{(\eta')^2}{2M} \left\| \boldsymbol{g}_{\mathcal{I}^k}^k \right\|_2^2 \right).$$

Using RSC property, we get

$$(*) \leq \frac{2s^*}{s + s^*} \left( \eta' f(\hat{\boldsymbol{\theta}}) - \eta' f(\boldsymbol{\theta}^k) + \frac{M - \eta' m}{2} \left\| \hat{\boldsymbol{\theta}} - \boldsymbol{\theta}^k \right\|_2^2 + \frac{(\eta')^2}{2M} \left\| \boldsymbol{g}_{\mathcal{I}^k}^k \right\|_2^2 \right). \tag{22}$$

Combining (20) with (22) and setting $\eta' = 2/3$, $s = 32(\frac{M}{m})^2 s^*$, we have $\frac{2s^*}{s + s^*} \leq \frac{m^2}{16M(M - \eta' m)} \leq \frac{3}{16}$, and

$$f(\boldsymbol{\theta}^{k+1}) - f(\boldsymbol{\theta}^k) \leq \frac{2s^*}{s + s^*} \eta' \left( f(\hat{\boldsymbol{\theta}}) - f(\boldsymbol{\theta}^k) \right) + \frac{m^2}{32M} \left\| \hat{\boldsymbol{\theta}} - \boldsymbol{\theta}^k \right\|_2^2 + \frac{1}{24M} \left\| \boldsymbol{g}_{\mathcal{I}^k}^k \right\|_2^2$$

$$- \frac{2}{9M} \left\| \boldsymbol{g}_{\hat{\mathcal{S}} \cup \mathcal{S}^k}^k \right\|_2^2 - \frac{1}{9M} \left\| \boldsymbol{g}_{\mathcal{S}^{k+1} \cup \mathcal{S}^k}^k \right\|_2^2.$$

Using $\mathcal{S}^{k+1} \setminus (\mathcal{S}^k \cup \hat{\mathcal{S}}) \subseteq \mathcal{S}^{k+1} \cup \mathcal{S}^k$ and splitting $\left\| \boldsymbol{g}_{\mathcal{I}^k}^k \right\|_2^2 = \left\| \boldsymbol{g}_{\hat{\mathcal{S}} \cup \mathcal{S}^k}^k \right\|_2^2 + \left\| \boldsymbol{g}_{\mathcal{S}^{k+1} \setminus (\mathcal{S}^k \cup \hat{\mathcal{S}})}^k \right\|_2^2$ gives

$$f(\boldsymbol{\theta}^{k+1}) - f(\boldsymbol{\theta}^k) \leq \frac{2s^*}{s + s^*} \eta' \left( f(\hat{\boldsymbol{\theta}}) - f(\boldsymbol{\theta}^k) \right) + \frac{m^2}{32M} \left\| \hat{\boldsymbol{\theta}} - \boldsymbol{\theta}^k \right\|_2^2 - \frac{13}{72M} \left\| \boldsymbol{g}_{\hat{\mathcal{S}} \cup \mathcal{S}^k}^k \right\|_2^2 - \frac{5}{72M} \left\| \boldsymbol{g}_{\mathcal{S}^{k+1} \cup \mathcal{S}^k}^k \right\|_2^2,$$

$$\leq \frac{2s^*}{s + s^*} \eta' \left( f(\hat{\boldsymbol{\theta}}) - f(\boldsymbol{\theta}^k) \right) - \frac{13}{72M} \left( \left\| \boldsymbol{g}_{\hat{\mathcal{S}} \cup \mathcal{S}^k}^k \right\|_2^2 - \frac{m^2}{4} \left\| \hat{\boldsymbol{\theta}} - \boldsymbol{\theta}^k \right\|_2^2 \right),$$

$$\leq \frac{2s^*}{s + s^*} \eta' \left( f(\hat{\boldsymbol{\theta}}) - f(\boldsymbol{\theta}^k) \right) - \frac{m}{12M} \left( f(\boldsymbol{\theta}^k) - f(\hat{\boldsymbol{\theta}}) \right), \tag{23}$$

where the last inequality above follows using Lemma C.2. Therefore,

$$f(\boldsymbol{\theta}^{k+1}) - f(\hat{\boldsymbol{\theta}}) \leq \left( 1 - \frac{m}{12M} \right) \left( f(\boldsymbol{\theta}^k) - f(\hat{\boldsymbol{\theta}}) \right) + \frac{2s^*}{s + s^*} \eta' \left( f(\hat{\boldsymbol{\theta}}) - f(\boldsymbol{\theta}^k) \right),$$

$$\leq \left( 1 - \frac{m}{12M} \right) \left( f(\boldsymbol{\theta}^k) - f(\hat{\boldsymbol{\theta}}) \right).$$

Iterating this yields

$$f(\boldsymbol{\theta}^k) - f(\hat{\boldsymbol{\theta}}) \leq \left( 1 - \frac{m}{12M} \right)^k \left( f(\boldsymbol{\theta}^0) - f(\hat{\boldsymbol{\theta}}) \right).$$

Therefore, $\forall \epsilon > 0$, the $k = O\left( \frac{M}{m} \log\left( \frac{f(\boldsymbol{\theta}^0) - f(\hat{\boldsymbol{\theta}})}{\epsilon} \right) \right)$-th iterate satisfies $f(\boldsymbol{\theta}^k) - f(\hat{\boldsymbol{\theta}}) \leq \epsilon$.

- **(Greedy pursuit)** Let $\mathcal{I}^k := \mathcal{S}^k \cup \{\text{top } S \text{ indices of } \nabla_{\boldsymbol{\theta}} f(\boldsymbol{\theta}^k) \text{ in magnitude}\}$. Define the intermediate point $\boldsymbol{z}^k$ by

$$\boldsymbol{z}^k_{\mathcal{S}^k} = \boldsymbol{\theta}^k_{\mathcal{S}^k}, \qquad \boldsymbol{z}^k_{\mathcal{I}^k \setminus \mathcal{S}^k} = -\frac{1}{M}\, \boldsymbol{g}^k_{\mathcal{I}^k \setminus \mathcal{S}^k}, \qquad \boldsymbol{z}^k_{(\mathcal{I}^k)^c} = \boldsymbol{0}.$$

Let $M = 2M_{s+S}$, and $m = 2m_{s+S+s^*}$. Since $\boldsymbol{g}^k_{\mathcal{S}^k} = \boldsymbol{0}$ and $\mathcal{S}^k \subseteq \mathcal{I}^k$,

$$f(\boldsymbol{z}^k) - f(\boldsymbol{\theta}^k) \leq \langle \boldsymbol{g}^k, \boldsymbol{z}^k - \boldsymbol{\theta}^k \rangle + \frac{M}{2}\, \left\| \boldsymbol{z}^k - \boldsymbol{\theta}^k \right\|_2^2 = -\frac{1}{2M}\, \left\| \boldsymbol{g}^k_{\mathcal{I}^k \setminus \mathcal{S}^k} \right\|_2^2.$$

Since $\mathcal{I}^k \setminus \mathcal{S}^k$ consists of the top $S$ coordinates of $\boldsymbol{g}^k$ in magnitude outside $\mathcal{S}^k$, for any set $J \subseteq (\mathcal{S}^k)^c$ with $|J| \leq S$,

$$\|\boldsymbol{g}^k_{\mathcal{I}^k \setminus \mathcal{S}^k}\|_2^2 \geq \|\boldsymbol{g}^k_J\|_2^2.$$

Taking $J = \hat{\mathcal{S}} \setminus \mathcal{S}^k$, and using $|\hat{\mathcal{S}} \setminus \mathcal{S}^k| \leq |\hat{\mathcal{S}}| \leq s^* \leq S$, we have

$$\|\boldsymbol{g}^k_{\mathcal{I}^k \setminus \mathcal{S}^k}\|_2^2 \geq \|\boldsymbol{g}^k_{\hat{\mathcal{S}} \setminus \mathcal{S}^k}\|_2^2.$$

By Lemma C.3, we obtain

$$\left\| \boldsymbol{g}^k_{\hat{\mathcal{S}} \setminus \mathcal{S}^k} \right\|_2^2 = \left\| \boldsymbol{g}^k_{\hat{\mathcal{S}} \cup \mathcal{S}^k} \right\|_2^2 \geq 2m\left(f(\boldsymbol{\theta}^k) - f(\hat{\boldsymbol{\theta}})\right).$$

Therefore

$$f(\boldsymbol{z}^k) - f(\boldsymbol{\theta}^k) \leq -\frac{m}{M}\left(f(\boldsymbol{\theta}^k) - f(\hat{\boldsymbol{\theta}})\right), \quad \text{i.e.} \quad f(\boldsymbol{z}^k) - f(\hat{\boldsymbol{\theta}}) \leq \left(1 - \frac{m}{M}\right)\left(f(\boldsymbol{\theta}^k) - f(\hat{\boldsymbol{\theta}})\right).$$

Let $\boldsymbol{\beta}^k = \arg\min_{\boldsymbol{\beta},\, \mathrm{supp}(\boldsymbol{\beta}) \subseteq \mathcal{I}^k} f(\boldsymbol{\beta})$ and $\tilde{\boldsymbol{\theta}}^k = P_s(\boldsymbol{\beta}^k)$. Then $f(\boldsymbol{\beta}^k) \leq f(\boldsymbol{z}^k)$, and by Lemma C.4,

$$
\begin{aligned}
f(\boldsymbol{\theta}^{k+1}) - f(\hat{\boldsymbol{\theta}}) &\leq f(\tilde{\boldsymbol{\theta}}^k) - f(\hat{\boldsymbol{\theta}}) \\
&= \left(f(\boldsymbol{\beta}^k) - f(\hat{\boldsymbol{\theta}})\right) + \left(f(\tilde{\boldsymbol{\theta}}^k) - f(\boldsymbol{\beta}^k)\right) \\
&\leq \left(1 + \frac{2M}{m} \cdot \frac{S}{s + S - s^*}\right)\left(f(\boldsymbol{\beta}^k) - f(\hat{\boldsymbol{\theta}})\right) + \frac{2M}{m^2} \cdot \frac{S(s + S + s^*)}{s + S - s^*} \cdot \|\nabla f(\hat{\boldsymbol{\theta}})\|_\infty^2 \\
&\leq \left(1 - \frac{m}{M}\right)\left(1 + \frac{2M}{m} \cdot \frac{S}{s + S - s^*}\right)\left(f(\boldsymbol{\theta}^k) - f(\hat{\boldsymbol{\theta}})\right) + \frac{2M}{m^2} \cdot \frac{S(s + S + s^*)}{s + S - s^*} \cdot \|\nabla f(\hat{\boldsymbol{\theta}})\|_\infty^2.
\end{aligned}
$$

Under the assumed size condition $s + S - s^* \geq 8\left(\frac{M}{m}\right)^2 S$, the contraction factor $\rho := \left(1 - \frac{m}{M}\right)\left(1 + \frac{2M}{m} \cdot \frac{S}{s+S-s^*}\right)$ satisfies $\rho \leq 1 - \frac{m}{2M} < 1$. Define

$$\Xi := \frac{2M}{m^2} \cdot \frac{S(s + S + s^*)}{s + S - s^*} \cdot \|\nabla f(\hat{\boldsymbol{\theta}})\|_\infty^2.$$

Then, iterating the recursion $f(\boldsymbol{\theta}^{k+1}) - f(\hat{\boldsymbol{\theta}}) \leq \rho\left(f(\boldsymbol{\theta}^k) - f(\hat{\boldsymbol{\theta}})\right) + \Xi$ yields

$$f(\boldsymbol{\theta}^k) - f(\hat{\boldsymbol{\theta}}) \leq \rho^k\left(f(\boldsymbol{\theta}^0) - f(\hat{\boldsymbol{\theta}})\right) + \frac{\Xi}{1 - \rho} := \varepsilon_k.$$

Lemma C.5 implies $\|\nabla f(\hat{\boldsymbol{\theta}})\|_\infty \leq C_\nabla \|\nabla f(\boldsymbol{\theta}^*)\|_\infty$, where $C_\nabla := 1 + \frac{2M_{2s^*+1}\sqrt{2s^*}}{m_{2s^*}}$.

Using Proposition 2 from Wong et al. (2020), there exists a positive constant $\tilde{c}$, and a free parameter $b > 0$, such that for $T \geq \sqrt{\frac{b+1}{\tilde{c}}} \log(pq)$, we have

$$\mathbb{P}\left(\|\nabla f(\boldsymbol{\theta}^*)\|_\infty \leq \frac{Q}{2} S_\alpha(T) \sqrt{\frac{\log(pq)}{T}}\right) \geq 1 - 8\exp\left(-b\log(pq)\right), \tag{24}$$

$$\text{where} \quad Q = 16\pi \sqrt{\frac{b+1}{\tilde{c}}}\left(\|\Sigma_{\boldsymbol{X}}\|_2\left(1 + \max_{1 \leq i \leq p}\|\boldsymbol{\Theta}^*_{:i}\|_2^2\right) + \|\Sigma_{\boldsymbol{Y}}\|_2\right).$$

Hence, on the event in Equation (24),

$$\Xi \le \frac{M}{2m^2} \cdot \frac{S(s + S + s^*)}{s + S - s^*} \cdot C_\nabla^2 Q^2 S_\alpha(T)^2 \frac{\log(2pq)}{T}.$$

Since $\rho \le 1 - \frac{m}{2M}$, we have

$$\frac{\Xi}{1 - \rho} \le \frac{M^2}{m^3} \cdot \frac{S(s + S + s^*)}{s + S - s^*} \cdot C_\nabla^2 Q^2 S_\alpha(T)^2 \frac{\log(2pq)}{T} =: \epsilon_{\mathrm{GP}}.$$

Therefore, for any $\epsilon > \epsilon_{\mathrm{GP}}$, if

$$k \ge C_2 \kappa(\Sigma_{\boldsymbol{X}}) \log\left(\frac{f(\mathbf{0})}{\epsilon - \epsilon_{\mathrm{GP}}}\right),$$

then

$$\varepsilon_k \le \epsilon,$$

and hence

$$f(\boldsymbol{\theta}^k) - \epsilon \le f(\hat{\boldsymbol{\theta}}).$$

Consider an arbitrary $s^*$-sparse vector $\bar{\boldsymbol{\theta}}$. Let $\boldsymbol{\delta} := \bar{\boldsymbol{\theta}} - \boldsymbol{\theta}^k$ and $|\mathcal{S}_{\boldsymbol{\delta}}| := |\mathrm{supp}(\boldsymbol{\delta})| \le s + s^*$. Since $\hat{\boldsymbol{\theta}}$ is the empirical loss minimizer over the set of $s^*$-sparse vectors, by the choice of $k$ we have

$$f(\boldsymbol{\theta}^k) - \varepsilon \ \le \ f(\hat{\boldsymbol{\theta}}) \ \le \ f(\bar{\boldsymbol{\theta}}).$$

Apply the RSC at sparsity $s + s^*$, we get

$$f(\bar{\boldsymbol{\theta}}) \ \le \ f(\boldsymbol{\theta}^k) + \left\langle \nabla f(\bar{\boldsymbol{\theta}}), \bar{\boldsymbol{\theta}} - \boldsymbol{\theta}^k \right\rangle - m_{s+s^*} \|\boldsymbol{\delta}\|_2^2.$$

Combining the two displays yields

$$-\varepsilon \ \le \ \langle \nabla f(\bar{\boldsymbol{\theta}}), \boldsymbol{\delta} \rangle - m_{s+s^*} \|\boldsymbol{\delta}\|_2^2.$$

Since $\langle \nabla f(\bar{\boldsymbol{\theta}}), \boldsymbol{\delta} \rangle \le \|\nabla f(\bar{\boldsymbol{\theta}})\|_\infty \|\boldsymbol{\delta}\|_1 \le \sqrt{s + s^*} \|\nabla f(\bar{\boldsymbol{\theta}})\|_\infty \|\boldsymbol{\delta}\|_2$, we have

$$m_{s+s^*} \|\boldsymbol{\delta}\|_2^2 - \sqrt{s + s^*} \|\nabla f(\bar{\boldsymbol{\theta}})\|_\infty \|\boldsymbol{\delta}\|_2 - \varepsilon \ \le \ 0.$$

Solving the quadratic inequality gives

$$\|\boldsymbol{\delta}\|_2 \le \frac{\sqrt{s + s^*} \|\nabla f(\bar{\boldsymbol{\theta}})\|_\infty}{m_{s+s^*}} + \sqrt{\frac{\varepsilon}{m_{s+s^*}}}. \tag{25}$$

When $\bar{\boldsymbol{\theta}} = \boldsymbol{\theta}^*$, let $\tilde{\boldsymbol{w}} := \tilde{\boldsymbol{y}} - \tilde{\boldsymbol{X}} \boldsymbol{\theta}^*$, then $\tilde{\boldsymbol{w}} := \mathrm{vec}(\boldsymbol{W})$, $\nabla f(\boldsymbol{\theta}^*) = \frac{1}{T} \tilde{\boldsymbol{X}}^\top (\tilde{\boldsymbol{X}} \boldsymbol{\theta}^* - \tilde{\boldsymbol{y}}) = -\frac{1}{T} \tilde{\boldsymbol{X}}^\top \tilde{\boldsymbol{w}}$. Therefore, we have $\|\nabla f(\boldsymbol{\theta}^*)\|_\infty = \frac{1}{T} \|\boldsymbol{X}^\top \boldsymbol{W}\|_\infty$.

Combining Equation (25) and Equation (24) yields the theorem. $\qquad\square$

The proof above relies on the following lemmas, which are adapted from Jain et al. (2014).

**Lemma C.1** (Lemma 1 from (Jain et al., 2014)). *For any index set $\mathcal{I}$, any $\boldsymbol{z} \in \mathbb{R}^{|\mathcal{I}|}$, let $\boldsymbol{\theta} = P_s(\boldsymbol{z})$. Then for any $\hat{\boldsymbol{\theta}} \in \mathbb{R}^{|\mathcal{I}|}$ such that $\|\hat{\boldsymbol{\theta}}\|_0 \le s^*$, $s^* < s$, we have*

$$\frac{\|\boldsymbol{\theta} - \boldsymbol{z}\|_2^2}{|\mathcal{I}| - s} \le \frac{\|\hat{\boldsymbol{\theta}} - \boldsymbol{z}\|_2^2}{|\mathcal{I}| - s^*}.$$

**Lemma C.2.** *Let $m = m_{2s+s^*}$. For any iterate $\boldsymbol{\theta}^k$, we have*

$$\left\|\boldsymbol{g}_{\mathcal{S}^k \cup \hat{\mathcal{S}}}^k\right\|_2^2 - \frac{m^2}{4} \left\|\hat{\boldsymbol{\theta}} - \boldsymbol{\theta}^k\right\|_2^2 \ge \frac{m}{2} \left(f(\boldsymbol{\theta}^k) - f(\hat{\boldsymbol{\theta}})\right),$$

*where $\boldsymbol{g}^k$ is the gradient of $f$ at $\boldsymbol{\theta}^k$.*

**Lemma C.3** (Lemma 3 from (Jain et al., 2014)). *Consider a function $f$ satisfying RSC/RSS at sparsity level $2s + s^*$ with parameters $M_{2s+s^*}(f) = M$, $m_{2s+s^*}(f) = m$. Let $\hat{\boldsymbol{\theta}} = \arg\min_{\boldsymbol{\theta}, \|\boldsymbol{\theta}\|_0 \le s^*} f(\boldsymbol{\theta})$ with $\hat{\mathcal{S}} = \operatorname{supp}(\hat{\boldsymbol{\theta}})$. Fix any subset $\mathcal{S}^k \subseteq [pq]$ with $|\mathcal{S}^k| \le s$ and define the fully-corrective estimator $\boldsymbol{\theta}^k = \arg\min_{\boldsymbol{\theta}, \operatorname{supp}(\boldsymbol{\theta}) \subseteq \mathcal{S}^k} f(\boldsymbol{\theta})$. Write $\boldsymbol{g}^k := \nabla f(\boldsymbol{\theta}^k)$ and let $\mathcal{M}_k := \hat{\mathcal{S}} \setminus \mathcal{S}^k$ and $\mathcal{F}_k := \mathcal{S}^k \setminus \hat{\mathcal{S}}$. Then,*

$$\left\|\boldsymbol{g}^k_{\mathcal{S}^k \cup \hat{\mathcal{S}}}\right\|_2^2 \ge 2m\left(f(\boldsymbol{\theta}^k) - f(\hat{\boldsymbol{\theta}})\right) + m^2 \left\|\boldsymbol{\theta}^k_{\mathcal{F}_k}\right\|_2^2.$$

**Lemma C.4.** *Let $\mathcal{I}^k \subseteq [pq]$ with $|\mathcal{I}^k| \le s + S$ and $\boldsymbol{\beta}^k = \arg\min_{\boldsymbol{\beta}, \operatorname{supp}(\boldsymbol{\beta}) \subseteq \mathcal{I}^k} f(\boldsymbol{\beta})$. Set $r := |\operatorname{supp}(\boldsymbol{\beta}^k - \hat{\boldsymbol{\theta}})| \le (s + S) + s^*$, $M = 2M_{s+S}$, and $m = 2m_{s+S+s^*}$. Let $\tilde{\boldsymbol{\theta}}^k = P_s(\boldsymbol{\beta}^k)$ be the hard-thresholded vector. Then*

$$f(\tilde{\boldsymbol{\theta}}^k) - f(\boldsymbol{\beta}^k) \le \frac{2M}{m} \cdot \frac{S}{s + S - s^*} \cdot \left(f(\boldsymbol{\beta}^k) - f(\hat{\boldsymbol{\theta}})\right) + \frac{2M}{m^2} \cdot \frac{S}{s + S - s^*} \cdot r \|\nabla f(\hat{\boldsymbol{\theta}})\|_\infty^2.$$

*Proof.* Write $\Delta := \tilde{\boldsymbol{\theta}}^k - \boldsymbol{\beta}^k$. By full correction on $\mathcal{I}^k$, we have $(\nabla f(\boldsymbol{\beta}^k))_{\mathcal{I}^k} = \mathbf{0}$. Moreover, $\operatorname{supp}(\Delta) \subseteq \mathcal{I}^k$; hence $\langle \nabla f(\boldsymbol{\beta}^k), \Delta \rangle = 0$. Applying RSS (order $s + S$) yields

$$f(\tilde{\boldsymbol{\theta}}^k) - f(\boldsymbol{\beta}^k) \le \langle \nabla f(\boldsymbol{\beta}^k), \Delta \rangle + \frac{M}{2} \|\Delta\|_2^2 = \frac{M}{2} \|\Delta\|_2^2.$$

By Lemma C.1,

$$\|\Delta\|_2^2 = \|P_s(\boldsymbol{\beta}^k) - \boldsymbol{\beta}^k\|_2^2 \le \frac{S}{s + S - s^*} \|\boldsymbol{\beta}^k - \hat{\boldsymbol{\theta}}\|_2^2.$$

Let $r := |\operatorname{supp}(\boldsymbol{\beta}^k - \hat{\boldsymbol{\theta}})| \le s + S + s^*$. Applying RSC gives

$$f(\boldsymbol{\beta}^k) - f(\hat{\boldsymbol{\theta}}) \ge \langle \nabla f(\hat{\boldsymbol{\theta}}), \boldsymbol{\beta}^k - \hat{\boldsymbol{\theta}} \rangle + \frac{m}{2} \|\boldsymbol{\beta}^k - \hat{\boldsymbol{\theta}}\|_2^2.$$

Using $\langle a, b \rangle \ge -\|a\|_\infty \|b\|_1 \ge -\sqrt{r}\, \|a\|_\infty \|b\|_2$, we obtain for $t := \|\boldsymbol{\beta}^k - \hat{\boldsymbol{\theta}}\|_2$,

$$f(\boldsymbol{\beta}^k) - f(\hat{\boldsymbol{\theta}}) \ge -\sqrt{r}\, \|\nabla f(\hat{\boldsymbol{\theta}})\|_\infty\, t + \frac{m}{2} t^2,$$

which implies

$$t^2 \le \frac{4}{m}\left(f(\boldsymbol{\beta}^k) - f(\hat{\boldsymbol{\theta}})\right) + \frac{4r}{m^2} \|\nabla f(\hat{\boldsymbol{\theta}})\|_\infty^2.$$

Combining the above displays yields

$$f(\tilde{\boldsymbol{\theta}}^k) - f(\boldsymbol{\beta}^k) \le \frac{M}{2} \cdot \frac{S}{s + S - s^*} \left(\frac{4}{m}(f(\boldsymbol{\beta}^k) - f(\hat{\boldsymbol{\theta}})) + \frac{4r}{m^2} \|\nabla f(\hat{\boldsymbol{\theta}})\|_\infty^2\right),$$

which proves the claim. $\qquad\square$

**Lemma C.5.** *Let $\boldsymbol{\theta}^*$ and $\hat{\boldsymbol{\theta}}$ be defined as above. Then the gradient of $f$ at $\hat{\boldsymbol{\theta}}$ is controlled by the gradient at $\boldsymbol{\theta}^*$ as*

$$\|\nabla f(\hat{\boldsymbol{\theta}})\|_\infty \le \left(1 + \frac{2M_{2s^*+1}\sqrt{2s^*}}{m_{2s^*}}\right) \|\nabla f(\boldsymbol{\theta}^*)\|_\infty.$$

*Proof.* Since $\hat{\boldsymbol{\theta}}$ minimizes $f$ over the support $\hat{\mathcal{S}} := \operatorname{supp}(\hat{\boldsymbol{\theta}})$, we have $(\nabla f(\hat{\boldsymbol{\theta}}))_{\hat{\mathcal{S}}} = \mathbf{0}$. For the quadratic loss,

$$\nabla f(\hat{\boldsymbol{\theta}}) - \nabla f(\boldsymbol{\theta}^*) = \tilde{\Gamma}(\hat{\boldsymbol{\theta}} - \boldsymbol{\theta}^*), \qquad \tilde{\Gamma} := \frac{1}{T} \tilde{\boldsymbol{X}}^\top \tilde{\boldsymbol{X}}.$$

For any coordinate $j$, since $|\operatorname{supp}(\hat{\boldsymbol{\theta}} - \boldsymbol{\theta}^*) \cup \{j\}| \le 2s^* + 1$, the RSS property implies

$$|e_j^\top \tilde{\Gamma}(\hat{\boldsymbol{\theta}} - \boldsymbol{\theta}^*)| \le 2M_{2s^*+1}\|\hat{\boldsymbol{\theta}} - \boldsymbol{\theta}^*\|_2.$$

Hence

$$\|\nabla f(\hat{\boldsymbol{\theta}})\|_\infty \le \|\nabla f(\boldsymbol{\theta}^*)\|_\infty + 2M_{2s^*+1}\|\hat{\boldsymbol{\theta}} - \boldsymbol{\theta}^*\|_2.$$

Next, since $\hat{\boldsymbol{\theta}}$ is the empirical minimizer over all $s^*$-sparse vectors and $\|\boldsymbol{\theta}^*\|_0 \leq s^*$, we have $f(\hat{\boldsymbol{\theta}}) \leq f(\boldsymbol{\theta}^*)$. Applying RSC of order $2s^*$ at $(\hat{\boldsymbol{\theta}}, \boldsymbol{\theta}^*)$ gives

$$f(\hat{\boldsymbol{\theta}}) - f(\boldsymbol{\theta}^*) \geq \langle \nabla f(\boldsymbol{\theta}^*), \hat{\boldsymbol{\theta}} - \boldsymbol{\theta}^* \rangle + m_{2s^*} \|\hat{\boldsymbol{\theta}} - \boldsymbol{\theta}^*\|_2^2.$$

Since the left-hand side is non-positive,

$$m_{2s^*} \|\hat{\boldsymbol{\theta}} - \boldsymbol{\theta}^*\|_2^2 \leq -\langle \nabla f(\boldsymbol{\theta}^*), \hat{\boldsymbol{\theta}} - \boldsymbol{\theta}^* \rangle \leq \|\nabla f(\boldsymbol{\theta}^*)\|_\infty \|\hat{\boldsymbol{\theta}} - \boldsymbol{\theta}^*\|_1.$$

Using $\|\hat{\boldsymbol{\theta}} - \boldsymbol{\theta}^*\|_1 \leq \sqrt{2s^*} \|\hat{\boldsymbol{\theta}} - \boldsymbol{\theta}^*\|_2$ yields

$$\|\hat{\boldsymbol{\theta}} - \boldsymbol{\theta}^*\|_2 \leq \frac{\sqrt{2s^*}}{m_{2s^*}} \|\nabla f(\boldsymbol{\theta}^*)\|_\infty.$$

Substituting this into the previous display completes the proof. $\qquad\square$

## D. Verification of Assumptions for Gaussian VARs

This appendix section verifies that the Gaussian VAR($d$) model in Section 3.3 satisfies Assumptions 2.2–2.4. As is standard, we consider a two-sided extension $\{\boldsymbol{Z}_t\}_{t\in\mathbb{Z}}$ of the process throughout; all statements then apply to the observed segment $\{(\boldsymbol{Z}_t)_{t=1}^{T+d}\}$ by restriction.

Let $\{\boldsymbol{Z}_t\}_{t\in\mathbb{Z}} \subset \mathbb{R}^p$ follow the finite-order VAR($d$) recursion

$$\boldsymbol{Z}_t = \sum_{k=1}^d \boldsymbol{A}_k \boldsymbol{Z}_{t-k} + \mathcal{E}_t, \qquad \mathcal{E}_t \overset{\text{i.i.d.}}{\sim} \mathcal{N}(\boldsymbol{0}, \Sigma_\varepsilon), \tag{26}$$

where $\Sigma_\varepsilon$ satisfies $0 < \lambda_{\min}(\Sigma_\varepsilon) \leq \lambda_{\max}(\Sigma_\varepsilon) < \infty$. Assume the VAR($d$) is *stable*:

$$\det\left(\boldsymbol{I}_p - \sum_{k=1}^d \boldsymbol{A}_k z^k\right) \neq 0, \qquad \forall\, |z| \leq 1. \tag{27}$$

Recall the definitions: $\boldsymbol{X}_t := (\boldsymbol{Z}_{t-1}^\top, \boldsymbol{Z}_{t-2}^\top, \ldots, \boldsymbol{Z}_{t-d}^\top)^\top \in \mathbb{R}^{dp}$, $\boldsymbol{Y}_t := \boldsymbol{Z}_t \in \mathbb{R}^p$. Define the companion matrix $\mathcal{A} \in \mathbb{R}^{dp \times dp}$ and the injection matrix $\boldsymbol{R} \in \mathbb{R}^{dp \times p}$ by

$$\mathcal{A} := \begin{pmatrix} \boldsymbol{A}_1 & \boldsymbol{A}_2 & \cdots & \boldsymbol{A}_{d-1} & \boldsymbol{A}_d \\ \boldsymbol{I}_p & \boldsymbol{0} & \cdots & \boldsymbol{0} & \boldsymbol{0} \\ & \ddots & \ddots & \vdots & \vdots \\ & & \boldsymbol{I}_p & \boldsymbol{0} & \boldsymbol{0} \end{pmatrix}, \qquad \boldsymbol{R} := \begin{pmatrix} \boldsymbol{I}_p \\ \boldsymbol{0} \\ \vdots \\ \boldsymbol{0} \end{pmatrix}.$$

Then the VAR($d$) recursion is equivalent to the companion-form VAR(1)

$$\boldsymbol{X}_t = \mathcal{A} \boldsymbol{X}_{t-1} + \boldsymbol{u}_t, \qquad \boldsymbol{u}_t := \boldsymbol{R} \mathcal{E}_t = (\mathcal{E}_t^\top, \boldsymbol{0}^\top, \ldots, \boldsymbol{0}^\top)^\top \in \mathbb{R}^{dp}. \tag{28}$$

The stability condition is equivalent to $\rho(\mathcal{A}) < 1$. Consequently, there exist constants $C_\mathcal{A} > 0$ and $0 < r < 1$ such that

$$\|\mathcal{A}^j\|_2 \leq C_\mathcal{A} r^j, \qquad \forall j \geq 0. \tag{29}$$

Define the partial sums

$$\boldsymbol{X}_t^{(n)} := \sum_{j=0}^n \mathcal{A}^j \boldsymbol{R} \mathcal{E}_{t-j}, \qquad n \geq 0.$$

We first show that $\{\boldsymbol{X}_t^{(n)}\}_{n \geq 0}$ is Cauchy in $L^2$. For $n > m$, using independence of the innovations and $\mathbb{E}[\mathcal{E}_t] = \boldsymbol{0}$,

$$
\begin{aligned}
\mathbb{E}\big\|\boldsymbol{X}_t^{(n)} - \boldsymbol{X}_t^{(m)}\big\|_2^2 &= \sum_{j=m+1}^{n} \mathbb{E}\big\|\mathcal{A}^j \boldsymbol{R}\, \mathcal{E}_{t-j}\big\|_2^2 \\
&= \sum_{j=m+1}^{n} \mathrm{tr}\Big(\mathcal{A}^j \boldsymbol{R}\, \Sigma_\varepsilon\, \boldsymbol{R}^\top (\mathcal{A}^j)^\top\Big) \\
&\leq \lambda_{\max}(\Sigma_\varepsilon) \sum_{j=m+1}^{n} \|\mathcal{A}^j \boldsymbol{R}\|_F^2 \\
&\leq \lambda_{\max}(\Sigma_\varepsilon)\, \|\boldsymbol{R}\|_F^2 \sum_{j=m+1}^{n} \|\mathcal{A}^j\|_2^2 \\
&\leq \lambda_{\max}(\Sigma_\varepsilon)\, \|\boldsymbol{R}\|_F^2\, C_{\mathcal{A}}^2 \sum_{j=m+1}^{n} r^{2j}.
\end{aligned}
$$

Since $\sum_{j \geq 0} r^{2j} < \infty$, the right-hand side tends to $0$ as $m, n \to \infty$. Hence $\{\boldsymbol{X}_t^{(n)}\}$ is Cauchy in $L^2$, and therefore converges in $L^2$ to a limit, denoted by

$$
\boldsymbol{X}_t := \sum_{j=0}^{\infty} \mathcal{A}^j \boldsymbol{R}\, \mathcal{E}_{t-j} \qquad \text{(convergence in } L^2 \text{)}. \tag{30}
$$

Moreover,

$$
\sum_{j=0}^{\infty} \mathbb{E}\big\|\mathcal{A}^j \boldsymbol{R}\, \mathcal{E}_{t-j}\big\|_2^2 \leq \lambda_{\max}(\Sigma_\varepsilon)\, \|\boldsymbol{R}\|_F^2 \sum_{j=0}^{\infty} \|\mathcal{A}^j\|_2^2 < \infty,
$$

so by Kolmogorov's convergence theorem applied coordinatewise, the series in Equation (30) also converges almost surely. Thus Equation (30) defines a well-posed causal MA($\infty$) solution of the companion-form process.

Let $\boldsymbol{J} := (\boldsymbol{I}_p, \boldsymbol{0}, \ldots, \boldsymbol{0}) \in \mathbb{R}^{p \times dp}$, so that $\boldsymbol{Z}_t = \boldsymbol{J} \boldsymbol{X}_t$. Applying the continuous linear map $\boldsymbol{J}$ to Equation (30) yields

$$
\boldsymbol{Z}_t = \sum_{j=0}^{\infty} \boldsymbol{\Phi}_j\, \mathcal{E}_{t-j}, \qquad \boldsymbol{\Phi}_j := \boldsymbol{J}\, \mathcal{A}^j \boldsymbol{R}, \quad j \geq 0, \tag{31}
$$

where the series converges both in $L^2$ and almost surely. Note that $\boldsymbol{\Phi}_0 = \boldsymbol{I}_p$, and $\{\boldsymbol{\Phi}_j\}_{j \geq 0}$ satisfies the recursion

$$
\boldsymbol{\Phi}_j = \sum_{k=1}^{\min(j,d)} \boldsymbol{A}_k\, \boldsymbol{\Phi}_{j-k}, \qquad j \geq 1.
$$

Furthermore, by Equation (29) and $\|\boldsymbol{J}\|_2 = \|\boldsymbol{R}\|_2 = 1$,

$$
\|\boldsymbol{\Phi}_j\|_2 \leq \|\boldsymbol{J}\|_2\, \|\mathcal{A}^j\|_2\, \|\boldsymbol{R}\|_2 \leq C_{\mathcal{A}} r^j, \qquad j \geq 0. \tag{32}
$$

Hence

$$
\sum_{j=0}^{\infty} \|\boldsymbol{\Phi}_j\|_2^2 < \infty, \qquad \sum_{j=0}^{\infty} \|\boldsymbol{\Phi}_j\|_F^2 < \infty. \tag{33}
$$

**Strict stationarity.** For each $n$, the finite moving-average process $\{\boldsymbol{Z}_t^{(n)}\}_{t \in \mathbb{Z}}$ is strictly stationary because it is a deterministic linear map of the i.i.d. innovations $\{\mathcal{E}_t\}$. Since $\boldsymbol{Z}_t^{(n)} \to \boldsymbol{Z}_t$ in $L^2$ (hence in distribution) for each fixed $t$, standard limit arguments imply that the limiting process $\{\boldsymbol{Z}_t\}_{t \in \mathbb{Z}}$ is also strictly stationary.

**Centering.** Because $\mathbb{E}[\mathcal{E}_t] = \boldsymbol{0}$ and the series converges in $L^2$,

$$
\mathbb{E}[\boldsymbol{Z}_t] = \sum_{j=0}^{\infty} \boldsymbol{\Phi}_j\, \mathbb{E}[\mathcal{E}_{t-j}] = \boldsymbol{0}.
$$

**Gaussianity.** Each $\boldsymbol{Z}_t$ in Equation (31) is an (almost surely convergent) linear functional of the jointly Gaussian sequence $\{\mathcal{E}_t\}_{t\in\mathbb{Z}}$, and therefore $\{\boldsymbol{Z}_t\}$ is a strictly stationary Gaussian process. In particular, all finite-dimensional vectors $(\boldsymbol{Z}_{t_1}, \ldots, \boldsymbol{Z}_{t_m})$ are multivariate Gaussian.

Since $(\boldsymbol{X}_t, \boldsymbol{Y}_t)$ is a linear transformation of $(\boldsymbol{Z}_t, \boldsymbol{Z}_{t-1}, \ldots, \boldsymbol{Z}_{t-d})$, it follows that $(\boldsymbol{X}_t, \boldsymbol{Y}_t)$ is jointly Gaussian and centered for each $t$. Moreover, strict stationarity of $\{\boldsymbol{Z}_t\}$ implies strict stationarity of $\{(\boldsymbol{X}_t, \boldsymbol{Y}_t)\}$, thereby verifying Assumptions 2.2–2.3 for the design/response pairs.

$\alpha$-**Mixing.** Let $\Sigma_{\boldsymbol{X}} := \mathrm{Cov}(\boldsymbol{X}_t)$ and $\Gamma_{\boldsymbol{X}}(\ell) := \mathrm{Cov}(\boldsymbol{X}_t, \boldsymbol{X}_{t+\ell})$ for $\ell \geq 0$. Stationarity and Equation (28) imply $\Gamma_{\boldsymbol{X}}(\ell) = \mathcal{A}^\ell \Sigma_{\boldsymbol{X}}$. For jointly Gaussian vectors, the maximal correlation (the $\rho$-mixing coefficient, Definition E.2) satisfies

$$\rho(\ell) = \left\|\Sigma_{\boldsymbol{X}}^{-1/2}\Gamma_{\boldsymbol{X}}(\ell)\Sigma_{\boldsymbol{X}}^{-1/2}\right\|_2 = \left\|\Sigma_{\boldsymbol{X}}^{-1/2}\mathcal{A}^\ell\Sigma_{\boldsymbol{X}}^{1/2}\right\|_2 \leq \sqrt{\kappa(\Sigma_{\boldsymbol{X}})}\,\|\mathcal{A}^\ell\|_2,$$

with $\kappa(\Sigma_{\boldsymbol{X}}) = \lambda_{\max}(\Sigma_{\boldsymbol{X}})/\lambda_{\min}(\Sigma_{\boldsymbol{X}})$. Combining with Equation (29) yields

$$\rho(\ell) \leq \sqrt{\kappa(\Sigma_{\boldsymbol{X}})}\,C_{\mathcal{A}}\,r^\ell.$$

By Fact E.3, $\alpha(\ell) \leq \rho(\ell)$ for all $\ell \geq 0$. Therefore,

$$\alpha(\ell) \leq \sqrt{\kappa(\Sigma_{\boldsymbol{X}})}\,C_{\mathcal{A}}\,r^\ell, \qquad \ell \geq 0,$$

which establishes geometric $\alpha$-mixing.

**Bounding condition number.** From Equation (28), we get $\Sigma_{\boldsymbol{X}} = \mathcal{A}\Sigma_{\boldsymbol{X}}\mathcal{A}^\top + \Sigma_u$ and $\Sigma_{\boldsymbol{Y}} = \boldsymbol{J}\Sigma_{\boldsymbol{X}}\boldsymbol{J}^\top$, where $\boldsymbol{J} := (\boldsymbol{I}_p, \boldsymbol{0}, \ldots, \boldsymbol{0})$, and $\Sigma_u = \begin{pmatrix} \Sigma_\varepsilon & \boldsymbol{0} \\ \boldsymbol{0} & \boldsymbol{0}_{(d-1)p \times (d-1)p} \end{pmatrix}$.

Iterating the equation $n$ times yields

$$\Sigma_{\boldsymbol{X}} = \mathcal{A}^n\Sigma_{\boldsymbol{X}}(\mathcal{A}^\top)^n + \sum_{j=0}^{n-1}\mathcal{A}^j\Sigma_u(\mathcal{A}^\top)^j. \tag{34}$$

Taking spectral norms and using submultiplicativity,

$$\left\|\mathcal{A}^n\Sigma_{\boldsymbol{X}}(\mathcal{A}^\top)^n\right\|_2 \leq \|\mathcal{A}^n\|_2^2\|\Sigma_{\boldsymbol{X}}\|_2 \leq C_{\mathcal{A}}^2 r^{2n}\|\Sigma_{\boldsymbol{X}}\|_2 \xrightarrow[n\to\infty]{} 0,$$

so letting $n \to \infty$ in Equation (34) gives the convergent series representation

$$\Sigma_{\boldsymbol{X}} = \sum_{j=0}^{\infty}\mathcal{A}^j\Sigma_u(\mathcal{A}^\top)^j. \tag{35}$$

Moreover, the series converges absolutely in $\|\cdot\|_2$ since

$$\sum_{j=0}^{\infty}\left\|\mathcal{A}^j\Sigma_u(\mathcal{A}^\top)^j\right\|_2 \leq \|\Sigma_u\|_2\sum_{j=0}^{\infty}\|\mathcal{A}^j\|_2^2 \leq \|\Sigma_u\|_2\sum_{j=0}^{\infty}C_{\mathcal{A}}^2 r^{2j} < \infty.$$

For any unit vector $\boldsymbol{v} \in \mathbb{R}^{dp}$ with $\|\boldsymbol{v}\|_2 = 1$, Equation (35) implies

$$\boldsymbol{v}^\top\Sigma_{\boldsymbol{X}}\boldsymbol{v} = \sum_{j=0}^{\infty}\boldsymbol{v}^\top\mathcal{A}^j\Sigma_u(\mathcal{A}^\top)^j\boldsymbol{v} \leq \lambda_{\max}(\Sigma_u)\sum_{j=0}^{\infty}\left\|(\mathcal{A}^\top)^j\boldsymbol{v}\right\|_2^2 \leq \lambda_{\max}(\Sigma_u)\sum_{j=0}^{\infty}\|\mathcal{A}^j\|_2^2.$$

Using $\lambda_{\max}(\Sigma_u) = \lambda_{\max}(\Sigma_\varepsilon)$ and $\|\mathcal{A}^j\|_2^2 \leq C_{\mathcal{A}}^2 r^{2j}$, we obtain

$$\boldsymbol{v}^\top\Sigma_{\boldsymbol{X}}\boldsymbol{v} \leq \lambda_{\max}(\Sigma_\varepsilon)\sum_{j=0}^{\infty}C_{\mathcal{A}}^2 r^{2j} = \lambda_{\max}(\Sigma_\varepsilon)\frac{C_{\mathcal{A}}^2}{1-r^2}.$$

Taking the supremum over all $\|\boldsymbol{v}\|_2 = 1$ yields

$$\lambda_{\max}(\Sigma_{\boldsymbol{X}}) = \|\Sigma_{\boldsymbol{X}}\|_2 \leq \lambda_{\max}(\Sigma_\varepsilon)\frac{C_{\mathcal{A}}^2}{1-r^2}, \quad \text{and hence} \quad \kappa(\Sigma_{\boldsymbol{X}}) \leq \kappa(\Sigma_\varepsilon)\cdot\frac{C_{\mathcal{A}}^2}{1-r^2}.$$

**Consequences for $Q/\lambda_{\min}(\Sigma_X)$.** Recall $\Theta^* = (A_1, \ldots, A_d)^\top \in \mathbb{R}^{dp \times p}$ and the companion matrix $\mathcal{A}$. Let $J = (I_p, 0, \ldots, 0) \in \mathbb{R}^{p \times dp}$. Then the top block row of $\mathcal{A}$ is $J\mathcal{A} = (A_1, \ldots, A_d) \in \mathbb{R}^{p \times dp}$, so

$$\|\Theta^*\|_2 = \|(J\mathcal{A})^\top\|_2 = \|J\mathcal{A}\|_2 \leq \|J\|_2 \|\mathcal{A}\|_2 = \|\mathcal{A}\|_2.$$

Moreover, for each standard basis vector $e_i$, $\|\Theta^*_{:i}\|_2 = \|\Theta^* e_i\|_2 \leq \|\Theta^*\|_2$. Therefore,

$$\max_{1 \leq i \leq p} \|\Theta^*_{:i}\|_2^2 \leq \|\Theta^*\|_2^2 \leq \|\mathcal{A}\|_2^2 \leq (C_{\mathcal{A}} r)^2,$$

and

$$\frac{Q}{\lambda_{\min}(\Sigma_X)} = 16\pi\sqrt{\frac{b+1}{\tilde{c}}}\,\kappa(\Sigma_X)\Big(2 + \max_{1 \leq i \leq p} \|\Theta^*_{:i}\|_2^2\Big) \leq 16\pi\sqrt{\frac{b+1}{\tilde{c}}}\,\kappa(\Sigma_\varepsilon)\frac{C_{\mathcal{A}}^2}{1-r^2}\Big(2 + C_{\mathcal{A}}^2 r^2\Big).$$

## E. Mixing Conditions

In order to extend results from the i.i.d. setting to dependent data, mixing conditions are well established and widely used in the stochastic processes literature (Bradley, 2005). We quantify temporal dependence via standard mixing coefficients, defined below.

**Definition E.1** ($\alpha$-mixing coefficient). Let $X$ and $Y$ be random elements with associated $\sigma$-algebras $\sigma(X)$ and $\sigma(Y)$. Define

$$\alpha(X, Y) := \sup\Big\{\big|\mathbb{P}(A \cap B) - \mathbb{P}(A)\mathbb{P}(B)\big| : A \in \sigma(X),\ B \in \sigma(Y)\Big\}.$$

For a stationary stochastic process $(X_t)_{t \in \mathbb{Z}}$, define, for $\ell \geq 1$,

$$\alpha(\ell) := \alpha\big(\sigma(X_s : s \leq 0),\ \sigma(X_s : s \geq \ell)\big).$$

The process is called $\alpha$-mixing (strongly mixing) if $\alpha(\ell) \to 0$ as $\ell \to \infty$.

**Definition E.2** ($\rho$-mixing coefficient). Let $X$ and $Y$ be random elements. The $\rho$-mixing coefficient is defined by

$$\rho(X, Y) := \sup\Big\{\mathrm{corr}(f(X), g(Y)) : f \in \mathcal{L}^2(\sigma(X)),\ g \in \mathcal{L}^2(\sigma(Y))\Big\},$$

where $\mathcal{L}^2(\mathcal{C})$ denotes the space of square-integrable, $\mathcal{C}$-measurable random variables. For a stationary process $(X_t)_{t \in \mathbb{Z}}$, $\rho(\ell)$ is defined analogously by replacing $\sigma(X)$ and $\sigma(Y)$ with $\sigma(X_s : s \leq 0)$ and $\sigma(X_s : s \geq \ell)$.

*Fact* E.3. For any stationary Gaussian process, the $\alpha$- and $\rho$-mixing coefficients satisfy

$$\forall \ell \geq 1, \qquad \alpha(\ell) \leq \rho(\ell) \leq 2\pi\,\alpha(\ell).$$

## F. Experiment implementation details

**Simulation Data.** After generating sparse coefficient matrices $\{A_k\}_{k=1}^d \subset \mathbb{R}^{p \times p}$, we form the companion matrix $\mathcal{A}$ and compute its spectral radius $\rho(\mathcal{A})$. If $\rho(\mathcal{A}) \geq 1$, we uniformly rescale each $A_k$ by factor $0.95/\rho(\mathcal{A})$ so that the resulting process is stable. Given $\{A_k\}$, we generate $T_{\mathrm{burn}} + T$ observations of the VAR($d$) process

$$Z_t = \sum_{k=1}^d A_k Z_{t-k} + \varepsilon_t, \qquad \varepsilon_t \sim \mathcal{N}(0, \sigma_\varepsilon^2 I_p),$$

discard the first $T_{\mathrm{burn}}$ burn-in observations, and retain the subsequent $T$ vectors.

**Algorithms.** We run IHT/SP/CoSaMP with target sparsity level $s$ and a fixed step size $\eta = 0.9/\big(\lambda_{\max}(X^\top X)/n\big)$ on the standardized design. We terminate when the support is unchanged for 20 consecutive iterations or when the objective improvement is below $10^{-10}$.

Subspace Pursuit is invoked with expansion sparsity level $S = s$; that is, at each iteration, SP expands the candidate support with expansion size $S = s$ and then prunes back to $s$ nonzeros after a restricted least squares refit.

CoSaMP follows the same procedure and stopping criteria as SP, except that it uses a larger expansion size $S = 2s$.

We implement the LASSO using the `cvxpy` package (Diamond & Boyd, 2016). Specifically, we select the regularization parameter $\lambda$ via a ten-point grid search, choose the value that minimizes the validation MSE, and then refit the resulting model on the full training set.

As an upper benchmark, the Oracle LS solves least squares restricted to the true support.

## G. More Experimental Results

### G.1. Additional Simulation with Minimum Signal Strength

As discussed in Section 4.1, Figure 1 suggests that coefficient recovery deteriorates primarily on very small signals. To isolate the effect of signal heterogeneity, we consider an additional sparse VAR instance where we enforce a minimum signal strength by hard-thresholding the ground-truth coefficients at 0.01, while keeping the overall sparsity pattern unchanged.

*Table 4.* Results on a Gaussian sparse VAR($d$) instance with enforced minimum signal strength: $p = 1$, $d = 100$, $T = 10,000$, $\sigma_A = 0.25$, $\sigma_\varepsilon = 0.2$, $s_{\text{perlag}} = 1$, $s_{\text{active}} = 10$, and all nonzero coefficients satisfy $|\theta_j^\star| > 0.01$. Metrics are the same as in Section 4.1.

| Setting | Method | Runtime | Train MSE | Test MSE | $\ell_2$-error | Precision | Recall | Sparsity level |
|---|---|---|---|---|---|---|---|---|
| | Oracle LS | 0.00 | 0.0400 | 0.0402 | 0.008 | 1.00 | 1.00 | 10 |
| | LASSO | 0.05 | 0.0401 | 0.0404 | 0.017 | 0.50 | 1.00 | 20 |
| VAR | IHT$_{(s=10)}$ | 0.02 | 0.0400 | 0.0402 | 0.008 | 1.00 | 1.00 | 10 |
| | SP$_{(s=10)}$ | 0.02 | 0.0400 | 0.0402 | 0.008 | 1.00 | 1.00 | 10 |
| | CoSaMP$_{(s=10)}$ | 0.01 | 0.0400 | 0.0402 | 0.008 | 1.00 | 1.00 | 10 |

Table 4 shows that enforcing a minimum signal strength largely eliminates the ambiguity induced by small coefficients. Figure 4 provides a coefficient-level view. In this regime, all exact sparse methods (IHT/SP/CoSaMP) recover the full support with perfect precision and recall, matching the oracle estimator in both test MSE and $\ell_2$ estimation error. In contrast, although the LASSO attains comparable predictive performance and achieves full recall, it still substantially over-selects.

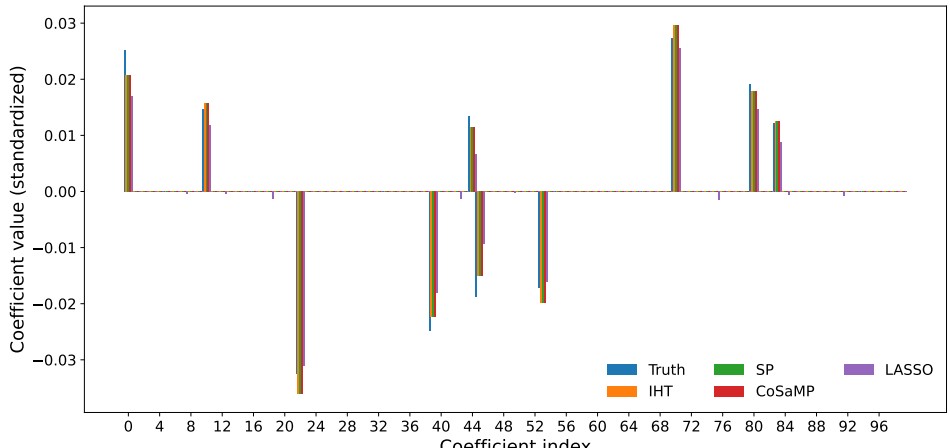

*Figure 4.* Coefficient estimates for the thresholded VAR instance in Table 4. Shown are the ground truth and estimates from IHT, SP, CoSaMP, and LASSO.

### G.2. Robustness to Sparsity Over-Specification

The main simulation tables use the true sparsity level $s = s^*$ for exact sparse methods. This is a stringent setting for support recovery because the algorithm is not allowed to include extra variables that may help compensate for finite-sample noise or correlated lag features. Our theory permits moderate over-specification of the target sparsity level, and Table 5 examines this effect on the multivariate sparse VAR instance from Table 2.

Moderate over-specification increases recall and can improve fitting, at the cost of lower precision and slightly higher runtime. Importantly, even the over-specified exact sparse runs remain faster than the LASSO baseline and keep the selected

Table 5. Effect of sparsity over-specification on the multivariate sparse Gaussian VAR($d$) experiment in Table 2.

| Method | Runtime | Train MSE | Test MSE | $\ell_2$-error | Precision | Recall | Sparsity |
|---|---|---|---|---|---|---|---|
| LASSO | 0.83 | 0.0397 | 0.0403 | 0.013 | 0.21 | 0.90 | 85 |
| IHT$_{(s=20)}$ | 0.20 | 0.0397 | 0.0402 | 0.010 | 0.80 | 0.80 | 20 |
| IHT$_{(s=25)}$ | 0.23 | 0.0397 | 0.0429 | 0.020 | 0.36 | 0.90 | 25 |
| IHT$_{(s=40)}$ | 0.39 | 0.0395 | 0.0429 | 0.024 | 0.25 | 1.00 | 40 |

model substantially sparser. This supports the use of exact sparse methods when the exact sparsity level is not known a priori but a moderately conservative upper bound is available.

### G.3. Sensitivity to VAR Conditioning

We further examine how performance changes as the VAR process becomes more ill-conditioned. Based on the simulation setting in Table 1 of Section 4.1, we conduct an additional sensitivity study by varying the VAR stability margin. After generating the coefficient matrices $\{A_k\}_{k=1}^d$, we form the companion matrix $\mathcal{A}$ and rescale the coefficients so that $\rho(\mathcal{A}) \leq \gamma < 1$. The original simulation in Table 1 uses $\gamma = 0.95$ to ensure stability. Increasing $\gamma$ moves the process closer to the stability boundary and typically yields a more ill-conditioned lagged design covariance $\Sigma_X$.

For each generated instance, we report both the population condition number $\kappa(\Sigma_X) = \lambda_{\max}(\Sigma_X)/\lambda_{\min}(\Sigma_X)$, computed from the stationary companion-form covariance, and its finite-sample analogue $\kappa(\widehat{\Sigma}_X)$, computed from the generated lag vectors. The results are summarized in Table 6.

Table 6. Condition-number sensitivity for the univariate sparse Gaussian VAR($d$) experiment. Larger $\gamma$ moves the process closer to the stability boundary.

| $\gamma$ | $\kappa(\Sigma_X)$ | $\kappa(\widehat{\Sigma}_X)$ | Method | Runtime | Train MSE | Test MSE | $\ell_2$-error | Precision | Recall | Sparsity |
|---|---|---|---|---|---|---|---|---|---|---|
| 0.950 | 1.10 | 1.58 | LASSO | 0.10 | 0.0412 | 0.0399 | 0.015 | 0.28 | 0.90 | 32 |
| | | | IHT$_{(s=10)}$ | 0.02 | 0.0414 | 0.0397 | 0.029 | 0.90 | 0.90 | 10 |
| | | | SP$_{(s=10)}$ | 0.02 | 0.0413 | 0.0397 | 0.010 | 0.90 | 0.90 | 10 |
| | | | CoSaMP$_{(s=10)}$ | 0.01 | 0.0413 | 0.0397 | 0.010 | 0.90 | 0.90 | 10 |
| 0.990 | 4.85 | 4.69 | LASSO | 0.19 | 0.0415 | 0.0399 | 0.218 | 0.26 | 0.80 | 31 |
| | | | IHT$_{(s=10)}$ | 0.04 | 0.0413 | 0.0398 | 0.213 | 0.90 | 0.90 | 10 |
| | | | SP$_{(s=10)}$ | 0.02 | 0.0415 | 0.0398 | 0.212 | 0.80 | 0.80 | 10 |
| | | | CoSaMP$_{(s=10)}$ | 0.02 | 0.0415 | 0.0398 | 0.212 | 0.80 | 0.80 | 10 |
| 0.999 | 43.53 | 47.31 | LASSO | 0.22 | 0.0414 | 0.0398 | 0.326 | 0.24 | 0.90 | 38 |
| | | | IHT$_{(s=10)}$ | 0.09 | 0.0413 | 0.0398 | 0.323 | 0.70 | 0.70 | 10 |
| | | | SP$_{(s=10)}$ | 0.04 | 0.0413 | 0.0398 | 0.319 | 0.80 | 0.80 | 10 |
| | | | CoSaMP$_{(s=10)}$ | 0.03 | 0.0414 | 0.0397 | 0.320 | 0.70 | 0.70 | 10 |

The results show a clear degradation trend as the condition number increases: both LASSO and exact sparse methods exhibit larger estimation error and some deterioration in support recovery, consistent with the dependence of the theory on conditioning. At the same time, prediction MSE remains relatively stable, suggesting a degree of empirical robustness for prediction. Runtime also increases as the process becomes more ill-conditioned, but the exact sparse methods continue to retain a computational advantage over LASSO.

### G.4. Experiments on Chicago Ridesharing Data

We validate our findings on the Chicago ridesharing dataset using the same estimation pipeline as in Section 4.2.

Table 7 summarizes runtime and prediction MSE. Among methods producing a 7-sparse model, SP and CoSaMP achieve the lowest test MSE while being substantially faster than the LASSO. The LASSO attains a competitive test error but is notably slower. IHT again exhibits a clear sparsity–fit trade-off: small $s$ underfits the data, while increasing $s$ improves prediction and approaches the performance of the best sparse estimators.

Beyond prediction, the selected lag sets highlight the interpretability advantage of exact sparse estimation. The LASSO produces a 7-sparse estimate with selected lags $\{1, 23, 24, 143, 167, 168, 172\}$. In contrast, IHT$_{(s=7)}$ selects

*Table 7.* Chicago ridesharing $\mathrm{AR}(d)$ estimation results.

| Method | Runtime | Train MSE | Test MSE | Sparsity |
|---|---|---|---|---|
| LASSO | 3.74 | 0.029 | 0.023 | 7 |
| IHT$_{(s=2)}$ | 0.30 | 0.278 | 0.212 | 2 |
| IHT$_{(s=3)}$ | 0.32 | 0.192 | 0.149 | 3 |
| IHT$_{(s=5)}$ | 0.35 | 0.092 | 0.073 | 5 |
| IHT$_{(s=7)}$ | 0.39 | 0.065 | 0.053 | 7 |
| SP$_{(s=7)}$ | 0.04 | 0.026 | 0.021 | 7 |
| CoSaMP$_{(s=7)}$ | 0.03 | 0.026 | 0.021 | 7 |

$\{1, 24, 144, 167, 168, 169, 192\}$, which aligns more closely with the expected periodic structure of the series.

Figure 5 provides a coefficient-selection trajectory for IHT as the sparsity level increases. Consistent with the NYC results, the objective value decreases monotonically with $s$, while the marginal improvement diminishes as weaker effects are added. Overall, the Chicago experiments corroborate the main conclusion: exact sparse solvers remain computationally efficient and tend to yield lag selections that more cleanly expose the underlying periodic structure than the LASSO.

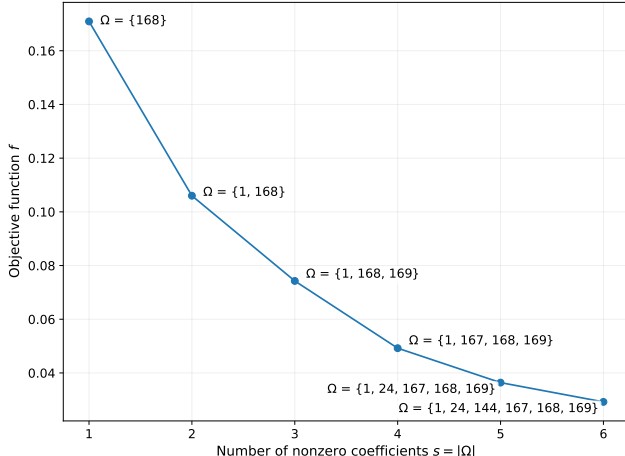

*Figure 5.* Illustration of the IHT estimations on the Chicago ridesharing trips at different sparsity levels. The support set and the number of nonzero coefficients are denoted as $\Omega$ and $|\Omega|$, respectively.

We further perform an oracle least squares (Oracle LS) refit restricted to each method's selected support set and report the resulting training and test MSE. Table 8 shows that, after controlling for sparsity (exact sparse solvers use $s = 7$, matching the sparsity level of the LASSO solution), the oracle-refit models achieve very similar predictive performance, with only negligible differences in MSE across methods. This suggests that the main distinction among these approaches is not prediction accuracy under an oracle refit, but rather the structure of the recovered support. In particular, the exact sparse methods (IHT/SP/CoSaMP) tend to select lag sets that are more interpretable and more consistent with the expected periodic patterns (e.g., daily and weekly lags and their local neighborhoods), whereas the LASSO selects a denser and less structured set of lags.

*Table 8.* Chicago ridesharing $\mathrm{AR}(d)$ estimation results: oracle least squares refit on the support set selected by each method (exact sparse solvers use $s = 7$, matching the sparsity level of the LASSO solution).

| Setting | Method | Support set | Runtime | Train MSE | Test MSE |
|---|---|---|---|---|---|
| Oracle LS | LASSO | $\{1, 23, 24, 143, 167, 168, 172\}$ | 0.02 | 0.028 | 0.022 |
| | IHT$_{(s=7)}$ | $\{1, 24, 144, 167, 168, 169, 192\}$ | 0.02 | 0.028 | 0.022 |
| | SP$_{(s=7)}$ | $\{1, 24, 167, 168, 169, 171, 192\}$ | 0.02 | 0.026 | 0.021 |
| | CoSaMP$_{(s=7)}$ | $\{1, 2, 24, 144, 167, 168, 170\}$ | 0.02 | 0.026 | 0.021 |

### G.5. Runtime Comparison Across Implementations

Since runtime is a central empirical claim of this paper, we additionally compare the exact sparse methods against optimized convex sparse-regression baselines on the NYC ridesharing experiment in Section 4.2. The main text reports a standard LASSO baseline implemented with `cvxpy`, which provides a reliable convex reference but may not reflect the speed of highly optimized solvers. To make the runtime comparison less dependent on a particular implementation, we also fit LASSO and elastic net using warm-started pathwise coordinate descent, while keeping the same chronological train/validation/test split, preprocessing, AR($d$) design, and validation protocol as in the main experiment. For elastic net, we use the standard parameterization with mixing parameter $0.9$.

*Table 9.* Optimized convex baselines for NYC ridesharing AR($d$) estimation. All methods use the same split, preprocessing, and validation protocol as in Section 4.2. The original results from the main experiment are included for reference.

| Method | Runtime | Train MSE | Test MSE | Sparsity |
|---|---|---|---|---|
| LASSO-CD | 0.51 | 0.031 | 0.015 | 10 |
| ElasticNet-CD | 1.06 | 0.031 | 0.015 | 10 |
| LASSO | 1.92 | 0.031 | 0.015 | 10 |
| IHT$_{(s=10)}$ | 0.32 | 0.062 | 0.030 | 10 |
| SP$_{(s=10)}$ | 0.04 | 0.028 | 0.012 | 10 |
| CoSaMP$_{(s=10)}$ | 0.06 | 0.028 | 0.013 | 10 |

Table 9 shows that coordinate descent substantially reduces the runtime of the convex baselines while preserving the same predictive accuracy and sparsity level. Nevertheless, the exact sparse methods remain computationally competitive: IHT$_{(s=10)}$ is faster than the optimized LASSO baseline, while SP$_{(s=10)}$ and CoSaMP$_{(s=10)}$ are an order of magnitude faster and also achieve lower test MSE in this experiment. These results indicate that the computational advantage of exact sparse methods is not solely an artifact of using a generic convex solver for LASSO.

