# OpenReview forum: "Sparse Regression with $\ell_0$ Constraints for $\alpha$-Mixing Time Series: Algorithms and Guarantees"
_ICML.cc/2026/Conference — ICML 2026 regular_

### Official Review · Reviewer_LoLp · 2026-02-23

**Soundness:** 3
**Presentation:** 3
**Significance:** 3
**Originality:** 3
**Overall Recommendation:** 5
**Confidence:** 4

**Summary:**

**Summary:**  The paper studies the theoretical performance of well known IHT algorithm when the data are generated from an $\alpha$-mixing time series. The results of the paper are not surprising but presents novel set of IHT convergence results. The simulation experiments show that IHT actually enjoys a better performance both in terms of statistical guarantee and computation.

**Compliance With Llm Reviewing Policy:**

Affirmed.

**Final Justification:**

My concerns have been adequately addressed. I have raised my score.

**Key Questions For Authors:**

**Questions:**
 - Do the convergence results hold if the data are not coming from a Gaussian process?
 - To what degree $\alpha$-mixing assumption be relaxed?
 - Does any minimax result hold in such cases?

**Limitations:**

yes

**Strengths And Weaknesses:**

**Strength:**
 - The paper is well motivated and well written with clear explanations.
 - The theoretical contributions are novel and broaden the knowledge of high-dimensional statistics.


**Weakness:**
  - The theoretical results seem to rely on the Gaussianity of the time series data. This is overly restricting as in many cases the data could be sub-Gaussian or even heavy-tailed.
 - The convergence rate in Thm 3.3 depends on $S_\alpha(T)$, and only goes to 0 if it is $o(\sqrt{T})$. This means on an average $\alpha(\ell) = o(T^{1-/2})$. This entails that $X_t$’s are almost independent.
 - There are no minimax results presented in the paper. Therefore, the sharpness of the upperbound could be loose.

**References:** In the introduction authors claim that analysis of LASSO or IHT has only been done in i.i.d. settings. However, both have been analyzed in a linear bandit setting, which involves highly dependent data. For example, [1] analyzes LASSO (there are more works on LASSO and bandit) and [2] analyzes IHT under bandit settings. It is recommended to cite these bandit works in the introduction section.

[1] Li, K., Yang, Y., & Narisetty, N. N. (2021). Regret lower bound and optimal algorithm for high-dimensional contextual linear bandit. Electronic Journal of Statistics, 15(2), 5652-5695.

[2] Roy, S., Chakraborty, S., & Basu, D. (2025, April). FLIPHAT: Joint Differential Privacy for High Dimensional Linear Bandits. In International Conference on Artificial Intelligence and Statistics (pp. 2359-2367). PMLR.

---

> ### Author Rebuttal · Authors · 2026-03-30
>
> We thank the reviewer for providing positive feedback and insightful comments. Below we address your specific points.
>
> ---
>
> ### **Response to Weakness 1 (also Question 1)**
>
> We thank the reviewer for this important question. We agree that the Gaussianity assumption is restrictive. Our view is that extending the results to broader sub-Gaussian classes is plausible, but it involves a clear **tail–dependence trade-off**.
>
> In particular, one can hope to relax Gaussianity to sub-Gaussian (or even sub-Weibull) tails by imposing a stronger dependence condition, e.g., moving from $\\alpha$-mixing to $\\beta$-mixing. This is consistent with Wong et al. (2020), who establish Lasso error bounds for both $\\alpha$-mixing Gaussian processes and $\\beta$-mixing sub-Weibull processes, showing that broader tails can be handled at the price of stronger dependence assumptions.
>
> For our paper, the main part of the analysis that would require substantial modification is the **concentration step** used to establish the RSC/RSS properties. Concretely, the Gaussian quadratic-form concentration steps based on Hanson–Wright-type arguments would need to be replaced by concentration inequalities for dependent sub-Gaussian sequences. By contrast, once suitable RSC/RSS conditions are established, the analyses of IHT, SP, and CoSaMP are largely deterministic and should carry over with insignificant modifications to constants and sample-size requirements.
>
> We will expand the discussion in the camera-ready version and highlight such an extension as an important direction for future work.
>
> ---
>
> ### **Response to Weakness 2 (also Question 2)**
>
> The concern is whether the requirement $S_\alpha(T) = o(\sqrt{T})$ makes the process “almost independent,” and, more broadly, how far the $\alpha$-mixing assumption can be relaxed.
>
> First, we would like to clarify that our result covers a broad class of short-range dependent processes, rather than only “almost independent” ones. $\alpha$-mixing is a standard and widely used notion of dependence in time-series analysis; intuitively, it requires that the dependence between events far apart in time decays as the lag increases. Among classical mixing conditions, it is already a relatively weak one; stronger notions include, for example, $\beta$-mixing, which requires the dependence between the distant past and future to vanish in a stronger, total-variation sense. We adopt $\alpha$-mixing because it is general enough to capture many dependent processes while still strong enough to support the nonasymptotic concentration arguments needed for our RSC/RSS analysis.
>
> Regarding possible relaxations, the key point is that $\alpha$-mixing enters the proofs mainly through the concentration step. That said, if one moves to substantially weaker dependence regimes, especially long-range dependence where $S_\alpha(T)$ grows faster than $\sqrt{T}$, then the effective sample size deteriorates and the current error bound no longer vanishes. In such settings, one should generally expect slower rates, stronger structural assumptions, or new proof techniques. Therefore, we view extensions beyond the current short-range dependence regime as an interesting direction for future work.
>
> ---
>
> ### **Response to Weakness 3 (also Question 3)**
>
> Please refer to our response to Reviewer 3bei.
>
> ---
>
> ### **Response to References**
>
> We thank the reviewer for pointing out these relevant bandit references. To clarify, our claim was not that LASSO has only been analyzed in i.i.d. settings. In fact, there is a substantial literature on LASSO-type methods beyond the i.i.d. regime, including sequential and adaptive settings. Rather, our intended claim concerned IHT-style sparse recovery guarantees, which, to the best of our knowledge, remain much less developed outside the classical i.i.d. setting.
>
> The suggested papers are indeed relevant and helpful to cite. Li, Yang, and Narisetty (2021) study high-dimensional contextual linear bandits and develop a sparse $\\ell_1$-based confidence-set method in an adaptively collected data setting, while Roy, Chakraborty, and Basu (2025) propose FLIPHAT for high-dimensional linear bandits with joint differential privacy, where a variant of noisy IHT is used as a sparse regression oracle and its estimation error is analyzed as part of the regret analysis. At the same time, these works address a different problem class from ours: their primary objective is regret minimization under adaptive data collection, whereas our paper studies parameter estimation and recovery guarantees for exact-sparse algorithms under temporally dependent stationary time series. We agree, however, that they are important related works showing that sparse estimators can be analyzed under highly dependent sequential data, and we will add and discuss them in the revised paper.

---

> > ### Author Rebuttal · Reviewer_LoLp · 2026-04-01
> >
> > My concerns have been adequately addressed.

---

> > > ### Author Response · Authors · 2026-04-02
> > >
> > > Thank you very much for taking the time to revisit our submission. We are grateful that our efforts to address your concerns have been recognized, and we sincerely appreciate the increased score. Your thoughtful review and feedback mean a great deal to us and have helped improve the quality of our paper.

---

### Official Review · Reviewer_j5nB · 2026-03-03

**Soundness:** 3
**Presentation:** 4
**Significance:** 3
**Originality:** 3
**Overall Recommendation:** 4
**Confidence:** 3

**Summary:**

This paper establishes finite-sample and computational guarantees for $\ell_0$-constrained (exact) sparse regression when the design comes from a temporally dependent time series, rather than i.i.d. samples. Under stationarity + Gaussianity + $\alpha$-mixing assumptions, the authors prove the RSC/RSS property of the empirical objective, and then show geometric convergence of the exact sparse estimation algorithms IHT and greedy pursuit. The theory is specialized to Gaussian VAR, relating the key quantities to stability and innovation covariance conditioning.

**Compliance With Llm Reviewing Policy:**

Affirmed.

**Final Justification:**

The rebuttal largely addresses my concerns. I will keep the positive score.

**Key Questions For Authors:**

1. Theory–practice alignment: In Theorem 3.3, the guarantees require $s \ge C_1 \kappa(\Sigma_X)^2 s^\*$ (and additional conditions for greedy pursuit). In experiments, you set $s=s^*$ (simulations) or small fixed $s$ (real data). Can you clarify whether the theoretical conditions are expected to hold in your experimental regimes? If not, can you provide empirical evidence (or discussion) about robustness when $s$ violates the theorem’s sufficient conditions?

2. LASSO sparsity and tuning protocol (real data): How exactly is the NYC LASSO configured to produce a “10-sparse” estimate in Table 3? Is $\lambda$ chosen to match sparsity, or is there post-hoc thresholding to the top-10 coefficients? Please specify, and ideally report sensitivity to tuning grids and alternative selection criteria.

3. Baseline fairness: Have you compared against a fast coordinate-descent LASSO (or elastic net) implementation and/or structured VAR regularizers? Since runtime is a central claim, an implementation-agnostic comparison would be important.

4. Dependence strength stress tests: Can you include experiments that vary dependence/mixing strength (e.g., by changing VAR stability margin (r) or correlation structure) and verify whether empirical error/runtime trends track the theory’s dependence on $S_\alpha(T)$ and conditioning?

**Limitations:**

yes

**Strengths And Weaknesses:**

Strengths

1. Addresses a real and well-motivated theory gap (temporal dependence + $\ell_0$ exact sparsity). The paper clearly articulates that many $\ell_0$ exact-sparse methods are widely used but their guarantees are largely limited to i.i.d. designs, and that this mismatch matters for time series applications.

2. Nonasymptotic results with explicit dependence on mixing and conditioning. The main bounds explicitly expose how estimation depends on $S_\alpha(T), \lambda_{\min}(\Sigma_X), \lambda_{\max}(\Sigma_X)$, and related quantities.   The VAR specialization further interprets these in terms of stability radius and innovation covariance conditioning.

3. Empirical results align with the interpretability motivation. In both NYC and Chicago ridesharing, the selected lags from exact sparse methods match intuitive daily/weekly structures (e.g., 24 and 168), and runtime differences are large under their implementations.

Weaknesses

1. Strong and somewhat restrictive assumptions (Gaussianity + $\alpha$-mixing + stationarity). The theory assumes strictly stationary Gaussian $\alpha$-mixing processes.  While the authors mention heavy-tailed/nonlinear extensions as future work, the current results may not directly cover many realistic time series with non-Gaussian innovations, regime shifts, or long-memory dependence.

2. Potential mismatch between theoretical conditions and experimental configurations. Theorem 3.3 requires sparsity inflation such as $s \ge C_1 \kappa(\Sigma_X)^2 s^\*$ for IHT (and additional constraints for greedy pursuit). Yet experiments run IHT/SP/CoSaMP with $s$ set to the true sparsity $s^*$ (oracle knowledge) in simulations, and use fixed small target sparsity in real data. This makes it unclear whether the empirical successes occur in regimes covered by the stated guarantees, or whether performance hinges on behavior outside the proven conditions.

3. Practical implementability of step size / constants is unclear. Theorem 3.3 specifies a step size depending on $\lambda_{\max}(\Sigma_X)$ and $\lambda_{\min}(\Sigma_X)$, which are typically unknown. The experimental implementation uses a different heuristic step size based on the empirical design. A discussion on how to estimate/choose theory-compliant step sizes (or how sensitive the guarantees are to step-size misspecification) would strengthen the paper.

4. Ambiguity in the real-data LASSO sparsity reporting. In Table 3 (NYC) the LASSO solution is reported as exactly 10-sparse. Standard LASSO with a validation-chosen $\lambda$ does not necessarily produce exactly a target sparsity unless one explicitly selects $\lambda$ to hit that sparsity or post-thresholds coefficients. The procedure should be clarified because it affects fairness and interpretation.

---

> ### Author Rebuttal · Authors · 2026-03-30
>
> Thanks for your detailed comments. We greatly appreciate your time and positive feedback.
>
> ---
>
> ### **Response to Weakness 1**
>
> Please refer to our response to Weakness 1 for Reviewer LoLp.
>
> ---
>
> ### **Response to Weakness 2 (also Q1)**
>
> Please see our response to Weakness 3 for Reviewer ewyY.
>
> ---
>
> ### **Response to Weakness 3**
>
> We thank the reviewer for this constructive suggestion and apologize for the lack of clarity.
>
> We agree that the step size in Theorem 3.3 depends on $\lambda_{\max}(\Sigma_X)$ and $\lambda_{\min}(\Sigma_X)$, which are typically unknown in practice. For this reason, in the experiments we use a data-driven approximation based on the empirical design matrix. As stated in Appendix F (Experiment Implementation Details, page 24), *“we run IHT/SP/CoSaMP with a fixed step size $\eta = 0.9 \big/ \bigl(\lambda_{\max}(X^\top X)/n\bigr)$.”*
>
> This choice is natural since $X^\top X/n$ is a principled empirical proxy for $\Sigma_X$, and $\lambda_{\max}(X^\top X/n)$ is the empirical smoothness constant of the objective. Therefore, choosing $\eta$ slightly below its reciprocal yields a conservative and standard step size for stable descent. In this sense, the experimental rule can be viewed as a practical plug-in approximation to the theory-guided choice.
>
> We agree that this connection should be stated more explicitly, and we will clarify it in the revised manuscript.
>
> ---
>
> ### **Response to Weakness 4 (also Q2)**
>
> We apologize for the confusion caused by Table 3. We would like to clarify that we did **NOT** tune LASSO to enforce an exactly 10-sparse solution, nor did we apply any post-hoc thresholding (e.g., keeping only the top-10 coefficients).
>
> Instead, in all experiments we selected the regularization parameter $\lambda$ for LASSO via a standard validation procedure. As stated in Appendix F (Experiment Implementation Details, page 24), *“we select the regularization parameter $\lambda$ via a ten-point grid search, choose the value that minimizes the validation MSE, and then refit the resulting model on the full training set.”* No additional tuning was performed to target a specific sparsity level. Under this standard procedure, the LASSO solution on the NYC dataset happened to contain exactly 10 nonzero coefficients.
>
> For a fair comparison, we then set the sparsity level of IHT, SP, and CoSaMP to $s=10$, so that the exact-sparse methods were compared against a model of comparable effective complexity. We did not choose $\lambda$ to match the sparsity level of the exact-sparse methods; rather, after observing that validation-tuned LASSO yielded a 10-sparse solution, we included $s=10$ for IHT/SP/CoSaMP as a natural comparison point. Table 3 also reports IHT with $s=2,3,5$, giving a broader picture across multiple sparsity levels.
>
> The same procedure was also used in Appendix G.2 (Experiments on Chicago Ridesharing Data). In Table 5 (page 25), LASSO yields a 7-sparse solution, which further confirms that we did not impose any extra sparsity matching or post-processing; the resulting sparsity simply depends on the validation-selected $\lambda$ for each dataset.
>
> We agree this should be made clearer, and we will revise the manuscript accordingly.
>
> ---
>
> ### **Response to Q3**
>
> Since runtime is a central claim of the paper, we agree that a more implementation-agnostic comparison is important. In the current submission, the LASSO baseline is implemented using the **cvxpy** package. While this provides a standard and reliable convex baseline, it may not fully reflect the speed of highly optimized sparse regression solvers, such as dedicated pathwise coordinate-descent methods.
>
> To address this concern, we additionally compare against **fast coordinate-descent LASSO and elastic net**. Concretely, both baselines are fit using standard warm-started pathwise coordinate descent, while keeping the same validation protocol as in the main experiments. For elastic net, we use the standard parametrization with mixing parameter set to $0.9$.
>
> Using the real-data experiment in Section 4.2 (page 8) as an example, we obtain the following supplementary results for Table 3 on the NYC ridesharing dataset:
>
> |Method|Runtime|Train MSE|Test MSE|Sparsity|
> |-|-:|-:|-:|-:|
> |LASSO|1.92|0.031|0.015|10|
> |LASSO-CD|0.51|0.031|0.015|10|
> |ElasticNet-CD|1.06|0.031|0.015|10|
>
> These additional results show that the exact-sparse methods remain faster even relative to optimized coordinate-descent baselines.
>
> ---
>
> ### **Response to Q4**
>
> Thank you for this constructive question. Our theory predicts that performance should deteriorate as dependence becomes stronger. We agree that this should be empirically verified. In the revision, we will add sensitivity experiments that vary dependence strength by changing the VAR stability margin and correlation structure, and examine whether the resulting error and runtime trends align with the theory. Due to the length limit, we omit the results here and will include them in the revised paper.

---

> > ### Author Rebuttal · Reviewer_j5nB · 2026-04-02
> >
> > The authors provided a detailed clarification on how the conditions required by Theorem 3.3 relate to the specific experimental settings, which addresses the concern I raised. I appreciate this explanation. That said, due to space constraints, the rebuttal does not include additional experiments examining how varying the mixing strength would affect empirical performance. I would encourage the authors to include such results and a brief analysis in the camera-ready version, as it would help further validate the theoretical dependence on temporal dependence/mixing.

---

> > > ### Author Response · Authors · 2026-04-03
> > >
> > > Thank you very much for your positive feedback and for your acknowledgment of our rebuttal. We sincerely appreciate your constructive suggestion to include additional experiments varying the mixing strength in order to further validate the theoretical dependence on mixing conditions. We confirm that we will include these results and analysis in the camera-ready version.

---

### Official Review · Reviewer_ewyY · 2026-03-13

**Soundness:** 3
**Presentation:** 3
**Significance:** 3
**Originality:** 3
**Overall Recommendation:** 4
**Confidence:** 4

**Summary:**

This paper studies $l_0$-constrained sparse linear regression for high-dimensional time series under stationary Gaussian $\alpha$-mixing assumptions. Existing analyses of exact sparse methods such as iterative hard thresholding (IHT) and greedy pursuit typically assume independent designs; this paper extends such guarantees to temporally dependent settings. The authors establish restricted strong convexity (RSC) and smoothness of the empirical least-squares loss with explicit constants depending on the population covariance spectrum and the mixing-sum quantity.

The paper then derives sample and iteration complexity guarantees for IHT and greedy pursuit. Under suitable regularity conditions, the algorithms converge linearly to a statistical neighborhood of the true parameter, with an error rate the same as in the classical sparse regression when the mixing coefficients are summable. Empirical evidence demonstrates competitive prediction performance with improved support interpretability relative to LASSO.

**Compliance With Llm Reviewing Policy:**

Affirmed.

**Final Justification:**

The paper is technically sound within its assumptions and provides a meaningful extension of ℓ₀-based methods to sparse VAR models under dependence, with competitive empirical performance.

The rebuttal is helpful and addresses several of my concerns. In particular, the discussion of Gaussianity clarifies the scope of the theory, and the additional experiments on sparsity and conditioning improve empirical support. While the reliance on joint normality remains and robustness under more challenging regimes is not fully explored, these limitations do not undermine the overall contribution.

Overall, I find the work to be a solid contribution with reasonable practical relevance. The rebuttal strengthens the paper, and I am comfortable recommending a weak accept.

**Key Questions For Authors:**

1. Do the authors expect the main results to extend to broader classes of processes (e.g., sub-Gaussian or finite-moment $\alpha$-mixing processes)? If so, which parts of the analysis would require significant modification?
2. How should practitioners choose the sparsity level in practice?
3. Do the authors expect that inference procedures (e.g., asymptotic distributions or confidence intervals) could be developed for $l_0$-constrained estimators? If so, what technical challenges would arise compared to existing inference approaches for $l_1$-based methods such as LASSO?

**Limitations:**

The paper discusses several key limitations of the proposed framework, including strong modeling assumptions such as Gaussianity, stationarity, and $\alpha$-mixing dependence. The paper does not have direct negative societal impacts beyond the standard considerations associated with statistical modeling.

**Strengths And Weaknesses:**

Strengths

1. The authors developed a unified theoretical framework for $L_0$ methods by establishing the RSC, which gives iteration and sample complexity bounds for IHT and greedy pursuit.
2. The derived error rates match classical sparse regression rates up to $\alpha$-mixing coefficients $S_\alpha (T)$, suggesting that dependence does not fundamentally worsen sample complexity under summable mixing.

Weaknesses

1. The most severe weakness is that all theoretical results rely on joint normality assumption, which is crucial for proving concentration via Hanson–Wright inequalities and mixing equivalences. Extensions to sub-Gaussian or heavy-tailed processes are not addressed and may be nontrivial.
2. The theoretical guarantees of the proposed method depend on the condition number $\kappa (\Sigma_X)$. The author should include some discussion on the method's performance for cases where this condition number can be large.
3. The proposed methods require specifying a working sparsity level $s$. While the theory allows $s\geq C\kappa^2 s^*$, the experiments use the true sparsity. The author should demonstrate robustness to moderate over-specification of sparsity e.g. $s=2s^*$ or to discuss practical tuning strategies.

---

> ### Author Rebuttal · Authors · 2026-03-30
>
> We sincerely thank the reviewer for the valuable time, positive feedback, and constructive suggestions.
>
> ---
>
> ### **Response to Weakness 1 (also Q1)**
>
> Please see our response to Weakness 1 for Reviewer LoLp.
>
> ---
>
> ### **Response to Weakness 2**
>
> If the population covariance $\Sigma_X$ is ill-conditioned, our guarantees weaken as they depend on $\kappa(\Sigma_X)$. A larger condition number leads to poorer effective curvature, slower contraction of IHT/SP/CoSaMP, and weaker finite-sample error bounds. This is not specific to our method; it reflects the inherent difficulty of sparse regression under poor conditioning.
>
> To further assess robustness, we add a simulation study with varying degrees of conditioning. Specifically, in our current setup we form the companion matrix, compute its spectral radius $\rho(\mathcal A)$, and, if needed, uniformly rescale the coefficient matrices by a factor such as $0.95/\rho(\mathcal A)$ to ensure stability (Appendix F, page 24). Changing the number $0.95$ toward $1$ moves the process closer to the stability boundary and typically produces more ill-conditioned design covariances. Our experiments suggest that the empirical performance remains robust under such variations. Due to the rebuttal length limit, we omit the full results here and will discuss them in the revised paper.
>
> ---
>
> ### **Response to Weakness 3 (also Q2)**
>
> First, regarding the experimental setup, we would like to clarify that using the true sparsity $s=s^\ast$ in the submission is, in fact, the most stringent setting for exact-sparse methods. In this sense, the empirical fact that IHT-style methods achieve competitive estimation and prediction performance, together with strong support recovery when using the true sparsity, provides even stronger evidence of the effectiveness of those methods. Our theory allows for moderate over-specification, e.g., $s \ge C \kappa^2 s^\ast$, and in additional experiments, we observed that moderate over-specification still yields competitive predictive performance and often improves fitting accuracy and support recovery, especially in terms of recall. The main cost of over-specification is increased computation; however, in our experiments, the runtime remains below that of LASSO. We will add these extra results in the revision to explicitly demonstrate robustness to moderate over-specification.
>
> **Additional experiment supplementing Table 2 in the submission**
> *(Simulation results on a multivariate sparse Gaussian VAR model)*
>
> |         Method | Runtime | Train MSE | Test MSE | $\ell_2$-error | Precision | Recall | Sparsity level |
> | -------------: | ------: | --------: | -------: | -------------: | --------: | -----: | -------------: |
> |          LASSO |    0.83 |    0.0397 |   0.0403 |          0.013 |      0.21 |   0.90 |             85 |
> | IHT$_{(s=20)}$ |    0.20 |    0.0397 |   0.0402 |          0.010 |      0.80 |   0.80 |             20 |
> | IHT$_{(s=25)}$ |    0.23 |    0.0397 |   0.0429 |          0.020 |      0.36 |   0.90 |             25 |
> | IHT$_{(s=40)}$ |    0.39 |    0.0395 |   0.0429 |          0.024 |      0.25 |   1.00 |             40 |
>
> Second, in practice, we view $s$ as a working model-size parameter rather than an unknown quantity that must be specified exactly. Its choice depends on the practitioner’s goal: a smaller $s$ favors interpretability and parsimonious support selection, while a moderately larger $s$ often improves prediction or support recall. A simple strategy is to start with a value somewhat larger than the expected number of active factors and tune it using validation performance or model stability. For example, if one aims to identify about 5 key factors, a reasonable starting point is $s=10$, followed by adjustment based on the selected support and validation error.
>
> ---
>
> ### **Response to Question 3**
>
> For $\ell_1$-based methods such as LASSO, inference is mainly developed along two lines. One is de-biased / desparsified inference, which corrects the bias of the LASSO estimator using an approximate inverse Gram matrix to obtain an asymptotically linear form, leading to asymptotic normality and confidence intervals for low-dimensional parameters. The other is selective inference, which conditions on the model-selection event and enables valid post-selection testing or interval construction. Both lines have been extensively studied for LASSO and related methods.
>
> By contrast, we do not currently know how to develop a comparable inference theory for the $\ell_0$-constrained estimators considered in our paper, and we therefore view this as an important direction for future work. The main difficulty is that existing $\ell_1$-based inference methods rely heavily on tools specific to convex regularization, such as the KKT/subgradient characterization used in de-biasing arguments. These tools are far less directly available for $\ell_0$-constrained estimators such as IHT, SP, and CoSaMP, which are inherently nonconvex and algorithmic.

---

> > ### Author Rebuttal · Reviewer_ewyY · 2026-04-02
> >
> > The rebuttal provides useful clarifications and partially addresses several of my concerns. In particular, the authors offer a clear discussion of the role of the Gaussianity assumption and identify the concentration step underlying the RSC/RSS conditions as the main technical bottleneck. While this improves the understanding of the theoretical scope, the extension beyond the Gaussian setting remains at a high level and is not developed in the current work.
> >
> > Regarding the sparsity specification, the additional experimental results and practical guidance on treating $s$ as a working parameter are helpful and largely address my concerns about robustness to over-specification. The discussion on inference is also clear and appropriately positions this as future work given the challenges associated with nonconvex ℓ₀-constrained estimators.
> >
> > However, the concern regarding dependence on the condition number remains only partially addressed. While the authors mention additional simulations, no concrete results are provided in the rebuttal, making it difficult to assess robustness under ill-conditioned settings. Overall, the rebuttal improves clarity and partially strengthens the empirical support, but some core theoretical and practical questions remain open.

---

> > > ### Author Response · Authors · 2026-04-04
> > >
> > > Thank you for the thoughtful follow-up comments. We are glad that our rebuttal helped clarify and address many of your concerns. Regarding the remaining concern about the method’s dependence on the condition number, we have conducted additional simulation studies stated in our "Response to Weakness 2", and we report the results below.
> > >
> > > Based on the same sparse VAR data-generating process as in Section 4.1, we furthur examine performance under varying degrees of conditioning. Concretely, after generating the coefficient matrices $\\{A_k\\}_{k=1}^d$, we form the companion matrix $\mathcal{A}$ and numerically enforce stability by shrinking the coefficients until the resulting process satisfies $\rho(\mathcal{A}) \leq \gamma$. We then vary $\gamma$ over values increasingly close to $1$, which moves the VAR process closer to the stability boundary and typically produces a more ill-conditioned design covariance $\Sigma_X$. For each generated instance, we compute the corresponding condition number of the lagged design covariance and run the same estimation pipeline as in the main paper.
> > >
> > > **Original setting in Table 1 (Page 7):** $\gamma=0.950$, with theoretical $\\kappa(\\Sigma_X)=1.10$ and empirical $\\kappa(\\hat{\\Sigma}_X)=1.58$.
> > > | Method| Runtime | Train MSE | Test MSE | $\ell_2$-error | Precision | Recall | Sparsity level |
> > > | -|-: |--: |--: |-: |-: |--: |-: |
> > > | LASSO|              0.10   |    0.0412 | 0.0399 | 0.015 | 0.28 | 0.90 |32 |
> > > | IHT$_{(s=10)}$|    0.02 |   0.0414 | 0.0397 | 0.029 |0.90 | 0.90 |10 |
> > > | SP$_{(s=10)}$|    0.02 |    0.0413 | 0.0397 | 0.010 | 0.90 |0.90 |10 |
> > > | CoSaMP$_{(s=10)}$|0.01 |0.0413| 0.0397 | 0.010 | 0.90 | 0.90 | 10 |
> > >
> > > **Additional result 1 with larger conditional number:** $\gamma=0.990$, with theoretical $\\kappa(\\Sigma_X)=4.85$ and empirical $\\kappa(\\hat{\\Sigma}_X)=4.69$.
> > > | Method| Runtime | Train MSE | Test MSE | $\ell_2$-error | Precision | Recall | Sparsity level |
> > > | -|-: |--: |--: |-: |-: |--: |-: |
> > > | LASSO|    0.19              | 0.0415 | 0.0399 | 0.218 | 0.26 | 0.80 |31 |
> > > | IHT$_{(s=10)}$|    0.04 | 0.0413 | 0.0398 | 0.213 | 0.90 | 0.90 |10 |
> > > | SP$_{(s=10)}$|    0.02  |  0.0415 | 0.0398 | 0.212 | 0.80 |0.80 |10 |
> > > |CoSaMP$_{(s=10)}$|0.02 |0.0415| 0.0398 | 0.212 | 0.80 | 0.80 | 10 |
> > >
> > > **Additional result 2 with substantially larger conditional number:** $\gamma=0.999$, with theoretical $\\kappa(\\Sigma_X)=43.53$ and empirical $\\kappa(\\hat{\\Sigma}_X)=47.31$.
> > > | Method| Runtime | Train MSE | Test MSE | $\ell_2$-error | Precision | Recall | Sparsity level |
> > > | -|-: |--: |--: |-: |-: |--: |-: |
> > > | LASSO|              0.22   |    0.0414 | 0.0398 | 0.326 | 0.24 | 0.90 |38 |
> > > | IHT$_{(s=10)}$|    0.09 |   0.0413 | 0.0398 | 0.323 |0.70 | 0.70 |10 |
> > > | SP$_{(s=10)}$|    0.04 |    0.0413 | 0.0398 | 0.319 | 0.80 |0.80 |10 |
> > > | CoSaMP$_{(s=10)}$|0.03|0.0414| 0.0397 | 0.320 | 0.70 | 0.70 | 10 |
> > >
> > > In particular, we compute $\kappa(\Sigma_X)=\frac{\lambda_{\max}(\Sigma_X)}{\lambda_{\min}(\Sigma_X)}$, where $\Sigma_X$ is the population covariance of the lagged design vector and is obtained from the stationary companion-form covariance. We also report the finite-sample analogue $\kappa(\hat{\Sigma}_X)$, where $\hat{\Sigma}_X$ computed from the generated lag vectors. Thus, $\kappa(\Sigma_X)$ measures the intrinsic conditioning of the underlying VAR process, while $\kappa(\hat{\Sigma}_X)$ measures the realized conditioning of the finite sample used in the experiment.
> > >
> > > Overall, these additional results show a clear degradation trend as the condition number increases. In particular, both LASSO and the exact-sparse methods exhibit larger estimation error and some deterioration in support recovery under more ill-conditioned settings, which is consistent with the theory. At the same time, prediction performance remains relatively stable across these settings, suggesting a degree of empirical robustness. We also observe that runtime increases with the condition number, while the exact-sparse methods continue to retain a computational advantage over LASSO.

---

### Official Review · Reviewer_3bei · 2026-03-18

**Soundness:** 4
**Presentation:** 3
**Significance:** 3
**Originality:** 3
**Overall Recommendation:** 5
**Confidence:** 3

**Summary:**

This paper studies constrained least-squares estimation for time series generated by mixing stationary Gaussian processes with sparse coefficients. The authors derive nonasymptotic statistical guarantees and computational complexity results for a class of exact sparse methods. They further apply the theoretical results to Gaussian VAR models and obtain new guarantees.

**Compliance With Llm Reviewing Policy:**

Affirmed.

**Final Justification:**

The paper is clearly written and well organized. Its theoretical contributions are strong. I will maintain my positive assessment.

**Key Questions For Authors:**

Could the authors comment on the optimality of the sample size requirements and error rates in Theorem 3.3?

**Limitations:**

yes

**Strengths And Weaknesses:**

# Strengths

The paper is clearly written and well organized. Its theoretical contributions are strong: it establishes nonasymptotic statistical guarantees for a class of exact sparse methods, such as IHT. The paper also derives new guarantees for Gaussian VAR models, including consistent estimation results and a characterization of how the estimation error depends on interpretable model parameters.

# Weaknesses

The theoretical results could be presented more clearly, particularly regarding the optimality of the sample size requirements and the error rates.

---

> ### Author Rebuttal · Authors · 2026-03-30
>
> Thank you for your positive and thoughtful feedback. We truly appreciate the time and care you invested in reviewing our work. Below, we respond to the specific concern raised.
>
> ---
>
> ### **Response to Weakness (also Key Question for Authors)**
>
> We agree that the discussion of optimality can be stated more clearly, and we will revise the presentation around Theorem 3.3 and Remark 3.4 accordingly.
>
> Our intended interpretation is as follows. Theorem 3.3 provides a nonasymptotic upper bound for $\\|\\theta^k-\\theta^\\ast\\|_2$ of the form
>
> $$
> \\frac{Q S_\\alpha(T)}{\\lambda_{\\min}(\\Sigma_X)}
> \\sqrt{\\frac{(s+s^\\ast)\\log(2pq)}{T}}
> +
> \\sqrt{\\frac{2\\epsilon}{\\lambda_{\\min}(\\Sigma_X)}}
> $$
>
> after a logarithmic number of iterations. In particular, under the regime highlighted in Remark 3.4—namely, summable mixing $S_\\alpha(T) \\le \\tilde{\\alpha} < \\infty$ for a constant $\\tilde{\\alpha}$—this bound reduces to
> $$
> \\|\\theta^k-\\theta^\\ast\\|_2=O \\left(\\sqrt{\\frac{s\\log(pq)}{T}}\\right)
> $$
>
> with sample size requirement $T\\gtrsim s\\log(pq)$.
>
> For comparison, in classical high-dimensional sparse linear regression with exact $s$-sparsity, the minimax $\\ell_2$ estimation benchmark is
>
> $$
> \\Theta\\!\\left(\\sqrt{\\frac{s\\log(d/s)}{n}}\\right),
> $$
>
> where $d$ denotes the ambient dimension and $n$ denotes the sample size; in our setting, $d=pq$ and $n=T$.
> The corresponding classical sample-size requirement is $n \\gtrsim s\\log(d/s)$, up to constant factors, in order to attain the minimax sparse-regression scaling. See, e.g., Raskutti et al. (2011) and Bellec et al. (2018). By contrast, many classical Lasso-type upper bounds are stated at the slightly looser but standard order $n \\gtrsim s\\log d$; see, e.g., Bickel et al. (2009).
>
> Therefore, our result matches the classical sparse-regression benchmark up to the usual $\\log d$ versus $\\log(d/s)$ gap. In this sense, we view the result as *near-minimax* (or *almost optimal*) relative to the classical benchmark.

---

> > ### Author Rebuttal · Reviewer_3bei · 2026-04-01
> >
> > Thank you for your response. I have no further questions.

---

> > > ### Author Response · Authors · 2026-04-02
> > >
> > > Thank you very much for your positive feedback and for your acknowledgment of our rebuttal. We sincerely appreciate your thoughtful review and the time you devoted to evaluating our work.

---

### Decision · Program_Chairs · 2026-04-30

**Decision:**

Accept (regular)

**Comment:**

In this work, the authors study theoretically and empirically the performance of estimators
for sparse regression problems in mixing time series. In particular, building on the substantial
literature in high-dimensional statistics, they establish theoretical guarantees for Iterative Hard Thresholding
and greedy pursuit methods in this setting, and give related results in simulations and ride-sharing data.
All of the reviewers gave this work a positive rating and viewed it as a meaningful contribution to
the time series literature --- therefore, I recommend acceptance.